# In situ generation of micrometer-sized tumor cell-derived vesicles as autologous cancer vaccines for boosting systemic immune responses

Yuxin Guo [1], Shao-Zhe Wang[1], Xinping Zhang[1], Hao-Ran Jia[1], Ya-Xuan Zhu[1], Xiaodong Zhang [1], Ge Gao[1], Yao-Wen Jiang[1], Chengcheng Li[1], Xiaokai Chen[1], Shun-Yu Wu[1], Yi Liu[1] & Fu-Gen Wu [1] ✉

Cancer vaccine, which can promote tumor-specific immunostimulation, is one of the most important immunotherapeutic strategies and holds tremendous potential for cancer treatment/prevention. Here, we prepare a series of nanoparticles composed of doxorubicin- and tyrosine kinase inhibitor-loaded and hyaluronic acid-coated dendritic polymers (termed HDDT nanoparticles) and find that the HDDT nanoparticles can convert various cancer cells to micrometer-sized vesicles (1.6–3.2 µm; termed HMVs) with ~100% cell-to-HMV conversion efficiency. We confirm in two tumor-bearing mouse models that the nanoparticles can restrain tumor growth, induce robust immunogenic cell death, and convert the primary tumor into an antigen depot by producing HMVs in situ to serve as personalized vaccines for cancer immunotherapy. Furthermore, the HDDT-healed mice show a strong immune memory effect and the HDDT treatment can realize long-term protection against tumor rechallenge. Collectively, the present work provides a general strategy for the preparation of tumor-associated antigen-containing vesicles and the development of personalized cancer vaccines.

Cancer immunotherapy has shown great promise in cancer treatment during the past decade[1–6]. Various kinds of cancer immunotherapeutic approaches, such as immune checkpoint blockade (ICB) therapy[7–12], cancer vaccines[13–16], and chimeric antigen receptor T-cell therapy[17,18], have been extensively investigated with desirable outcomes. Among them, cancer vaccine, which can arouse tumor-specific immunostimulation, is one of the most important immunotherapeutic strategies and holds tremendous potential for cancer treatment[19–23]. Besides, the neoantigen- and messenger RNA (mRNA)-based vaccines have achieved encouraging results in cancer patients[21,24,25]. However, despite great efforts made to develop cancer vaccines, they are still in the stage of cancer prophylaxis and evoking substantial immune responses in cancerous patients remains a huge challenge, mainly due to the weak immunogenicity of these vaccines, the immunosuppressive tumor microenvironment (TME), and the low relevance between the antigens in these vaccines and the tumors of specific patients[26,27].

To improve the therapeutic outcomes of cancer vaccines, autologous tumors have been used to produce cancer vaccines (namely personalized cancer vaccines) in vitro or in vivo[28–33]. Benefiting from their tumor-specific tumor-associated antigens (TAAs), the autologous tumor-cell vaccines exhibit a stronger immune response-inducing ability than traditional cancer vaccines[34–36]. However, in vitro preparation of autologous vaccines faces the problems of complex procedure, low yield, and unsatisfactory efficacy, limiting the clinical

[1]State Key Laboratory of Bioelectronics, School of Biological Science and Medical Engineering, Southeast University, 2 Sipailou Road, Nanjing 210096, P. R. China. ✉e-mail: wufg@seu.edu.cn

applications of autologous cancer vaccines. On the other hand, the massive in situ production of autologous tumor-cell vaccines in vivo, which avoids the complicated vaccine preparation procedures in vitro, is a highly desirable choice for producing cancer vaccines.

In this work, we fabricate a series of nanobombs (NBs) comprising doxorubicin (DOX)- and tyrosine kinase inhibitor (TKI)-loaded and hyaluronic acid (HA, for tumor targeting)-coated dendritic polymers (termed HDDT NBs) (Fig. 1). We find that after treating cancer cells with the HDDT NBs, almost all the cells (10−30 μm) are turned into uniform micrometer-sized vesicles (1.6−3.2 μm) with ~100% cell-to-vesicle conversion efficiency (i.e., all the cancer cells could be effectively converted into HDDT-induced micrometer-sized vesicles, HMVs). Although several strategies for cell-derived vesicle production have been reported[37–48], most of them require complicated and rigorous physical stimulation conditions (e.g., extrusion, light irradiation, sonication, heat shock, and freeze-thaw), making it difficult or even impossible to mildly produce cancer cell-derived vesicles in situ. In contrast, our vesicle-producing strategy based on the HDDT NBs, which induces cancer cell death and realizes the in situ massive production of cancer cell-derived vesicles, requires no external stimuli and avoids the strict storage conditions required for the non-in-situ vaccines. Further studies reveal that after systemic administration, the HDDT NBs (taking the HA−dendrimer (Den)−DOX−apatinib (Apa) (HDDA) NBs as an example) can actively accumulate in the tumor regions, restrain the tumor growth in different tumor models, massively induce immunogenic cell death (ICD) and produce HMVs in situ, translate the tumor tissues into the antigen depots, evoke intratumoral and systemic immune responses alone or in combination with ICB therapy, and establish strong immunological memory effects to efficiently protect the cured mice from tumor rechallenge.

## Results

### Design and characterization of different HDDT NBs

Four representative TKIs (i.e., Apa, lapatinib (Lap), sorafinib (Sor), and dasatinib (Das)) were used in our study. Considering that Apa is one of the most widely used TKIs and numerous studies have shown its good biosafety and superb anticancer activity, we chose Apa as the representative TKI in our study. As a representative example, the HDDA NBs were prepared according to the procedure shown in Fig. 1a. First, DOX and Apa were coencapsulated into Den via hydrophobic interaction to form Den−DOX−Apa (DDA) nanoparticles (NPs) with an average diameter of ~35.1 nm (as illustrated by the transmission electron microscopy (TEM) results in Supplementary Fig. 1a, b). Then, the obtained DDA NPs were mixed with HA (110 kDa) to fabricate the HDDA NBs. The encapsulation ratios of Apa and DOX in HDDA NBs were ~85% and ~100%, respectively, and the drug loading coefficients of Apa and DOX were 3.5% and 8.1%, respectively. In addition, HA−Den−DOX−Lap (HDDL), HA−Den−DOX−Sor (HDDS), and HA−Den−DOX−Das (HDDD) NBs were prepared in a similar way. The average diameter of the obtained HDDA NBs was ~35.9 nm (Fig. 2a, b). However, the hydrodynamic diameters of DDA and HDDA were much larger (80.6 ± 45.8 nm and 95.9 ± 31.3 nm) as measured by dynamic light scattering (DLS) (Fig. 2c and Supplementary Fig. 1c), which was probably due to the hydration layers on the surfaces of DDA and HDDA. The polydispersity index (PDI) values of DDA and HDDA were 0.324 and 0.222, respectively, indicating the good aqueous dispersity of the two types of NPs. Next, the zeta potential values of DDA and HDDA were measured to be +17.7 and −16.4 mV, respectively (Fig. 2d), which could be attributed to the amino groups of Den in DDA and the carboxyl groups of HA in HDDA. Besides, the DLS results indicated that the hydrodynamic diameter of the HDDA NBs only became a bit larger and

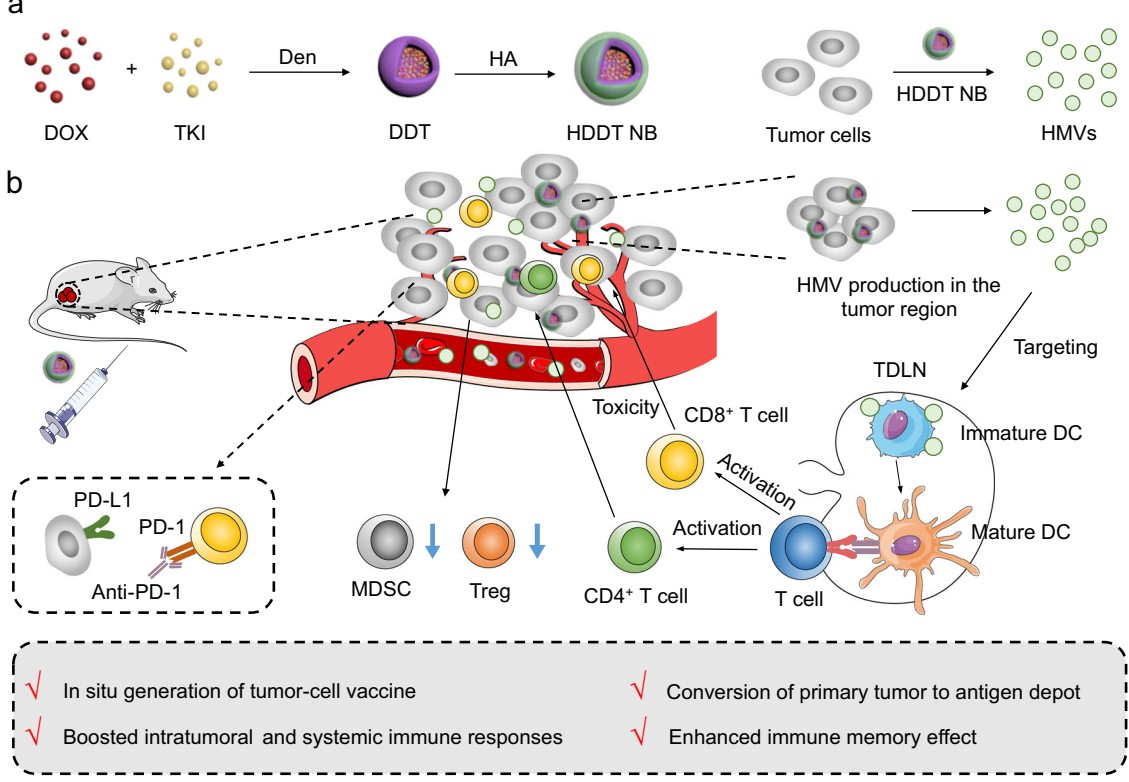

**Fig. 1 | Schematic illustration of the HDDT NB-induced in situ generation of HMVs that act as autologous tumor-cell vaccines for systemic immune responses. a** Fabrication of the HDDT NB and preparation of HMVs in vitro. **b** Scheme of the HDDT NB-induced systemic immune responses for cancer treatment. DOX doxorubicin, TKI tyrosine kinase inhibitor, Den dendrimer, HA hyaluronic acid, HDDT DOX- and TKI-loaded and HA-coated dendritic polymer, NB nanobomb, HMV HDDT-induced micrometer-sized vesicle, TDLN tumor-draining lymph node, DC dendritic cell, MDCS myeloid-derived suppressor cell, Treg regulatory T cell, PD-1 programmed cell death protein 1, PD-L1 programmed cell death ligand 1.

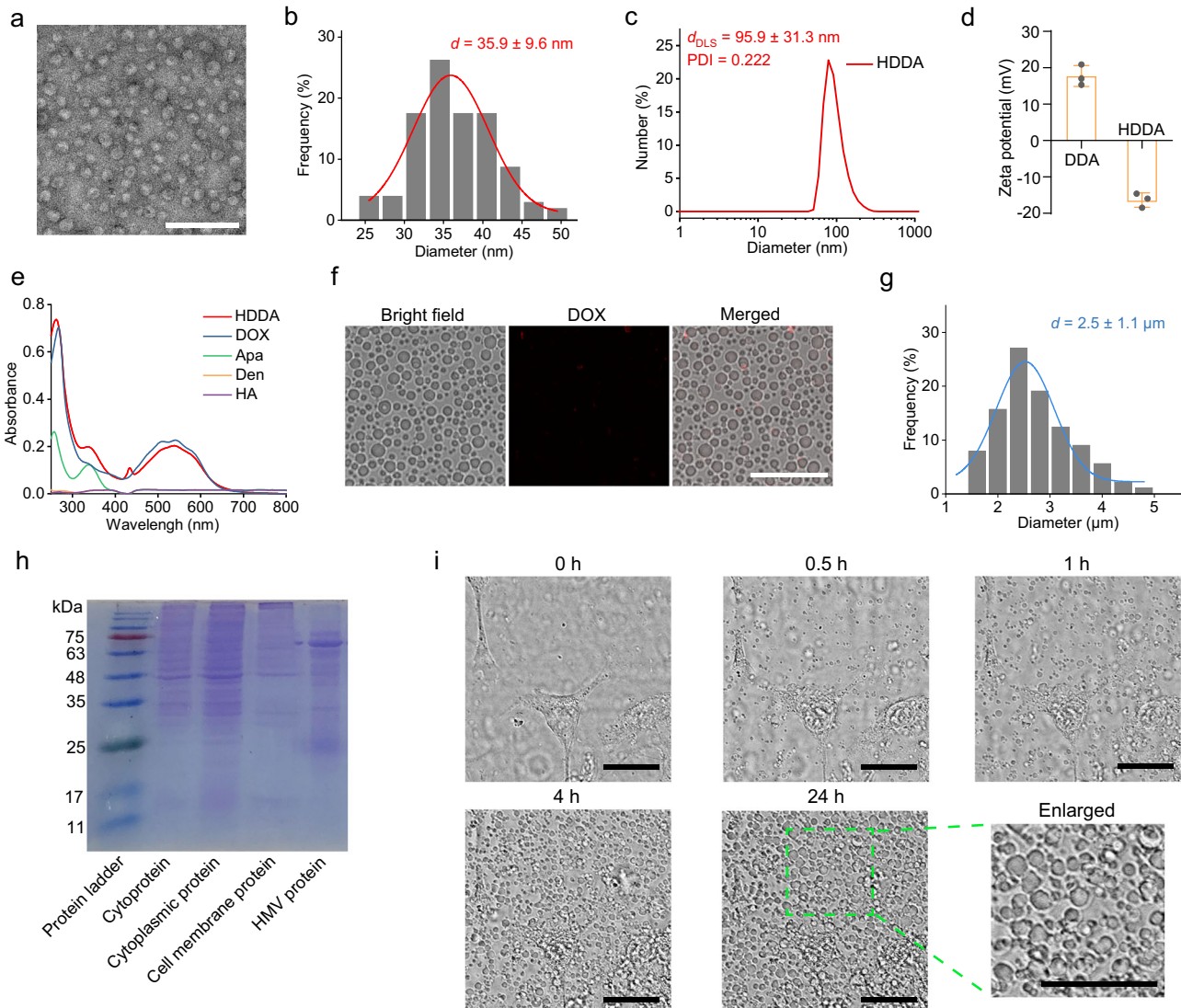

**Fig. 2 | Preparation, characterization, and HMV production of HDDA NBs.**
**a** TEM image and **b** corresponding size distribution histogram of HDDA NBs. Scale bar: 300 nm. **c** Hydrodynamic diameter result of HDDA NBs. **d** Zeta potentials of DDA and HDDA. Data are presented as mean ± standard deviation (SD). $n = 3$ samples per group. **e** Ultraviolet−visible (UV−vis) absorption spectra of the solutions or suspensions containing HA, Den, Apa, DOX, or HDDA. **f** Confocal microscopic images and **g** corresponding size distribution histogram of HDDA-treated (for 24 h) 4T1 cells. Scale bar: 30 μm. **h** SDS-PAGE results of the cytoproteins, cytoplasmic proteins, cell membrane proteins, and HMV proteins in (or from) 4T1 cells. **i** Real-time confocal microscopic images of HDDA-treated 4T1 cells recorded at 37 °C using a live cell imaging system (DOX concentration: 7.2 μg/mL). Scale bars: 30 μm. For **a**−**c** and **e** the experiments were performed independently for three times with similar results. For **f**−**i**, the experiments were performed independently for five times with similar results. Source data are provided as a Source Data file.

the PDI value remained smaller than 0.3 during the 7 d observation period, indicating the good colloidal stability of the HDDA NBs (Supplementary Fig. 2). The ultraviolet−visible (UV−vis) absorption spectra supported the successful fabrication of HDDA (Fig. 2e). Moreover, the HA−Den−DOX (HDD) and HA−Den−Apa (HDA) NPs were also synthesized for comparison purposes.

### Cellular internalization and subcellular distribution of HDDA NBs

To study the anticancer effect of HDDA NBs, we first examined the internalization of HDDA by murine mammary carcinoma (4T1) cells. The time-dependent confocal microscopic images (Supplementary Fig. 3a) and flow cytometric results (Supplementary Fig. 3b, c) indicated that a large number of HDDA NBs were internalized by the 4T1 cells after 0.5 h, and the cellular internalization amounts of the NBs were higher than those of free DOX at the indicated time points. The

enhanced cellular internalization of HDDA was possibly attributed to the tumor cell-targeting capacity of HA since it can actively target the CD44-overexpressed tumor cells. To verify the tumor cell-targeting ability of HDDA, the cellular internalization of HDDA by HMEC-1 (a human dermal microvascular endothelial cell line), NIH 3T3 (a mouse embryo fibroblast cell line), and 4T1 cells was studied. As illustrated by the confocal images, the HDDA-treated 4T1 cells showed stronger red fluorescence signals than the HDDA-treated HMEC-1 and NIH 3T3 cells (Supplementary Fig. 4), indicating the higher internalization level of HDDA by 4T1 cells. Besides, we pretreated the CD44-overexpressed 4T1 cells with HA before the addition of HDDA NBs, for blocking the CD44 molecules on the cell surface. The results indicated that the cell internalization of HDDA NBs was partially influenced by the pretreated HA molecules (Supplementary Fig. 5a), demonstrating that the cell internalization of HDDA NBs was related to the CD44 molecules on the cell surface. Next, we also studied the endocytosis of HDDA NBs using

another CD44-overexpressed tumor cell line MDA-MB-231 (a triple-negative human breast cancer cell line). As shown in Supplementary Fig. 5b, the HDDA endocytosis level of the MDA-MB-231 cells was similar to that of the 4T1 cells. Thus, the above results verified the tumor cell-targeting capacity of the HDDA NBs, which facilitated their further in vivo applications.

Next, the endocytosis mechanism of HDDA NBs was further studied. The 4T1 cells were pretreated with various endocytosis inhibitors to block specific internalization pathways or incubated at 4 °C to inhibit energy-dependent endocytosis, and were then incubated with HDDA NBs. The confocal fluorescence images and flow cytometric results (Supplementary Figs. 6 and 7) indicated that the internalization of HDDA NBs was significantly slowed down in 4T1 cells after incubation at 4 °C, indicating that the endocytosis process was energy-dependent. Besides, methyl-$\beta$-cyclodextrin (M$\beta$CD), 5-(N,N-dimethyl) amiloride hydrochloride (amiloride), and genistein could also partially affect the uptake of HDDA NBs by the cells, while chlorpromazine hydrochloride (CPZ) had little influence on the endocytosis process (Supplementary Figs. 6 and 7), suggesting that the endocytosis process was lipid raft-, macropinocytosis-, and caveolae-dependent and was clathrin-independent, respectively. The above results demonstrated that the internalization of HDDA NBs is a complicated and energy-dependent process.

After cellular internalization, many NPs will be trapped by the endo/lysosomes, which may significantly influence the therapeutic effect of the loaded drugs[49–51]. Thus, the subcellular distribution of HDDA NBs after internalization was studied. As shown in Supplementary Fig. 8, the fluorescence signals of DOX from the HDDA NBs overlapped well with those of LysoTracker Green (a commercial fluorescent probe for lysosomes) at 10 min after NB treatment, demonstrating the entrapment of HDDA by lysosomes. When the incubation time increased to 1 h, the DOX escaped from lysosomes and entered the cell nucleus, indicating the time-dependent lysosomal escape process. The lysosomal escape may be attributed to the "proton sponge" effect of the polyamine molecules of the nanocarriers (e.g., the Den in HDDA NBs)[49,52,53].

### Formation of autologous tumor-cell vaccines

Next, we observed the morphological changes of the 4T1 cancer cells after incubation with the HDDA NBs. Interestingly, as illustrated by the confocal imaging results, almost all the 4T1 cells lost the initial cellular structure and were converted to HMVs after treatment with HDDA NBs for 1 d (Fig. 2f). The average diameter of the HMVs was ~2.5 μm (Fig. 2g), and such a vesicular size was appropriate for further in vivo applications. Besides, we incubated the 4T1 cells with various concentrations of HDDA NBs (DOX concentration: 1.8, 3.6, 5.0, 7.2, 10, and 14.5 μg/mL) and observed the generation of HMVs via confocal microscopy. As shown in Supplementary Fig. 9, massive HMVs were observed in the 7.2 μg/mL group after 4 h incubation. Thus, we chose this concentration for the HMV production. To investigate the components of the HMVs, the analyses of the cytoproteins, cytoplasmic proteins, cell membrane proteins, and HMV proteins were carried out by sodium dodecyl sulfate-polyacrylamide gel electrophoresis (SDS-PAGE) (Fig. 2h). The profile of the HMV proteins was different from those of the other groups, indicating that the HMVs come from various cellular sources. Furthermore, the HMV production process was studied using a live cell imaging system. The real-time confocal microscopic images revealed that the 4T1 cells began to notably produce HMVs at 0.5 h after the HDDA NB treatment, and massive HMV production was observed after 4 h (Fig. 2i). Next, we employed a bicinchonininc acid (BCA) protein assay to quantify the HMV yield (based on protein content) in the HDDA-treated 4T1 cells, which was measured to be ~62% (Supplementary Fig. 10). The very high yield of HMVs is crucial for realizing potential in vitro and in vivo applications. As for whether HMVs contain HDDA NBs, we quantified the DOX and Apa molecules (the toxic

components in the HDDA NBs) in the HMVs via high-performance liquid chromatography (HPLC). The HPLC analysis results indicated that the DOX and Apa contents versus the protein content (which was determined by a BCA protein assay kit) in the HMVs were 0.081% and 0.084%, respectively (Supplementary Fig. 11). Thus, we could confirm that the contents of DOX and Apa in the HMVs were at extremely low levels and could be reasonably neglected.

It is well known that tumor cell lysates have various immunostimulatory components and can act as cancer vaccines to initiate tumor-specific immune responses[54,55]. We wondered if the HMVs might be used as autologous tumor-cell vaccines for cancer immunotherapy. To investigate if the HDDA NBs could also convert other cell lines into HMVs, the NBs were incubated with NIH 3T3 cells, HPAEpiC (human pulmonary alveolar epithelial cells), B16F10 cells (a murine melanoma cell line), MCF-7 cells (a human breast cancer cell line), and MCF-7/ADR cells (adriamycin-resistant MCF-7, a multidrug-resistant human breast cancer cell line). Not surprisingly, these cells were also changed to HMVs with different diameters (Supplementary Figs. 12 and 13). In sharp contrast, the other samples, Apa, DOX, DOX + Apa, HDA, and HDD all failed to convert the 4T1 cells into HMVs (Supplementary Fig. 14). Next, we investigated whether HDDL, HDDD, and HDDS NBs could also change cells into micrometer-sized vesicles. Surprisingly, we found that all the three NBs could also successfully convert the 4T1 cells into HMVs (Supplementary Fig. 15). The above results demonstrated that all the different types of HDDT NBs could induce cancer cells to form HMVs, indicating that the type of TKIs did not affect the HMV induction property of the NBs. In our following studies, the HDDA NBs were chosen as the representative NBs.

### Anticancer effect of HDDA NBs and formation mechanism of HMVs

Then, the therapeutic efficacy of the HDDA NBs was studied using 4T1 cells. Compared with free DOX, free Apa, DOX + Apa, HDA, and HDD, the HDDA NBs showed the highest toxicity toward 4T1 cells (Fig. 3a), which may be attributed to the synergistic effect of DOX and Apa and the elevated cellular uptake of the NBs. Besides, real-time cell analysis (RTCA) experiments were also carried out to monitor the growth of 4T1 cells after different treatments. RTCA is a method capable of conducting uninterrupted, label-free, and real-time analysis of the cells over the course of the experiment. The RTCA results in Fig. 3b showed that the cell index of the HDDA-treated group decreased gradually after drug administration at 17.5 h, and almost all the cells were killed (cell index = 0.66) at 60.8 h. For the HDA- and HDD-treated groups, the growth of cells was also partially inhibited, indicating their moderate toxicity. The Apa could not impede cell proliferation, possibly owing to its poor water solubility. Moreover, the DOX and "DOX + Apa" had good inhibitory effect on cell growth, implying the high cytotoxicity of DOX. Since the intracellular reactive oxygen species (ROS) level is an important marker in the DOX-mediated chemotherapy, the ROS production capacity of various samples in the 4T1 cells was evaluated by the 2',7'-dichlorodihydrofluorescein diacetate (DCFH-DA) ROS probe (Fig. 3c). The HDDA-treated cells showed the highest green fluorescence intensity (around 4-fold higher than that in the control group), indicating the strongest ROS generation capability of HDDA among all the tested samples. Next, the apoptosis/necrosis levels of the 4T1 cells after various treatments were determined via flow cytometry. The results demonstrated that the HDDA-treated cells had the highest apoptosis rate (Q2 + Q4) of 62.42% (Fig. 3d, e), which indicated that apoptosis might be the major pathway of the HDDA-induced cell death. We deduced from the above results that the HMV production was an ROS- and apoptosis-mediated process.

Since the HMV production was accompanied by cell apoptosis, we wondered if the HMVs belonged to the apoptotic bodies (ApoBDs), which usually have diameters of 1–5 μm and high expression levels of annexin V and cleaved caspase-3[56–59]. Therefore, the expression of

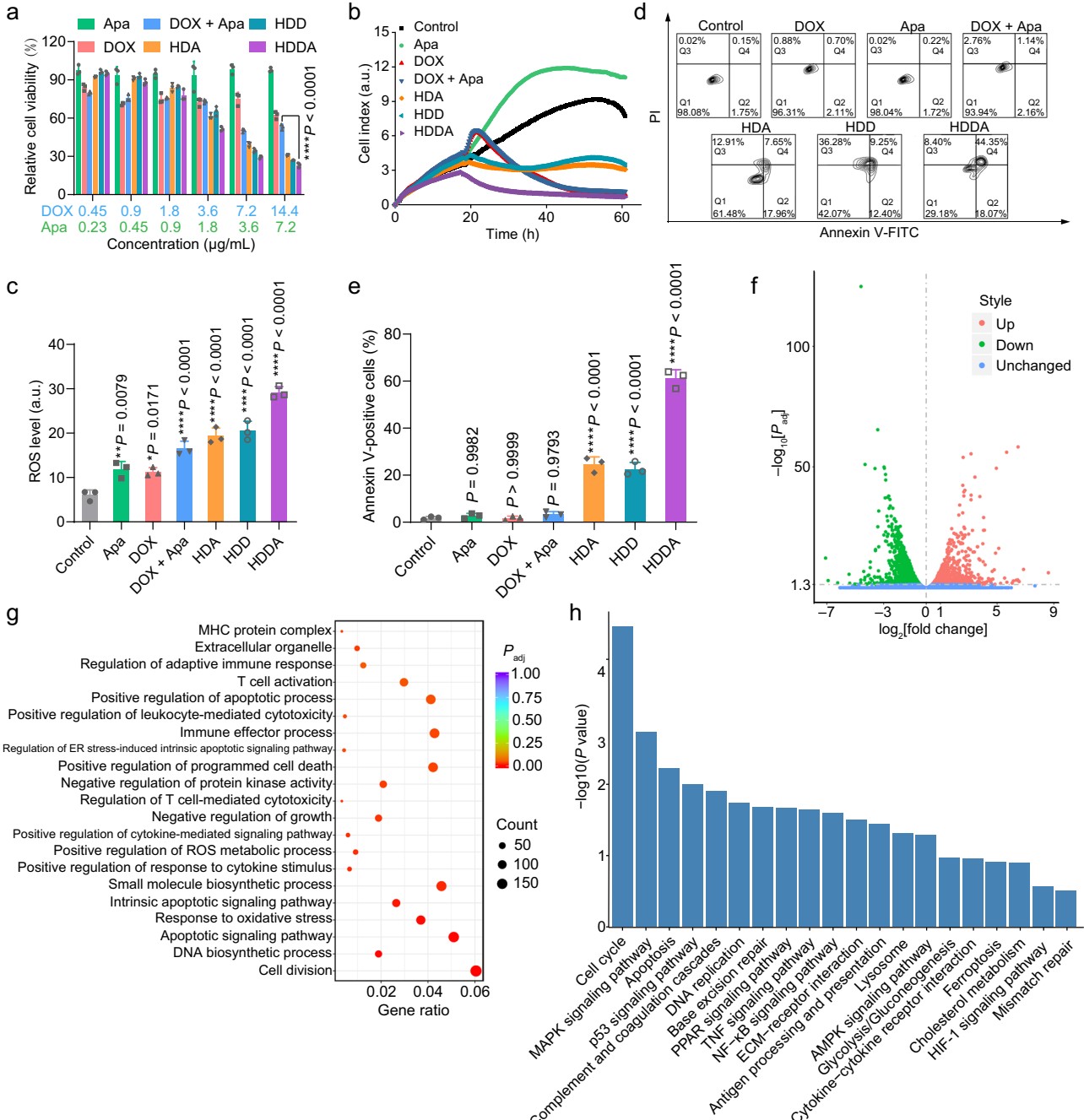

**Fig. 3 | Cytotoxicity of HDDA NBs and transcriptomic analyses of HDDA-treated 4T1 cells. a** Relative viabilities of 4T1 cells after treatment with various amounts of Apa, DOX, DOX + Apa, HDA, HDD, and HDDA, respectively. Data are presented as mean ± SD. $n = 3$ biologically independent samples per group. ****$P < 0.0001$. **b** RTCA results of 4T1 cells after incubation with culture medium (control), Apa, DOX, DOX + Apa, HDA, HDD, and HDDA, respectively. The experiments were performed independently for three times with similar results. **c** Flow cytometric results of 4T1 cells with various treatments as illustrated. The ROS probe DCFH-DA was used to reflect the intracellular ROS content. Data are presented as mean ± SD. $n = 3$ biologically independent samples per group. *$P < 0.05$, **$P < 0.01$, ****$P < 0.0001$. **d** Flow cytometric results and **e** corresponding apoptosis (annexin V-positive) rates of 4T1 cells after different treatments as illustrated. Data are presented as mean ± SD. $n = 3$ biologically independent samples per group. ****$P < 0.0001$. The DOX and Apa concentrations in all the samples in **b**–**e** were fixed at 3.6 and 1.8 μg/mL,

respectively. **f**–**h** Transcriptomic analysis of 4T1 cells after culture medium (control) or HDDA treatment. **f** Volcano plot of all DEGs between the control group and the HDDA group. **g** Dot plot showing the GO enrichment analysis results of selected DEGs between the control group and the HDDA group. $P_{adj}$, adjusted $P$ value using Benjamini–Hochberg correction. MHC major histocompatibility complex, BP biological process, CC cellular component, MF molecular function, **h** Histogram showing the KEGG enrichment analysis results of selected DEGs between the control group and the HDDA group. MAPK mitogen-activated protein kinase, PPAR peroxisome proliferator-activated receptor, ECM extracellular matrix, AMPK adenosine 5′-monophosphate-activated protein kinase, HIF-1 hypoxia-inducible factor-1. For **f**–**h**, $n = 2$ biologically independent samples per group. Statistical significance in **a**, **c**, and **e** was calculated via one-way ANOVA with a Tukey's post-hoc test. Statistical significance in **f**–**h** was calculated via two-tailed Student's $t$-test. Source data are provided as a Source Data file.

annexin V on HMVs was studied by flow cytometry. As shown in Supplementary Fig. 16, the annexin V expression level of HMVs was lower than that of the 4T1 cells without apoptosis, which had a low annexin V expression level, indicating the low annaexin V content of the HMVs. Besides, the Western blot results showed that the caspase-3 and cleaved caspase-3 contents of HMVs were also very low relative to those in the 4T1 cells without apoptosis (Supplementary Fig. 17). However, since the exact definition of ApoBDs is still missing and there is also no consensus on their composition[56–59], we can only claim that cell apoptosis plays an important role in HMV production, while we are not sure whether the HMVs belong to ApoBDs or not.

Then, to figure out the mechanism of HMV formation, the transcriptomic analysis of the 4T1 cells after HDDA treatment was carried out. The 4T1 cells with culture medium treatment were set as the control group. To be specific, DEG (differentially expressed gene) analysis, GO (Gene Ontology) enrichment analysis, and KEGG (Kyoto Encyclopedia of Genes and Genomes) enrichment analysis were carried out to compare the transcriptomes of the culture medium (control)- and HDDA-treated 4T1 cells. After HDDA treatment, 3425 DEGs were detected, among which 1523 DEGs were upregulated and 1902 DEGs were downregulated. The distribution of these DEGs was shown by the volcano plot (Fig. 3f), and the results revealed the significant difference between the control and HDDA groups. As illustrated by the GO enrichment analysis, the DEGs were mainly involved in apoptotic pathways (e.g., apoptotic process, endoplasmic reticulum (ER) stress-induced intrinsic apoptotic signaling pathway, and intrinsic apoptotic signaling pathway), immune response-associated signaling pathways (e.g., T cell activation, adaptive immune response, leukocyte-mediated cytotoxicity, and cytokine-mediated signaling pathway), and ROS-related pathways (e.g., ROS metabolic process and response to oxidative stress) (Fig. 3g and Supplementary Fig. 18). Besides, the GO analysis also revealed that cell growth and some basic cellular functions (e.g., protein kinase activity, small molecule biosynthetic process, microtubule motor activity, and adenosine triphosphatase (ATPase) activity) were influenced after HDDA treatment. More interestingly, the cellular component (CC) analysis showed that various kinds of membrane-containing organelles or membranes, such as extracellular organelle/exosome/vesicle, vacuolar part, late endosome membrane, endocytic vesicle membrane, secretory granule membrane, and filopodium membrane, were all detected in the HDDA-treated 4T1 cell samples (Supplementary Fig. 18), indicating the complicated source of the HMVs. On the other hand, the KEGG enrichment analysis further demonstrated that DEGs of the HDDA-treated cells were enriched in the apoptosis-related pathways (e.g., p53 signaling pathway) and immune response-associated signaling pathways (e.g., cytokine–cytokine receptor interaction, tumor necrosis factor (TNF) signaling pathway, nuclear factor-κB (NF-κB) signaling pathway, and antigen processing and presentation) (Fig. 3h and Supplementary Fig. 19). Collectively, we can confirm that HDDA can induce acute cell apoptosis and influence the basic cellular functions of 4T1 cells. Besides, the finding that the HMVs originate from various kinds of membranes is consistent with the SDS-PAGE result, and the HMV production is an apoptosis-mediated process.

Then to further study the protein profile of HMVs, the proteomic analysis of HMVs was carried out. As one of the most widely used cell debris, the sonication-induced cell debris (termed "Soni" sample) was also subjected to proteomic analysis for comparison purpose. The 4T1 cells were also used as another control group. First, the SDS-PAGE assay was carried out to study the cytoproteins (control), "Soni" sample proteins, and HMV proteins. The results shown in Supplementary Fig. 20 indicated that the protein profile of HMVs was very different from those of the other two samples, demonstrating that HMVs were different from cells and cell debris.

Next, to further study the protein profile of HMVs, the proteomic analyses of 4T1 cells (control), 4T1 cell-derived "Soni" sample, and

HMVs were carried out. The statistical results of the total detected proteins and the Venn diagram revealed that the 4T1 cells (control) and the "Soni" sample contained more types of proteins than the HMVs (Control: 3603, "Soni" sample: 3412, HMVs: 1496) (Supplementary Fig. 21a, b), indicating that a large portion of the proteins were lost during the preparation and isolation process of HMVs. Compared with 4T1 cells, 1970 kinds of proteins were absent in HMVs and 44 kinds of proteins were only present in HMVs (Supplementary Fig. 21c). Besides, compared with 4T1 cells, 288 kinds of proteins were downregulated and 255 kinds of proteins were upregulated in HMVs (Supplementary Fig. 21c). On the contrary, the 4T1 cells and "Soni" sample shared similar protein profiles (Supplementary Fig. 21a–c), indicating that most of the proteins were retained when preparing the "Soni" sample. Then, the profiles of the 1426 kinds of proteins shared by the three samples were shown via the heat map (Supplementary Fig. 21d). The results indicated that the control and "Soni" sample had similar protein profiles with regard to these shared proteins, and the protein profile of HMVs was largely different from those of the other two samples. Then, to further investigate the differences between the HMV group and the other two groups, the cytolocalization of the detected proteins of these three samples was analyzed. The results indicated that HMVs contained the proteins from all the subcellular structures of the cells, and the cytolocalization of the proteins was different in these three samples (Supplementary Fig. 21e–g).

Furthermore, we compared the protein profiles of the control and HMVs groups. First, the representative proteins that could only be detected in the HMVs sample were analyzed (Supplementary Fig. 22a and Supplementary Data 1). Notably, some extracellular matrix (ECM)- and cell adhesion-related proteins were detected in HMVs (e.g., secreted protein, acidic and rich in cysteine (SPARC), transforming growth factor-beta-induced protein ig-h3, prolyl endopeptidase fibroblast activation protein (FAP), neogenin, ECM protein 1, and thrombospondin-4), indicating that the ECM and cell adhesion ability of the cells were influenced during the HMVs-producing process. Then, several protein kinase-related proteins (e.g., epidermal growth factor (EGF)-containing fibulin-like ECM protein 1, protein kinase C-binding protein NELL2, and neurobeachin) were also detected in HMVs, indicating that the TKI Apa (an important component of HDDA NBs) played an important role in HMV production. Moreover, some immune system-related proteins (e.g., prolyl endopeptidase FAP, cell adhesion molecule 1, and complement C1q TNF-related protein 3) in HMVs might promote the immunostimulation effect of HMVs. Next, the proteins that were overexpressed in HMVs were also analyzed (Supplementary Fig. 22b and Supplementary Data 2). Compared with the control group, the protein kinase-related protein levels were elevated in the HMVs group. Besides, some immune system-related proteins (e.g., protein canopy homolog 4, mitogen-activated protein kinase 4, cation-independent mannose-6-phosphate receptor, junctional adhesion molecule A, coxsackievirus and adenovirus receptor homolog, and X-linked inhibitor of apoptosis protein (XIAP)-associated factor 1) were also overexpressed in HMVs, which might be the reason of the strong immunostimulation effect of HMVs. Besides, some membrane-containing organelle (e.g., lysosome, Golgi, and ER)-related proteins were detected in HMVs (Supplementary Fig. 22b and Supplementary Data 2) and the protein content in HMVs from these organelles was also higher than that in control sample (4T1 cells) (Supplementary Fig. 21e, g), indicating that membrane-containing organelles might participate in the production of HMVs.

Moreover, the GO and KEGG enrichment analyses of the control and HMVs groups were conducted. The GO analysis revealed that the proteins related with different kinds of membranes, cytoplasm, and membrane-containing organelles were overexpressed in HMVs (Supplementary Fig. 22c), indicating that the HMVs were derived from different membranes/cytoplasm/organelles. Some immune-related proteins were also detected by the GO and KEGG enrichment

 

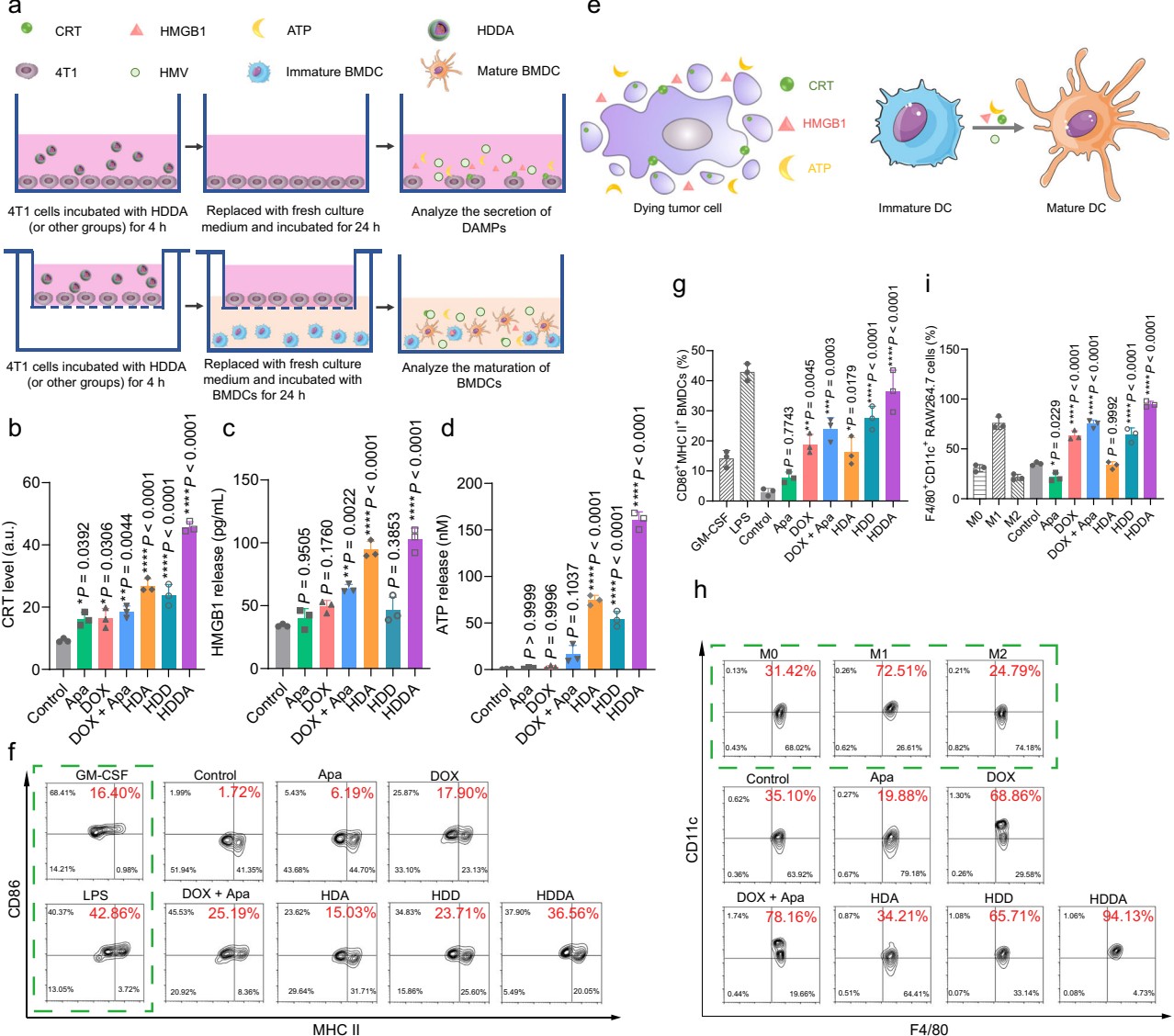

**Fig. 4 | In vitro ICD and immunostimulation triggered by HDDA NBs.**
**a** Schematic illustration of DAMP secretion and ICD-induced BMDC maturation in vitro. **b** Detection of CRT on 4T1 cells by flow cytometry. Quantification of extracellular release of **c** HMGB1 and **d** ATP. **b**–**d** Data are presented as mean ± SD. $n = 3$ biologically independent samples per group. *$P < 0.05$, **$P < 0.01$, ****$P < 0.0001$. **e** Scheme illustrating the mechanism of DC maturation facilitated by the ICD-induced DAMPs (e.g., CRT, ATP, and HMGB1) from dying tumor cells.

Representative flow cytometric plots (**f**, **h**) and relative quantification results (**g**, **i**) of matured DCs (CD11c⁺CD86⁺MHC II⁺) and M1-like macrophages (F4/80⁺CD11c⁺) in vitro. Data are presented as mean ± SD. $n = 3$ biologically independent samples per group. *$P < 0.05$, **$P < 0.01$, ***$P < 0.001$, ****$P < 0.0001$. Statistical significance in **b**–**d**, **g**, and **i** was calculated via one-way ANOVA with a Tukey's post-hoc test. Source data are provided as a Source Data file.

analyses (Supplementary Fig. 22c, d), ensuring the immunostimulation effect of HMVs.

## In vitro ICD effect of HDDA NBs and immunostimulatory effect of HMVs

Since HDDA NBs could induce severe cell lysis and HMV production, we wondered if this kind of cell death belongs to ICD and could enhance the release of danger-associated molecular patterns (DAMPs). Thus, three vital DAMPs, i.e., calreticulin (CRT, which moved from ER to plasma membrane), high-mobility group box 1 protein (HMGB1, secreted from nucleus), and adenosine triphosphate (ATP, released out of cells), were analyzed to study the HDDA-induced ICD effect (Fig. 4a, the upper panel). The confocal fluorescence images (Supplementary Fig. 23) and flow cytometric results (Fig. 4b) demonstrated that the HDDA-treated 4T1 cells had the highest CRT exposure level on the plasma surface, indicating the greater ICD effect of HDDA than

other drugs. Then, the HMGB1 secretion and ATP release amounts were also measured by enzyme-linked immunosorbent assay (ELISA) and a luciferin-based ATP assay, respectively (Fig. 4c, d). Compared with the other groups, the HDDA-treated group showed the highest HMGB1 secretion and ATP release levels. Collectively, the HDDA NBs could induce ICD of 4T1 cells by enhancing the release of DAMPs.

The cancer cells undergoing ICD can produce "eat me" signal (DAMPs) and present it to immature DCs for DC maturation[60] (Fig. 4e), which is responsible for stimulating the immune system. Owing to the increased DAMP release by the HDDA-treated tumor cells (Supplementary Fig. 24), the HMVs could carry these DAMPs (e.g., CRT). Moreover, the HMVs derived from tumor cell lysates might have TAAs on their surface. Thus, HMVs might act as the autologous tumor-cell vaccine for cancer immunotherapy.

Next, the maturation of dendritic cells (DCs) and M1-like repolarization of macrophages induced by the "secretome" of HDDA-treated

cells were investigated. The DC maturation was studied first by using bone marrow-derived dendritic cells (BMDCs). 4T1 cells were seeded into the upper chambers of the transwells (pore size: 8 µm, which allows the HMVs to go through) and pretreated with culture medium (control), DOX, Apa, DOX + Apa, HDD, HDA, and HDDA, respectively. Next, the culture medium was replaced by fresh culture medium, and the immature BMDCs were seeded into the lower chambers of the above transwells (Fig. 4a, the lower panel). The immature BMDCs treated with granulocyte macrophage colony-stimulating factor (GM-CSF) and lipopolysaccharide (LPS) were set as negative control and positive control, respectively. After 24 h incubation, flow cytometry was used to analyze the percentage of matured BMDCs (CD11c$^+$CD86$^+$MHC II$^+$) (Fig. 4f, g). Not surprisingly, a significant proportion of matured BMDCs (36.56%) were observed in the HDDA group, which was slightly lower than that of the positive control (42.86%) and much higher than that of other groups. The high BMDC maturation rate may be attributed to the significant ICD effect of HDDA NBs and the sufficient TAAs and DAMPs on the HMVs. Besides, the DOX + Apa group also had a higher percentage of matured BMDCs (25.19%) than that of the DOX group (17.90%) and the Apa group (6.19%), indicating the synergistic effect of these two drugs. Negligible BMDC maturation (1.72%, much lower than that of the negative control) was observed in the control group (4T1 cells without drug treatment), demonstrating the immunosuppressive effect of the unbroken 4T1 tumor cells.

The phenotype of tumor-associated macrophages (TAMs) is another important marker of immunostimulation level[61,62]. M1-like TAMs exhibit anticancer activity and immunostimulatory effect, while M2-like TAMs promote tumor growth and exhibit immunosuppressive effect. Thus, the repolarization of the M2-like TAMs to the M1 phenotype is a promising strategy to reverse the immunosuppressive TME and improve antitumor immunity. It has been reported that ROS have the ability to repolarize the M2-phenotype TAMs to the M1-phenotype ones[63]. Inspired by the ROS-producing ability of the HDDA NBs, we next investigated whether the HDDA NBs could repolarize the M2-phenotype TAMs by using the M2-like RAW 264.7 mouse macrophages, and the procedure was similar to that shown in Fig. 4a (the lower panel) except that the BMDCs were replaced by RAW 264.7 cells. The culture medium-, interleukin-4 (IL-4)-, and LPS + interferon-gamma (IFN-γ)-treated RAW 264.7 cells were set as M0-, M2-, and M1-like TAMs, respectively (Fig. 4h and Supplementary Fig. 25). A high rate of M1-like TAMs (F4/80$^+$CD11c$^+$) (94.13%) was observed in the HDDA group (Fig. 4h, i), indicating the satisfactory effect of the "secretome" of HDDA-treated 4T1 tumor cells on the repolarization of the M2-like TAMs to the M1 phenotype. The purpose of the above transwell experiments was to simulate the actual situation in the TME, because the immature DCs and M2-like macrophages in the tumor tissue could interact with all the tumor cell-produced "secretome" after HDDA treatment but not only HMVs after HDDA-induced tumor cell death. Next, we further studied the immunostimulation capacity of HMVs in vitro by directly treating the immature DCs and M2-like macrophages with HMVs. The "Soni" sample was used for comparison purpose. As shown in the flow cytometric results, the rates of matured DCs (CD11c$^+$CD80$^+$CD86$^+$) (64.78%) and M1-like macrophages (F4/80$^+$CD11c$^+$) (31.25%) were remarkably increased after HMV treatment (Supplementary Figs. 26 and 27), and the immunostimulation effect of HMVs was stronger than that of the "Soni" sample. Moreover, we also analyzed the levels of various cytokines (e.g., tumor necrosis factor-α (TNF-α), interleukin-1β (IL-1β), and transforming growth factor-β (TGF-β)) secreted by macrophages after different treatments via ELISA assay (Supplementary Fig. 28). The results further proved the immunostimulation effect of the "secretome" of HDDA-treated 4T1 tumor cells.

## In vivo antitumor and immunostimulation effects of HDDA NBs

To assess whether the HDDA NBs could also induce ICD in vivo and make tumor cells produce HMVs for cancer immunotherapy, 4T1 breast tumor-bearing BALB/c mouse models were established. First, the hemolytic effect of the HDDA NBs was studied. The results indicated that the incubation of red blood cells (RBCs) with HDDA NBs only caused little hemolysis even at a DOX concentration of 14.5 µg/mL (Supplementary Fig. 29), confirming the good hemocompatibility of the NBs. Then, since the fluorescence of DOX in HDDA could not be observed clearly by the in vivo fluorescence imaging system, the 1,1'-dioctadecyl-3,3,3'3'-tetramethylindotricarbocyanine iodide (DiR) was used as the model of fluorescent drug payloads and loaded into the HDDA NBs (forming DiR@HDDA NBs). In vivo and ex vivo imaging results indicated that a large number of DiR@HDDA NBs were located in the tumor areas after intravenous injection (Supplementary Figs. 30 and 31), possibly owing to the enhanced permeability and retention (EPR) effect of the NBs and the tumor-targeting ability of HA. Besides, the low fluorescence levels in the major organs and tumor sites of the DiR@HDDA NBs-treated mice at 7 d postinjection demonstrated that the NBs could be excreted by the mice after a certain time period.

Encouraged by the significant in vitro anticancer effect and the satisfactory tumor accumulation of the HDDA NBs, the in vivo antitumor efficacy of the HDDA NBs was evaluated using the 4T1 tumor-bearing BALB/c mice. All the mice were randomly divided into 5 groups and intravenously (i.v.) injected with different drugs for 3 times (Fig. 5a). Notably, the HDDA NB treatment showed the most significant tumor suppression effect than the other drugs and long-term survival for at least 60 days for over 50% of the mice (Fig. 5b–e). Besides, the HDD NP treatment could also inhibit tumor growth, but no mouse could survive for over 60 days. The tumor growth in the DOX and HDA groups was only slightly suppressed. In addition, the representative hematoxylin and eosin (H&E)- and the terminal deoxynucleotidyl transferase (TdT)-mediated dUTP nick end labeling (TUNEL) assay kit-stained tumor tissue slices from different groups are illustrated in Fig. 5f, g. A large proportion of the tumor tissues were destroyed in the HDDA group, demonstrating the significant tumor ablation ability of the HDDA NBs. In contrast, the tumor tissues of the HDD group were partially damaged, while the tumor tissues of the DOX and HDA groups were only slightly influenced. Moreover, the most TUNEL assay kit-stained green fluorescence signals were observed in the HDDA group, indicating the highest rate of apoptotic cells in the tumor tissues from the HDDA-treated mice. Collectively, the HDDA NBs achieved strong antitumor effect in 4T1 tumor-bearing BALB/c mice, possibly owing to their high tumor accumulation, ICD effect, and immunostimulation capability.

Based on the in vitro experiments, we deduced that after the HDDA-induced tumor ablation, the HMVs would be produced and acted as the autologous tumor-cell vaccines for intratumoral and systemic immunostimulation. Therefore, the in situ HMV production in the tumor tissues from the HDDA-treated mice was studied. First, the membrane marker CD44 was used to track the HMVs in tumor tissues. The Western blot analysis and immunofluorescence staining results of the CD44 in HMVs in vitro revealed the high CD44 expression level in the HMVs (Supplementary Figs. 17 and 32). Then, the immunofluorescence staining of the CD44 in tumor tissues after phosphate-buffered saline (PBS) (control) or HDDA treatment was carried out. As illustrated in Fig. 5h, a large number of vesicles were stained by anti-CD44-fluorescein isothiocyanate (FITC) (green) in the tumor slice collected from the HDDA-treated mouse, demonstrating the in situ production of HMVs in the tumor tissues. Besides, TEM was also used to observe the in situ HMV production in the tumor tissues from the HDDA-treated mice. As illustrated by the TEM images, the in situ production of HMVs was observed in the tumor tissues from the HDDA-treated 4T1 tumor-bearing BALB/c mice (Fig. 5i and Supplementary Fig. 33), while the tumor tissues from PBS-treated mice did not contain HMV-like vesicles (Supplementary Fig. 34). In conclusion, we confirmed that HMVs were produced in the tumor tissues after HDDA treatment.

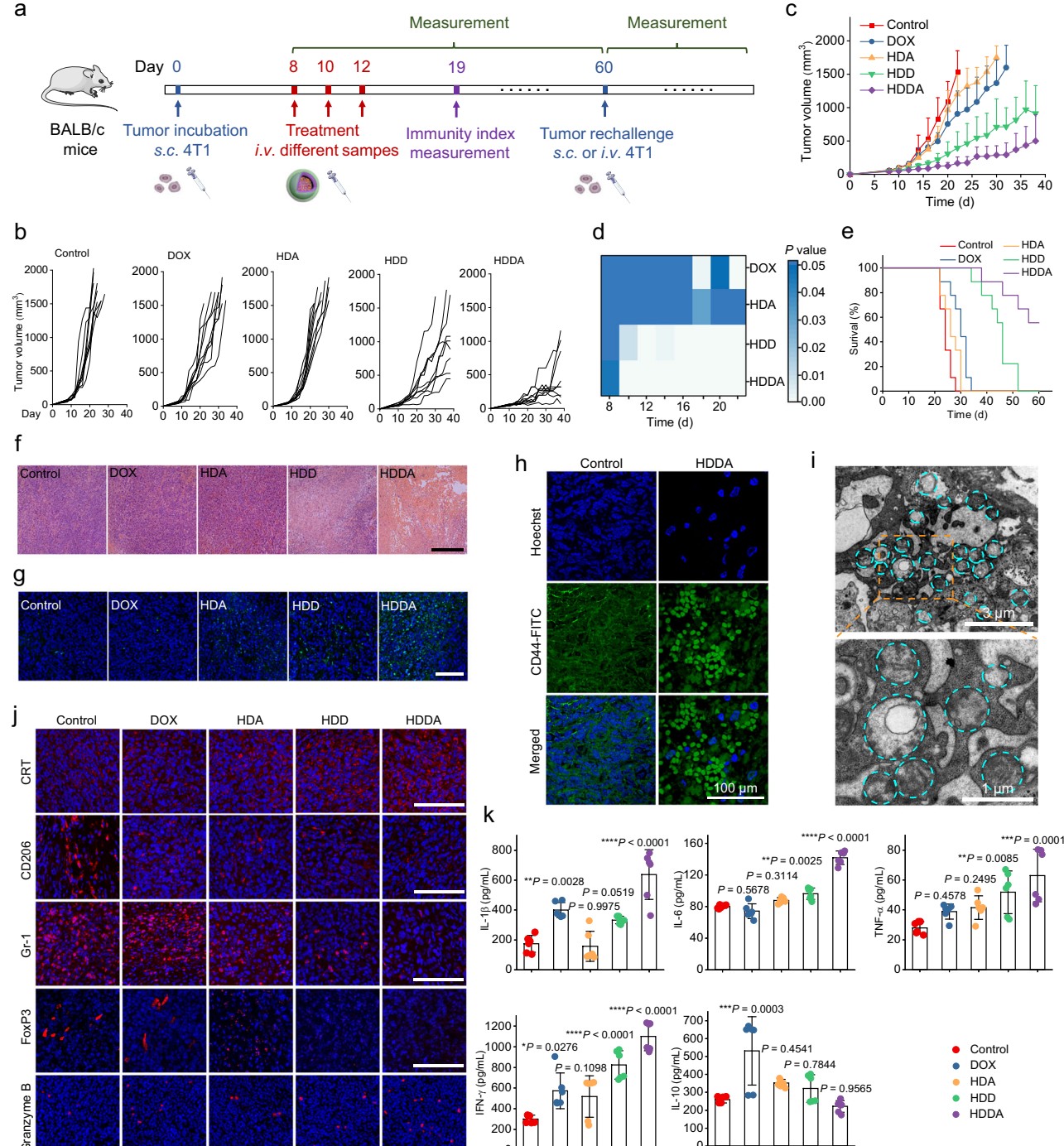

**Fig. 5 | HDDA NBs display high antitumor therapeutic efficacy and strong intratumoral immunostimulatory effect in the 4T1 tumor model.**
**a** Experimental outline showing the treatment steps and the procedures for evaluating the therapeutic and immunostimulation outcomes of different groups in 4T1 tumor-bearing BALB/c mice. **b** Individual 4T1 tumor growth curves of the mice after different treatments. **c** Average tumor growth curves of the 4T1 tumor-bearing BALB/c mice after various treatments and **d** corresponding heat map of the $P$ values between the control group and other groups. The mice treated with PBS were set as the control group. Data are presented as mean + SD. $n = 9$ mice per group.
**e** Survival rates of the 4T1 tumor-bearing BALB/c mice after various treatments.
**f, g** Confocal microscopic images of representative H&E staining results (scale bar: 300 μm) (**f**) and TUNEL assay results (scale bar: 100 μm) (**g**) of the tumor tissue slices from the 4T1 tumor-bearing BALB/c mice sacrificed at day 7 after different treatments. Before imaging, the cell nuclei were stained with Hoechst 33342 (Hoechst). **h** Representative confocal fluorescence images of the CD44 (green)

immunofluorescence staining results of the tumor tissue slices from the 4T1 tumor-bearing BALB/c mice sacrificed at day 3 after PBS (control) or HDDA treatment. CD44 was stained with CD44-FITC antibody. Before imaging, the cell nuclei were stained with Hoechst. **i** Representative TEM images of the tumor tissue from a 4T1 tumor-bearing BALB/c mouse at day 3 after HDDA treatment. The blue dashed circles indicate the positions of HMVs. Note that the enlarged images without the blue dashed circles have also been given in Supplementary Fig. 33. **j** Representative confocal fluorescence images of the immunofluorescence staining results (scale bars: 100 μm) of CRT, CD206, Gr-1, FoxP3, and granzyme B of the tumor tissue slices from the 4T1 tumor-bearing BALB/c mice sacrificed at day 7 after different treatments. For **f**–**j**, $n = 3$ mice per group. **k** Intratumoral expression levels of IL-1β, IL-6, TNF-α, IFN-γ, and IL-10 analyzed by ELISA. Data are presented as mean ± SD. $n = 6$ mice per group. *$P < 0.05$, **$P < 0.01$, ***$P < 0.001$, ****$P < 0.0001$. Statistical significance in **d**, **k** was calculated via one-way ANOVA with a Tukey's post-hoc test. Source data are provided as a Source Data file.

Next, we speculated that the produced HMVs might act as the cancer vaccines for antitumor immunostimulation. To begin with, we examined the ICD effect and intratumoral profiles of DCs, TAMs, T cells, and cytokines, which were known to be involved in antitumor immune responses. As shown in Fig. 5j, the most CRT antibody-stained red fluorescence signals were observed in the HDDA group, indicating the highest ICD level in the tumor tissues from the HDDA-treated mice. As for the cytokines, the ELISA assay results demonstrated that the intratumoral contents of proinflammatory cytokines (e.g., IL-1β, interleukin-6 (IL-6), TNF-α, and IFN-γ) were substantially increased and the intratumoral content of antiinflammatory interleukin-10 (IL-10) was around the baseline level after HDDA treatment (Fig. 5k). The above results indicated that the HDDA-induced tumor ablation in vivo is a proinflammatory process.

It has been reported that cytokines play important roles in regulating some immune cells such as antigen-presenting cells (APCs)[64,65]. Therefore, we next evaluated the immunocellular composition in the TME after different treatments. The results indicated that the HDDA NBs could markedly increase the proportion of the M1-like TAMs (F4/80$^+$CD11c$^+$) which overexpress CD11c (Supplementary Figs. 35 and 36a, c), indicating the immunostimulation effect of the HDDA-induced cell lysis. Accordingly, fewer red fluorescence signals of CD206 were observed in the HDDA-treated tumor tissues (Fig. 5j), implying that the number of M2-like TAMs in the TME was decreased. In addition to M2-like TAMs, two other types of immunosuppressive cells (i.e., myeloid-derived suppressor cells (MDSCs) and regulatory T cells (Tregs)) in the TME were also studied. Similarly, fewer red fluorescence signals of Gr-1 (Ly6G/Ly6C; the marker of MDSCs) and FoxP3 (forkhead box P3; the marker of Tregs) were observed in the tumor tissues from HDDA-treated mice (Fig. 5j), indicating the decreased number of MDCSs and Tregs. In contrast, the Gr-1 and FoxP3 signals were only slightly influenced in the DOX group, possibly due to the poor tumor-targeting and insufficient ICD-inducing ability of free DOX. Besides, the M2-like macrophages (F4/80$^+$CD163$^+$), MDSCs (CD11b$^+$Gr-1$^+$), and Tregs (CD3$^+$CD4$^+$FoxP3$^+$) in the tumor regions were also analyzed by flow cytometry (Supplementary Fig. 37). The results indicated that the proportions of these three kinds of immunosuppressive cells were all decreased in the HDDA group, which was similar to the immunofluorescence staining results shown in Fig. 5j. Moreover, the intratumoral proportion of matured DCs (CD11c$^+$CD86$^+$MHC II$^+$), which can present antigens to T cells, was also elevated in the HDDA group (Supplementary Fig. 36b, d). We then studied if the upregulation of immunoactivation factors and the downregulation of immunosuppression factors by the HDDA NBs in the TME could enhance local antitumor immune responses. The confocal fluorescence images illustrated that the HDDA treatment could increase the intratumoral levels of helper T cells (CD3$^+$CD4$^+$) and cytotoxic T cells (CD3$^+$CD8$^+$) (Supplementary Fig. 38), indicating that the HDDA-induced ICD and immunoactivation could eventually evoke the local antitumor immune responses. Besides, the HDD-treated mice also had a moderately increased number of intratumoral CD4$^+$ and CD8$^+$ T cells, indicating that the HDD NPs could also enhance the local immune responses to some extent. Moreover, the highest granzyme B (a cytotoxic protein produced by CD8$^+$ T cells[66,67]) signal was detected in the tumor tissues from HDDA-treated mice (Fig. 5j), further indicating the strong immunostimulation effect of the HDDA NBs. Collectively, the above results demonstrated that HDDA can convert the immunosuppressive TME to a proinflammatory and tumoricidal one.

### Transcriptomic analysis of the 4T1 tumor tissues of the HDDA-treated mice

Next, to further understand the underlying mechanisms of the antitumor and immunostimulatory effects of HDDA NBs and the HMV production in the tumor tissues of the HDDA-treated mice, we then studied the transcriptomic features of the 4T1 tumor tissues after HDDA NB treatment. The tumor tissues of the PBS-treated 4T1 tumor-bearing BALB/c mice were set as the control group. As shown by the volcano plot and heat map, 1396 DEGs were detected, among which 830 DEGs were upregulated and 566 DEGs were downregulated after the HDDA NB treatment (Fig. 6a, b, the gene names and relative expression levels of the DEGs in the heat map were listed in the Supplementary Data 3), indicating the significant difference between the HDDA group and the control group.

Furthermore, the GO enrichment analysis and the KEGG enrichment analysis of the tumor tissues were performed (Fig. 6c, d, and Supplementary Figs. 39 and 40). As illustrated by the GO enrichment analysis, the tumor cell growth and some basic cellular functions (e.g., growth factor activity, ROS metabolic process, protein secretion, receptor ligand activity, and enzyme inhibitor activity) of the tumor cells were influenced after HDDA treatment (Fig. 6c and Supplementary Fig. 39). Besides, the component and organization of ECM, which are the main factors capable of explaining the formation of the immunosuppressive TME and the poor penetration of the antitumor drugs in the tumor tissues from the HDDA-treated mice, were also influenced (Fig. 6c and Supplementary Fig. 39). More importantly, we found that most of the DEGs in the HDDA-treated group were enriched in inflammatory signaling pathways and immune response-associated signaling pathways, such as macrophage activation, leukocyte-mediated immunity and cytotoxicity, immune effector process, myeloid leukocyte activation and migration, cytokine-mediated signaling pathway, lymphocyte-mediated immunity, innate immune response, natural killer (NK) cell-mediated immunity, granulocyte migration, and the production of IL-6 and other cytokines (Fig. 6c and Supplementary Fig. 39), indicating the immunostimulatory effect of the HDDA NBs. In addition, the CC analysis results of the 4T1 tumor tissues revealed that different kinds of membranes were also influenced after HDDA treatment, and these results were similar to those of the HDDA-treated 4T1 cells (Supplementary Fig. 18). Moreover, the KEGG enrichment analysis further illustrated the alteration of the immune response-associated signaling pathways (e.g., NF-κB signaling pathway, cytokine–cytokine receptor interaction, leukcyte transendothelial migration, toll-like signaling pathway, and cytokine signaling pathway) after HDDA treatment (Fig. 6d and Supplementary Fig. 40). Collectively, the transcriptomic analysis of the tumor tissues of the HDDA-treated mice further revealed that the HDDA-induced tumor cell death can induce immunostimulation.

### HDDA NBs potentiate systemic antitumor immune responses for prevention of tumor recurrence and metastasis

Encouraged by the robust local antitumor immune responses, severe cell lysis, and massive intratumoral HMV production after HDDA treatment, we next investigated if the HMVs could induce systemic antitumor immune responses. The tumor-draining lymph nodes (TDLNs) and spleens were retrieved from 4T1-bearing BALB/c mice 7 d after various treatments. As illustrated by the flow cytometric results (Fig. 7a, d and Supplementary Fig. 41), the mice after HDDA treatment showed the strongest capacity in the enrichment of matured DCs (CD11c$^+$CD86$^+$MHC II$^+$) in TDLNs and spleens among all the groups. Besides, the HDDA NB treatment could also markedly elevate the proportion of the M1-like TAMs (F4/80$^+$CD11c$^+$) in TDLNs and spleens (Fig. 7b, d and Supplementary Fig. 41), which also demonstrated the systemic immunostimulation effect of the HDDA NBs. Next, to study the systemic antitumor T-cell immune responses, the T cells collected from TDLNs and spleens were also analyzed by flow cytometry. Not surprisingly, the rates of CD4$^+$ and CD8$^+$ T cells in the TDLNs of the mice in the HDDA group were approximately 2-fold higher than those in the control group, and the HDDA treatment also increased the CD4$^+$ and CD8$^+$ T cells in spleens (Fig. 7c, d and Supplementary Figs. 41 and 42).

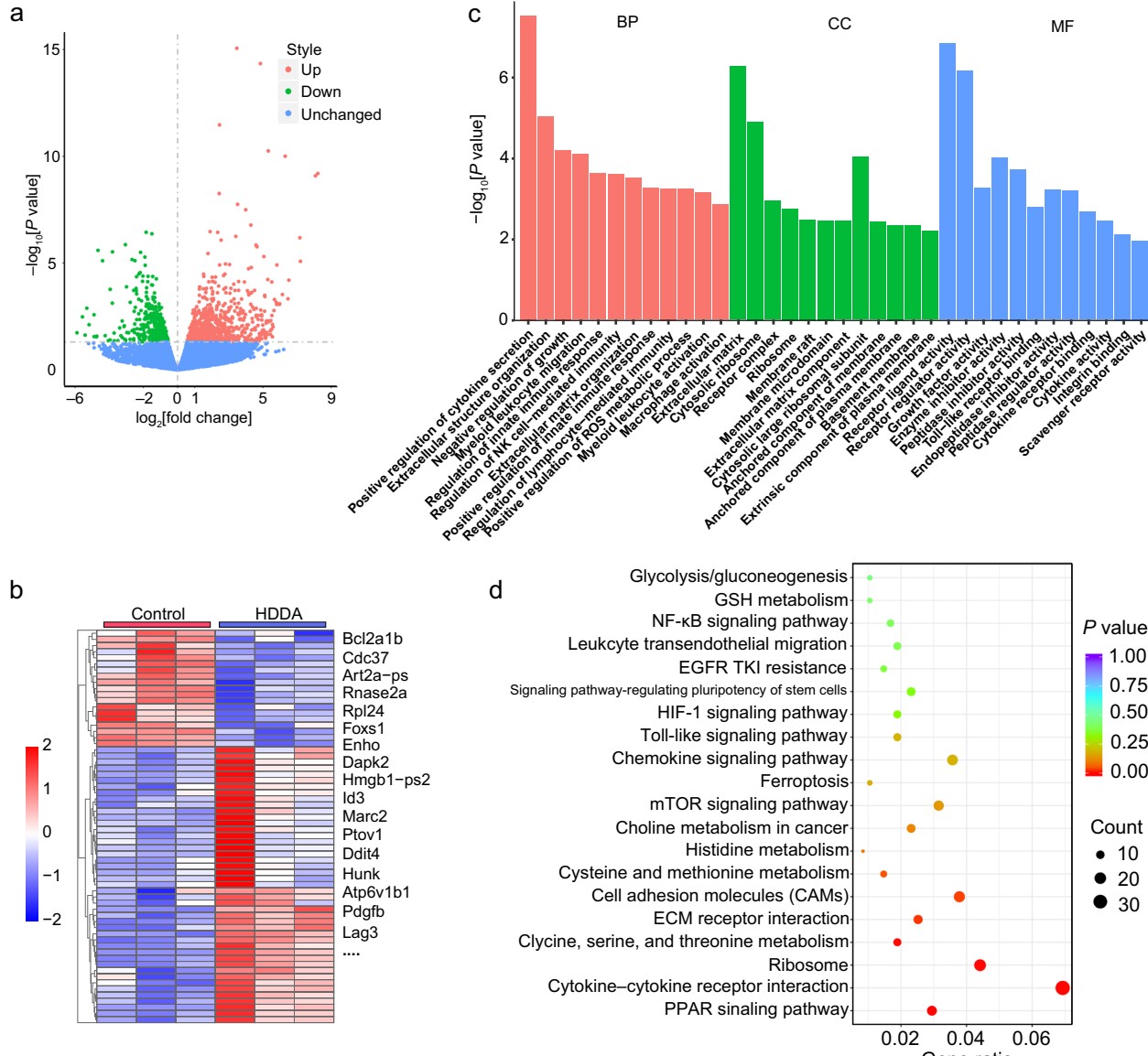

**Fig. 6 | Transcriptomic analysis of 4T1 tumor tissues after HDDA treatment. a** Volcano plot of all DEGs between the control group and the HDDA group. **b** Heat map of the selected DEGs between the control group and the HDDA group. Red: upregulation; blue: downregulation; false discovery rate <0.05. Some key DEGs are listed on the right of the map. **c** Histogram showing the GO enrichment analysis results of some selected DEGs between the control group and the HDDA group. BP biological process, CC cellular component, MF molecular function. **d** Dot plot showing the KEGG enrichment analysis results of some selected DEGs between the control group and the HDDA group. GSH glutathione, EGFR epidermal growth factor receptor, HIF-1 hypoxia-inducible factor-1, mTOR mammalian target of rapamycin. *n* = 3 mice per group. Statistical significance in **a**, **c**, and **d** was calculated via two-tailed Student's *t*-test.

Besides, the concentrations of various proinflammatory and antiinflammatory cytokines in serum were measured. The levels of proinflammatory cytokines (e.g., IL-1β, IL-6, TNF-α, and IFN-γ) in serum were significantly elevated and the antiinflammatory IL-10 was close to the baseline level after HDDA treatment (Fig. 7e). In addition, the HDDA treatment could also make the swollen spleen of the 4T1 tumor-bearing mice return to the baseline level at day 14 after treatment (Fig. 7f, g), possibly due to the reduced inflammation, indicating that the HDDA-induced systemic inflammation could be eventually reduced to the normal level, which is important for the safe use of HDDA NBs.

Collectively, the HDDA treatment could induce strong antitumor systemic immune responses.

In addition, to study if the HDDA-induced systemic immune responses were durable, we rechallenged the 4T1 tumor-bearing mice that survived after the first HDDA treatment with 4T1 tumor cells by either subcutaneous or intravenous injection. Notably, in the HDDA group, the growth of the subcutaneous tumors of the mice that were subcutaneously (s.c.) injected with tumor cells was almost completely suppressed, all the mice in the HDDA group survived within the 40 d observation period, and no tumor nodules were observed in the lungs of the mice that were i.v. injected with tumor cells (Fig. 7h–k), indicating that HDDA could induce durable antitumor immune responses. Besides, we further studied the central memory T cells (T$_{CM}$, CD3$^+$CD8$^+$CD44$^+$CD62L$^+$) and effector memory T cells (T$_{EM}$, CD3$^+$CD8$^+$CD44$^+$CD62L$^-$) in the spleens of the 4T1-bearing BALB/c mice 30 d after different treatments. The flow cytometric results indicated that the percentages of T$_{EM}$ and T$_{CM}$ in the CD3$^+$CD8$^+$ T cells in the HDDA group showed a significant increase compared with those in the control mice (Supplementary Fig. 43). Together, we could confirm the strong immune memory effect of the HDDA-treated mice.

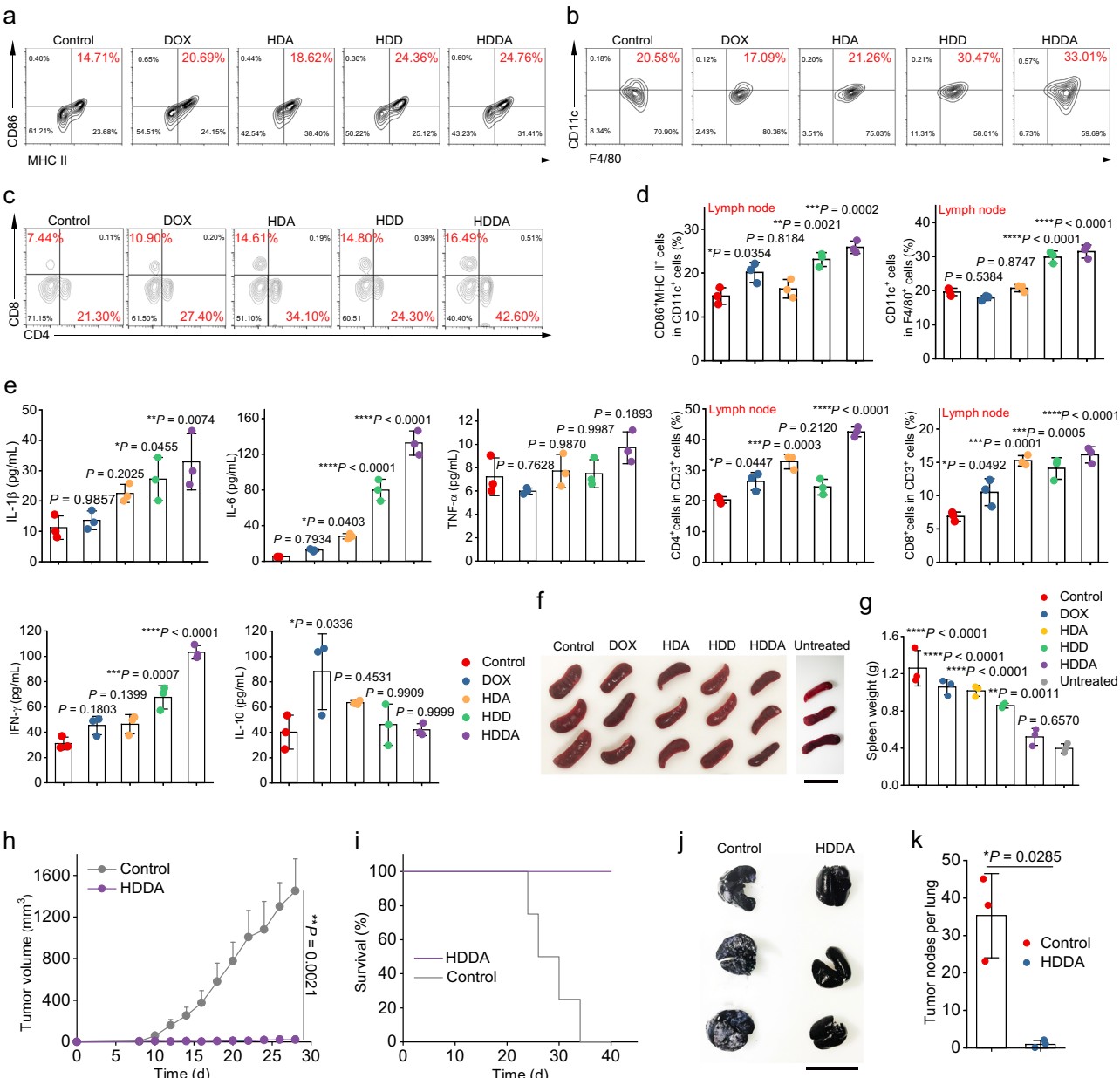

**Fig. 7 | HDDA NBs trigger systemic immune responses and durable immuno-logical memory in 4T1 tumor-bearing BALB/c mice. a–d** Representative flow cytometric plots (**a–c**) and relative quantification results (**d**) of matured DCs (CD11c⁺CD86⁺MHC II⁺), M1-like macrophages (F4/80⁺CD11c⁺), helper T cells (CD3⁺CD4⁺), and cytotoxic T cells (CD3⁺CD8⁺) in the lymph nodes retrieved from the 4T1-bearing BALB/c mice 7 d after different treatments. Data are presented as mean ± SD. $n = 3$ mice per group. *$P < 0.05$, **$P < 0.01$, ***$P < 0.001$, ****$P < 0.0001$. **e** Concentrations of IL-1β, IL-6, TNF-α, IFN-γ, and IL-10 in serum (collected from the mice 2 d after different treatments) analyzed by ELISA. Data are presented as mean ± SD. $n = 3$ mice per group. *$P < 0.05$, **$P < 0.01$, ***$P < 0.001$, ****$P < 0.0001$. **f, g** Photographs (**f**) and corresponding weights (**g**) of the spleens retrieved from the mice 14 d after different treatments. **$P < 0.01$, ****$P < 0.0001$. Scale bar: 3 cm. Data are presented as mean ± SD. $n = 3$ mice per group. **h, i** Tumor growth curves (**h**) and survival rates (**i**) after subcutaneous rechallenge (i.e., the HDDA-healed 4T1 tumor-bearing mice were s.c. injected with 4T1 tumor cells). The healthy mice that were s.c. injected with 4T1 tumor cells were set as the control group. Data are presented as mean + SD. **$P < 0.01$. **j, k** Photograph of India ink-stained lungs (the white nodules indicate the tumor nodules) (**j**) and quantification results of the tumor nodules in lungs (**k**) retrieved from the healthy (control) or HDDA-treated BALB/c mice after intravenous rechallenge of 4T1 cells. Data are presented as mean ± SD. *$P < 0.05$. Scale bar: 3 cm. $n = 3$ mice per group. Statistical significance in **d**, **e**, and **g** was calculated via one-way ANOVA with a Tukey's post-hoc test. Statistical significance in **h**, **k** was calculated via two-tailed Student's $t$-test. Source data are provided as a Source Data file.

Further, the in vivo biocompatibility of the HDDA NBs was analyzed. The PBS-treated mice were set as the control group. The results showed that neither abnormal changes in the routine blood analysis and biochemical analysis results nor tissue damage in the histopathologic images of the major organs in the HDDA-treated mice were observed (Supplementary Figs. 44–49). Besides, only sporadic TUNEL staining signals (green) were detected in the tissue slices of the major organs in the HDDA

group, demonstrating that HDDA did not cause cell apoptosis in normal tissues (Supplementary Fig. 50). The urine routine test was also carried out, and all the urine routine indexes in the HDDA group were similar to those in the control group and remained in the normal ranges (Supplementary Fig. 51), further ensuring the good biosafety of the HDDA NBs. In addition, there was no apparent body weight loss during the 30-day observation period in all the groups (Supplementary Fig. 52). Together, the HDDA NBs

showed excellent in vivo biocompatibility, ensuring their potential clinical applications.

## HMVs act as autologous cancer vaccines to induce systemic anticancer immunostimulation for tumor prevention in vivo

To investigate if the HMVs could act as autologous tumor-cell vaccines and play an essential role in inducing the systemic antitumor immune responses, we first prepared and purified HMVs from 4T1 cells in vitro, and labeled them with 1,1'-dioctadecyl-3,3,3',3'-tetramethylindodicarbocyanine perchlorate (DiD) (forming DiD@HMVs). The BALB/c mice were s.c. injected with DiD@HMV suspension (100 μL, protein concentration: 10 mg/mL) (Supplementary Fig. 53a) and the lymph nodes were collected 2 d after injection. The flow cytometric results illustrated that 47.8% of the cells from the lymph nodes had the fluorescence signals of DiD (Supplementary Fig. 53b), indicating that the DiD@HMVs could target and enter the cells in the lymph nodes after subcutaneous injection. As for the reasons for lymph node targeting, we consulted a review[68], which claimed that after intramuscular or subcutaneous injection, the activated DCs and the vaccines can travel through afferent lymph vessels to the draining lymph nodes. We expect that the HMVs could also activate the DCs at the injection site and travel through afferent lymph vessels with the help of and together with the activated DCs to the draining lymph nodes. Besides, the appropriate particle size of HMVs may also facilitate the lymph node traveling process. Besides, the maturation of the DCs and the types of T cells in the lymph nodes were also tested by flow cytometry. The results indicated that the mice after HMV treatment showed significant enrichment of matured DCs (CD11c$^+$CD86$^+$CD80$^+$) in lymph nodes (Supplementary Fig. 53c). Further, we found that the rates of cytotoxic T cells (CD3$^+$CD8$^+$) and helper T cells (CD3$^+$CD4$^+$) in the HMVs group were also markedly increased (Supplementary Fig. 53d).

To further validate the role of HMVs in the systemic antitumor immune responses in a melanoma model, we also prepared DiD@HMVs from the HDDA-treated B16F10 cells. The C57BL/6 mice were s.c. injected with DiD@HMVs (100 μL, 10 mg/mL) (Supplementary Fig. 53e) and the immunocellular compositions in lymph nodes were analyzed. Similar to the results in the BALB/c mice, the DiD@HMVs could also target the cells in the lymph nodes after subcutaneous injection (Supplementary Fig. 53f). In addition, the rates of matured DCs (CD11c$^+$CD86$^+$CD80$^+$) and cytotoxic T cells (CD3$^+$CD8$^+$) in lymph nodes after DiD@HMV treatment were also elevated (Supplementary Fig. 53g, h), indicating that the HMVs could act as autologous tumor-cell vaccines for cancer immunotherapy. The above results illustrated that the HMVs could act as autologous tumor-cell vaccines for immunostimulation.

Moreover, the prevention of tumor growth in the BALB/c mice inoculated with PBS (control), HMVs, or the "Soni" sample was evaluated. Compared with the other two groups, the HMVs showed the most significant tumor suppression during the 30 d observation period with negligible body weight loss (Supplementary Fig. 54a–d). In addition, a large proportion of the tumor tissues was destroyed in the HMVs group as revealed by the representative H&E- and TUNEL assay kit-stained tumor tissue slices (Supplementary Fig. 54e, f). Collectively, we can claim that the HMVs are responsible for the observed antitumor responses and can act as cancer vaccines for immunostimulation and tumor prevention in vivo.

## HDDA NBs delay tumor growth and potentiate ICB therapy in a highly aggressive melanoma tumor model

Encouraged by the immunostimulation effect of the HMVs in the C57BL/6 mice, we further studied the anticancer performance of HDDA NBs using a highly aggressive and poorly immunogenic B16F10 melanoma model, which is more challenging for immunotherapy.

Besides, ICB therapy (i.e., programmed cell death protein 1 (PD-1) blockade therapy) was also introduced to enhance the immunostimulation effect of the HDDA NBs. The experiments were performed using the B16F10 tumor-bearing C57BL/6 mice, which were randomly divided into five groups (n = 10/group) and i.v. injected with various samples (e.g., PBS, DOX, HDA, HDD, and HDDA) for 3 times, respectively (Fig. 8a). On the next day of each injection, half of the mice in each group were intraperitoneally (i.p.) injected with the antimouse PD-1 monoclonal antibody (anti-PD-1, 10 mg/kg). The tumor sizes in different groups were measured every other day. The results illustrated that the anti-PD-1 treatment alone only slightly suppressed tumor growth (Fig. 8b–e), possibly due to the limited T cell infiltration in the immunosuppressive TME of B16F10 melanoma. In contrast, PD-1 blockade by anti-PD-1 antibody could enhance the antitumor effect of HDD and HDDA NBs, and the "HDDA + anti-PD-1" treatment showed a long-term survival period of at least 40 days for 40% of the mice (Fig. 8b–e). In addition, the representative H&E- and TUNEL assay kit-stained tumor tissue slices from different groups were shown in Supplementary Figs. 55 and 56. Severe damage of the tumor tissues was observed in the "HDDA + anti-PD-1" group (Supplementary Fig. 55), demonstrating the significant tumor ablation effect of HDDA plus anti-PD-1. Besides, the tumor tissues of the HDD and "HDD + anti-PD-1" groups were partially destroyed, while the tumor tissues of the HDA, HDD, "DOX + anti-PD-1", and "HDA + anti-PD-1" groups were only slightly influenced (Supplementary Fig. 55). On the other hand, the "HDDA + anti-PD-1" group also showed the most TUNEL assay kit-stained green fluorescence signals (Supplementary Fig. 56), implying the highest apoptosis level of the "HDDA + anti-PD-1"-treated tumor tissues among all the groups. Collectively, the "HDDA + anti-PD-1" exhibited excellent therapeutic ability toward B16F10 melanoma. We deduced from the above results that the HDDA NBs could induce the ICD of B16F10 tumor cells and produce HMVs for immunostimulation, which would stimulate DCs, improve the intratumoral T cell infiltration, and further potentiate PD-1 blockade therapy.

Further, to verify our above conjecture, we studied the antitumor immune responses of different groups. First, the immunocellular compositions in the TME after various treatments were evaluated. After "HDDA + anti-PD-1" treatment, the fluorescence signals of the markers of immunosuppressive MDSCs and Tregs (Gr-1 and FoxP3) were significantly decreased (Supplementary Figs. 57 and 58), indicating that the immunosuppressive TME was changed by the "HDDA + anti-PD-1" treatment. Besides, the rate of matured DCs (CD11c$^+$CD86$^+$CD80$^+$) in the TME after "HDDA + anti-PD-1" treatment was remarkably elevated (Supplementary Figs. 59 and 60). Since the matured DCs could present antigens to T cells, the rates of the intratumoral helper T cells (CD3$^+$CD4$^+$) and cytotoxic T cells (CD3$^+$CD8$^+$) in the "HDDA + anti-PD-1" group were further increased (Supplementary Fig. 60), implying the elevated local antitumor immune responses. Moreover, after "HDDA + anti-PD-1" treatment, the intratumoral concentration of proinflammatory cytokines (e.g., IL-1β, IL-6, TNF-α, and IFN-γ) were increased and the antiinflammatory IL-10 was slightly decreased (Supplementary Fig. 61). The above results demonstrated that the combined treatment of HDDA and anti-PD-1 could induce strong local antitumor immune responses in the B16F10 melanoma model.

Next, to study if the HDDA NBs could elicit the systemic antitumor immune responses in the B16F10 melanoma model, the immunocellular compositions in TDLNs and spleens after various treatments were evaluated. Not surprisingly, a significant enrichment of matured DCs (CD11c$^+$CD86$^+$CD80$^+$) in TDLNs and spleens was observed in the "HDDA + anti-PD-1" group (~2.2-fold and ~1.7-fold increase relative to the control group, respectively) (Fig. 8f, g and Supplementary Fig. 62). As for the T cells, after "HDDA + anti-PD-1" treatment, the rates of CD4$^+$ and CD8$^+$ T cells retrieved from TDLNs were approximately 2.4- and

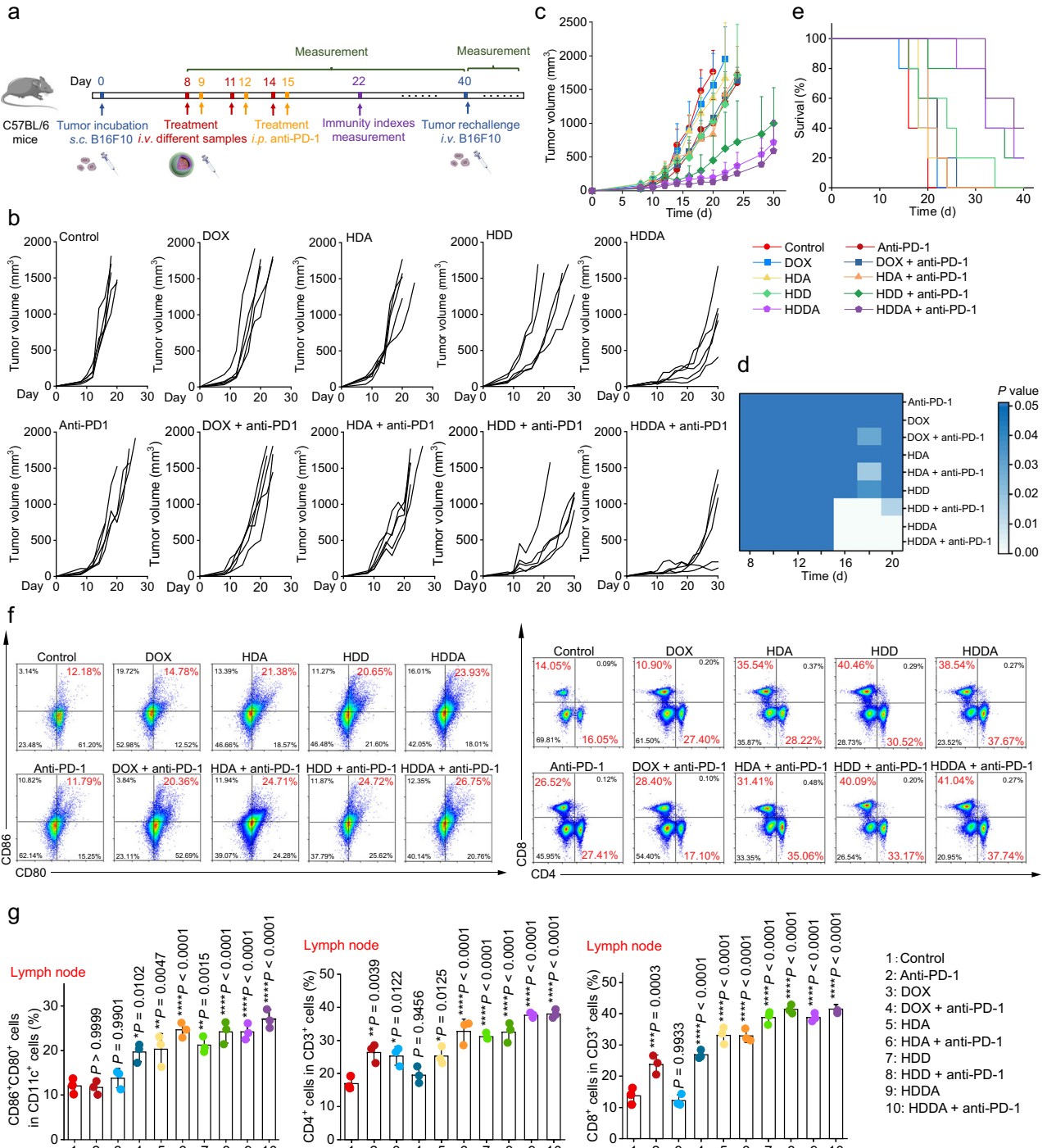

**Fig. 8 | HDDA NBs exhibit high antitumor therapeutic efficacy, systemic immune responses, and durable immunological memory in the B16F10 tumor model. a** Experimental outline showing the treatment steps and procedures for evaluating the therapeutic and immunostimulation outcomes of different groups in B16F10 tumor-bearing C57BL/6 mice. **b** Individual B16F10 tumor growth curves of the mice after different treatments. **c** Average tumor growth curves of the B16F10 tumor-bearing C57BL/6 mice after various treatments and **d** corresponding heat map of the $P$ values between the control group and other groups. The mice treated with PBS were set as the control group. Data are presented as mean + SD. $n = 5$ mice per group. **e** Survival rates of the B16F10 tumor-bearing C57BL/6 mice after various treatments. **f, g** Representative flow cytometric plots (**f**) and relative quantification results (**g**) of matured DCs (CD11c⁺CD86⁺CD80⁺), helper T cells (CD3⁺CD4⁺), and cytotoxic T cells (CD3⁺CD8⁺) in the lymph nodes retrieved from the B16F10 tumor-bearing C57BL/6 mice 7 d after different treatments. Data are presented as mean ± SD. $n = 3$ mice per group. *$P < 0.05$, **$P < 0.01$, ***$P < 0.001$, ****$P < 0.0001$. Statistical significance in **d**, **g** was calculated via one-way ANOVA with a Tukey's post-hoc test. Source data are provided as a Source Data file.

2.9-fold higher than those in the control group (Fig. 8f, g), and similar results were also obtained in spleens (Supplementary Fig. 62). Additionally, the concentrations of various proinflammatory and antiin-flammatory cytokines in serum were analyzed. Compared with the control group, substantial increase of IL-1β, IL-6, TNF-α, and IFN-γ and evident decrease of IL-10 were observed in the serum samples of the mice treated with "HDDA + anti-PD-1" (Supplementary Fig. 63). The above results confirmed that "HDDA + anti-PD-1" could induce strong

systemic antitumor immune responses in the poorly immunogenic melanoma model.

Furthermore, the B16F10 tumor-bearing C57BL/6 mice that survived after the first "HDDA + anti-PD-1" treatment were rechallenged with B16F10 tumor cells by intravenous injection. Notably, no tumor nodules in the lungs were observed after intravenous rechallenge (Supplementary Figs. 64 and 65), demonstrating the strong immune memory effect in the "HDDA + anti-PD-1"-healed mice. Importantly, there was also no apparent body weight loss in different groups during the observation period (Supplementary Fig. 66). Besides, routine blood analysis results and histopathologic images of major organs showed no abnormal changes after "HDDA + anti-PD-1" treatment (Supplementary Figs. 67 and 68). Collectively, these biosafety assay results demonstrated the satisfactory in vivo biocompatibility of the "HDDA + anti-PD-1" treatment.

## Discussion

In this study, we developed a simple and effective strategy based on the HDDT NBs (e.g., HDDA, HDDL, HDDS, and HDDD NBs) to induce robust cancer cell death and realize the in situ massive production of HMVs without the need of external stimuli like light irradiation, sonication, and electric field. Importantly, we found that the cancer cells could be turned into HMVs (average size: 1.6–3.2 μm) with ~100% cell-to-HMV conversion efficiency. We also found that the HDDT NBs (taking the HDDA NBs as an example) could accumulate in the tumor region, induce robust ICD, reverse the immunosuppressive TME, and massively produce HMVs. As a result, the tumor tissue was transformed into a depot of autologous vaccines that triggered the systemic tumor-specific immune responses (e.g., maturation of DCs, enrichment of M1-like TAMs and cytotoxic and helper T cells, and establishment of immunological memory). Moreover, we found that the HDDA NBs-triggered systemic tumor-specific immune responses could be further enhanced by ICB therapy. As shown in two tumor models, the HDDA NBs could restrain the tumor growth and evoke intratumoral and systemic immune responses alone or in combination with ICB therapy. In addition, the HDDA NBs could establish strong immunological memory in the cured mice, which could efficiently protect the cured mice from tumor rechallenge.

On the other hand, traditional cancer vaccines containing specific antigens have poor therapeutic outcomes owing to the low relevance between the antigens in these vaccines and the tumors of specific patients. In our strategy, the primary tumor itself acts as the depot of the in situ autologous tumor-cell vaccines containing all the tumor-specific DAMPs and TAAs for immunostimulation. In addition, the massive and highly efficient in situ production of HMVs avoids the complicated vaccine manufacture in vitro and the strict storage conditions required for these non-in-situ vaccines. Further, the two drugs (DOX and TKIs) we used in the HDDT NBs have already been widely applied in clinic for many years and the Den and HA have been proved to have satisfactory biocompatibility, which facilitate the potential clinical applications of the HDDT NBs. Collectively, the HMV production strategy provides a simple and promising nanotechnology-based strategy for in situ generation of autologous tumor-cell vaccines and may shed light on the development of personalized cancer vaccines.

## Methods
### Materials
DOX•HCl was ordered from Beijing Huafeng United Technology Co., Ltd. (China). Sodium dodecyl sulfate (SDS), LPS, Triton X-100, and glutaraldehyde (50%, mass fraction) were ordered from Sigma-Aldrich (Shanghai, China). HA (110 kDa) was purchased from Shanghai Macklin Biochemical Co., Ltd. (Shanghai, China). Apa, Das, Sor, trimethylamine, CPZ, bovine serum albumin (BSA), genistein, amiloride, and MβCD

were obtained from Aladdin (Shanghai, China). Lap was ordered from Energy Chemical (Shanghai, China). The ethylenediamine core amine-terminated G4 polyamidoamine (PAMAM) dendrimer (Den) (polydispersity index < 1.08) was bought from Dendritech (Midland, USA). Collagenase IV was obtained from BioFroxx (Guangzhou, China). Deoxyribonuclease I was ordered from Leagene Biotechnology Co., Ltd. (Beijing, China). Methylthiazolyldiphenyl-tetrazolium bromide (MTT) was bought from Shanghai Yuanye Bio-Technology Co., Ltd. Dimethyl sulfoxide (DMSO) and methanol were obtained from Shanghai Lingfeng Chemical Reagent Co., Ltd. IL-4 and GM-CSF were bought from Pepro Tech Inc. (Rocky Hill, USA). DiD and DiR were obtained from KeyGEN BioTECH Co., Ltd. (Nanjing, China). Hoechst 33342 was purchased from Beyotime Institute Biotechnology (Shanghai, China). FITC-labeled goat anti-rabbit immunoglobulin G (IgG) antibody (catalog number (cat. no.) bs-0296G-FITC), anti-mouse CD44-FITC antibody (cat. no. bs-0521R-FITC), and enhanced chemiluminescence (ECL) reagent were obtained from Bioss Antibodies (Beijing, China). The horseradish peroxidase (HRP)-labeled goat anti-rabbit IgG (H + L) antibody (cat. no. E-AB-1003) was bought from Elabscience Biotechnology Co., Ltd. (Wuhan, China). Anti-mouse CD44 (cat. no. mAb #3570) and anti-mouse caspase-3 (cat. no. mAb #9668) were bought from Cell Signaling Technology (Beverly, MA, USA). Anti-mouse glyceraldehyde-3-phosphate dehydrogenase (GAPDH) antibody (cat. no. ab8245) was ordered from Abcam (Cambridge, UK). Anti-mouse cleaved caspase-3 (cat. no. Asp175) was obtained from Affinity Biosciences (Jiangsu, China). PD-1 monoclonal antibody (cat. no. BP1046) was obtained from BioXcell (New Hampshire, USA). Anti-mouse CRT (cat. no. 27298-1-ap) was purchased from Proteintech (Wuhan, China). Anti-mouse CD4 (cat. no. GB13064-2), anti-mouse CD8 (cat. no. GB13429), anti-mouse granzyme B (cat. no. GB14092), anti-mouse Gr-1 (cat. no. GB11229), anti-mouse FoxP3 (cat. no. GB11093), anti-mouse CD206 (cat. no. GB13438), and cyanine3 (Cy3)-labeled goat anti-rabbit IgG antibody (cat. no. GB21303) were obtained from Wuhan Servicebio Technology Co., Ltd. (China). Anti-mouse CD3-PE (cat. no. 12-0032-82), anti-mouse CD4-FITC (cat. no. 11-0041-82), anti-mouse CD8a (CD8)-APC (cat. no. 17-0081-81), anti-mouse CD8-PE-Cy7 (cat. no. 25-0081-81), anti-mouse MCH II-PE (cat. no. 12-5321-82), anti-mouse CD11c-FITC (cat. no. 11-0114-82), anti-mouse CD80-PE (cat. no. 12-0801-82), anti-mouse CD86-PE-Cy7 (cat. no. 25-0862-82), and TRIzol reagent were bought from Invitrogen (Carlsbad, USA). Anti-mouse CD11c-PE (cat. no. 117308) and anti-mouse F4/80-FITC (cat. no.124611) were purchased from BioLegend (San Diego, USA). The HMGB1 ELISA kit was bought from Cusabio Biotech Co., Ltd. (Wuan, China). Mouse cytokine ELISA kits for IL-1β (cat. no. 88-7013-22), IL-6 (cat. no. 88-7064-22), and TNF-α (cat. no. 88-7324-22) were purchased from Invitrogen (Carlsbad, USA), and IL-10 (cat. no. EK210/4-96) and IFN-γ (cat. no. EK280/3-48) were bought from MultiSciences (Lianke) Biotech, Co., Ltd. (Hangzhou, China). Deionized water (18.2 MΩ•cm) was obtained from a Milli-Q system (Millipore, Billerica, MA). Dialysis membranes with a molecular weight cut-off (MWCO) of 250 kDa were purchased from Spectrum Labs (Rancho Dominguez, USA). Ultrafiltration tubes (MWCO = 100 kDa) were obtained from Merck Millipore Ltd. (Shanghai, China). Unless otherwise mentioned, all the antibodies are diluted 200 times before use.

### Preparation of DDA NPs
Den was dissolved in methanol with a concentration of 1 mM. DOX•HCl was dissolved in methanol and was neutralized with triethylamine to obtain the HCl-free DOX solution (10 mM). Apa was dissolved in methanol with a concentration of 5.8 mg/mL. Then 50 μL of DOX solution, 25 μL of Apa solution, and 150 μL of Den solution were mixed and the suspension was vortexed for 20 s, followed by evaporation under nitrogen gas. The obtained powder was resuspended with 1 mL of deionized water under bath sonication for 2 min, and the resultant suspension was centrifuged (6700 g, 10 min) to remove the

precipitates. The DDA NPs were in the supernatant and the concentrations of DOX and Apa in the NPs were quantified by UV−vis spectroscopy.

### Preparation of HDDA NBs

For HA coating, 1 mL of the above-obtained DDA suspension (with the DOX concentration of 0.5 mM) was added to 1 mL of HA solution (1 mg/mL) dropwise under agitation and the mixture was further vortexed for 20 s to form the HDDA NBs. The resultant HDDA suspension was dialyzed (MWCO = 250 kDa) to remove free HA molecules. For comparison purpose, the HDD and HDA NPs were prepared in a similar way.

### Preparation of HDDL, HDDS, and HDDD NBs

For the fabrication of HDDL, Den was dissolved in $H_2O$ with a concentration of 1 mM. DOX•HCl was dissolved in DMSO and was neutralized with triethylamine to obtain the HCl-free DOX solution (20 mM). Lap was dissolved in DMSO with a concentration of 5.8 mg/mL. Then, 150 µL of Den solution was added into 800 µL $H_2O$. Next, 25 µL of DOX solution and 25 µL of Lap solution were mixed and vortexed for 20 s, and the resultant mixture was added to the above Den solution dropwise under stirring to obtain the Den−DOX−Lap NPs. The resultant Den−DOX−Lap suspension was centrifuged (6700 g for 10 min) to remove the precipitates. For HA coating, 1 mL of the above obtained Den−DOX−Lap suspension (with the DOX concentration of 0.5 mM) was added to 1 mL of HA solution (1 mg/mL) dropwise under agitation and the mixture was further vortexed for 20 s to form the HDDL NBs. The resultant HDDL suspension was dialyzed (MWCO = 250 kDa) to remove the free HA molecules. The HDDS and HDDD NBs were obtained in a similar way.

### Preparation of HMVs and the "Soni" sample

To prepare and isolate HMVs, 4T1 cells were seeded into 6-well plates and incubated for 24 h. Next, the cells were treated with HDDA NBs (DOX: 14.5 µg/mL) for 24 h. The suspensions were collected, washed with PBS, and purified via ultrafiltration (MWCO = 100 kDa) for at least 3 times to obtain the PBS suspension of HMVs. To prepare the "Soni" sample, $1 \times 10^7$ 4T1 cells were collected via centrifugation (-100 g, 5 min), and were resuspended in PBS to obtain the 4T1 cell suspension. Then, the cell suspension was treated with bath sonication for 10 min and washed with PBS via ultrafiltration (MW = 100 kDa) for at least 3 times to obtain the PBS suspension of the "Soni" sample. The protein contents of HMVs and the "Soni" sample were determined by a BCA protein assay kit (Beyotime, China).

### Characterization

The encapsulation ratio (ER) and loading coefficient (DL) of Apa or DOX in HDDA NBs were calculated via the equations shown below (1 and 2):

$$ER(\%) = \frac{\text{Weight of Apa/DOX loaded in HDDA/}}{\text{Weight of Apa/DOX fed initially}} \times 100\% \quad (1)$$

$$DL(\%) = \text{Weight of Apa/DOX loaded in HDDA/Weight of HDDA} \times 100\% \quad (2)$$

in which the weight of Apa or DOX loaded in HDDA was determined by UV−vis spectroscopy.

The morphology and size of HDDA and DDA were studied using a transmission electron microscope (JEM2100, JEOL Ltd., Japan). The zeta potentials and hydrodynamic sizes of HDDA and DDA were characterized by a zetasizer instrument (Nano ZS, Malvern Instruments, UK). The UV−vis absorption spectra of DOX, Apa, HA, Den, and HDDA solutions/suspensions were measured by a Shimadzu UV-2600 spectrophotometer (Japan).

### Cell culture and animal model

4T1 (cat. no. KG338), B16F10 (cat. no. KG078), MCF-7 (cat. no. KG031), MCF-7/ADR (cat. no. KG0311), NIH 3T3 (cat. no. KG085), and RAW 264.7 (cat. no. KG240) cells were obtained from KeyGEN BioTECH, China. HMEC-1 (cat. no. bio-74149), HPAEpiC (cat. no. bio-73467), and MDA-MB-231 (cat. no. bio-69126) cells were purchased from American Type Culture Collection (ATCC), USA. MCF-7, 4T1, RAW 264.7, and MCF-7/ADR cells were cultured in Roswell Park Memorial Institute (RPMI) 1640 (Gibco, USA) containing 10% fetal bovine serum (FBS), 0.08 mg/mL streptomycin, and 80 U/mL penicillin at 37 °C in 5% $CO_2$. B16F10, HPAEpiC, HMEC-1, MDA-MB-231, and NIH 3T3 cells were incubated in Dulbecco's modified Eagle's medium (DMEM) (Gibco, USA) containing 10% FBS, 0.08 mg/mL streptomycin, and 80 U/mL penicillin at 37 °C in 5% $CO_2$.

BALB/c and C57BL/6 mice (female, 6−8 weeks) were ordered from Yangzhou University Medical Center (Yangzhou, China). Mice were housed in groups of 5 mice per individually ventilated cage with constant room temperature ($25 \pm 3$ °C) with 12 h dark−light cycles and a relative humidity of 40−70%. All mice had access to food and water ad libitum. The 4T1 xenograft tumor model was built by inoculating $8 \times 10^6$ 4T1 cells into the back of BALB/c mice. The B16F10 xenograft tumor model was built by inoculating $8 \times 10^6$ B16F10 cells into the back of C57BL/6 mice. For the maximal tumor size/burden, we adhered to the guideline of the maximal tumor diameter of 20 mm according to the national standard (i.e., the Assessment Guidelines for Humane Endpoints in laboratory animal (RB/T 173-2018)), and the maximal tumor size/burden in this study was not exceeded.

### Endocytic pathway investigation

To investigate the endocytic pathways, different endocytosis inhibitors such as amiloride (inhibitor of macropinocytosis-dependent endocytosis, 10 µg/mL), CPZ (inhibitor of clathrin-mediated endocytosis, 5 µg/mL), genistein (inhibitor of caveolae-mediated endocytosis, 50 µg/mL), and MβCD (inhibitor of lipid raft-dependent endocytosis, 5 mg/mL) were separately added to 4T1 cells and incubated for 2 h at 37 °C in serum-free culture medium. After the pretreatment, the culture medium was carefully replaced with HDDA (DOX: 3.6 µg/mL) in complete RPMI 1640 containing different inhibitors (except MβCD) with further incubation for 2 h. To study the energy-dependent endocytosis, 4T1 cells were also incubated at 4 °C for 2 h. After that, the culture medium was carefully replaced with HDDA (DOX: 3.6 µg/mL)-containing culture medium and the cells were further incubated at 4 °C for 2 h. Then, the treated cells were collected and analyzed by a flow cytometer (NovoCyte 2070 R, ACEA Biosciences Inc., USA) or a confocal microscope (TCS SP8, Leica, Germany).

### HMV protein analysis by SDS-PAGE

To study the composition of the 4T1 cell-produced HMV proteins, the cytoproteins, cytoplasmic proteins, cell membrane proteins, and HMV proteins of 4T1 cells were collected using a membrane and cytosol protein extraction kit (Beyotime, China) and the SDS-PAGE assay was performed following a standard protocol.

### Cell viability assay

4T1 cells were seeded in 96-well plates at a density of $5 \times 10^3$ cells/well. 24 h later, the cell culture medium was replaced by fresh culture medium containing various concentrations of DOX, Apa, DOX + Apa, HDD, HDA, or HDDA. After another 24 h, MTT assay was performed to determine the relative cell viabilities. The viability of untreated cells (control) was set as 100%.

### RTCA experiments

The real-time cytotoxicities of culture medium (control), DOX, Apa, DOX + Apa, HDA, HDD, and HDDA (DOX: 3.6 µg/mL, Apa: 1.8 µg/mL) were studied by a real-time cell analyzer−the iCELLigence system

(ACEA Biosciences, Inc.). Briefly, 4T1 cells were seeded into 8-well L8 E-plates ($1.5 \times 10^4$ cells/well) with 450 µL culture medium in each well. Next, different drugs were added separately by replacing the initial culture media with the drug-containing media when the cell index (CI) value reached 2.8–4.5. The real-time proliferation processes of the cells were recorded based on an impedance-based system and were presented as cell growth curves by the software of the instrument.

### Intracellular ROS detection
The production of ROS was determined by the ROS probe DCFH-DA (KeyGEN BioTECH, China). Specifically, 4T1 cells after different treatments were collected and treated with DCFH-DA (10 µM) for 30 min (37 °C, 5% $CO_2$). Then all the cells were measured by the flow cytometer.

### Apoptosis/necrosis assay
To study the cellular apoptosis/necrosis induced by culture medium (control), DOX, Apa, DOX + Apa, HDA, HDD, and HDDA (DOX: 3.6 µg/mL, Apa: 1.8 µg/mL), we carried out the annexin V-FITC/propidium iodide (PI) apoptosis detection assay. In brief, the 4T1 cells after respective treatments were collected, washed three times with PBS, treated with an apoptosis detection kit (KeyGEN BioTECH, China), and then measured by the flow cytometer.

### Confocal imaging analyses on the internalization by 4T1 cells and endo-lysosomal escape of HDDA NBs
For confocal imaging, 4T1 cells were seeded in Lab-Tek 8-well chamber slides ($1 \times 10^4$ cells/well) and incubated for 1 d. Then, the cells were incubated with various drugs. Confocal images were taken by the confocal microscope.

To study the cellular internalization of DOX and HDDA, 4T1 cells were treated with DOX or HDDA (DOX: 1.8 µg/mL) for various time periods. Next, the media were replaced by the Hoechst working solution (10 µg/mL) and further incubated for 10 min. Then, the cells were washed with PBS and observed by the confocal microscope at the excitation wavelengths of 405 (for Hoechst) and 552 (for DOX) nm, respectively.

To study the endo-lysosomal escape of HDDA NBs, 4T1 cells were incubated with HDDA NBs (DOX: 3.6 µg/mL) for different time periods. Next, the culture media were replaced by the media containing Hoechst (10 µg/mL) and LysoTracker Green (0.5 µg/mL, purchased from KeyGEN BioTECH, China) and further incubated for 15 min. Finally, the 4T1 cells were washed with PBS and observed by the confocal microscope at the excitation wavelengths of 405 (for Hoechst), 488 (for LysoTracker Green), and 552 (for DOX) nm, respectively.

### Comparative analysis on the cellular internalization of HDDA NBs by 4T1 and MDA-MB-231 cells
To investigate the cellular internalization of HDDA NBs by 4T1 and MDA-MB-231 cells, the two types of cells were treated with HDDA (DOX: 3.6 µg/mL) for 3 h. Then, the treated cells were collected and analyzed by the flow cytometer.

### Western blot analysis
To perform Western blot analysis, the cytoproteins of 4T1 cells, cell membranes (4T1), cytoplasm (4T1), and HDDA-induced HMVs (obtained from 4T1 cells) were collected as mentioned before. The Western blot analyses of CD44, caspase-3, and cleaved caspase-3 were performed following standard protocols. GAPDH was used as the loading control.

### Detection of ICD biomarkers
The CRT expression on the cell membrane was studied via immunofluorescence assay and flow cytometry. For immunofluorescence assay, 4T1 cells were seeded into Lab-Tek 8-well chamber slides at a density of $1 \times 10^4$ cells per well and cultured for 1 d. After that, culture medium (control), DOX, Apa, DOX + Apa, HDA, HDD, and HDDA (DOX: 3.6 µg/mL, Apa: 1.8 µg/mL) were added to the cells separately and incubated for 4 h. After being washed with PBS, the cells were fixed in 4% glutaraldehyde solution for 10 min. Next, the cells were blocked by 5% BSA for 3 h (25 °C), treated with the anti-CRT antibody (1/400) overnight (4 °C), washed with PBS, and then treated with the FITC-labeled goat anti-rabbit IgG antibody (1/300) for 3 h (25 °C). After being stained with Hoechst, the cells were observed under the confocal microscope. The sample treatment procedures of the flow cytometry analysis were similar to those of confocal microscopic imaging.

The contents of the HMGB1 and ATP released by the 4T1 cells after different treatments were analyzed by the HMGB1 ELISA kit and enhanced ATP assay kit (Beyotime, China) following the manufacturers' protocols.

### Analyses of DC maturation and macrophage polarization in vitro
BMDCs were collected from femurs and tibias of BALB/c mice (female, 6–8 weeks of age). Briefly, the mouse bone marrow cells were collected and incubated in the X-VIVO 15 culture medium (Lonza, Switzerland) with IL-4 (10 ng/mL) and GM-CSF (20 ng/mL) for 5 d to obtain the immature BMDCs. Besides, 4T1 cells were seeded into the upper chambers of the transwells (24-well insert; pore size, 8 µm; Corning, USA) and incubated for 1 d. Next, the cells were incubated separately with culture medium, DOX, Apa, DOX + Apa, HDD, HDA, and HDDA (DOX: 7.2 µg/mL, Apa: 3.6 µg/mL) for 4 h. Next, the culture medium was replaced by fresh culture medium. After that, the immature BMDCs were seeded into the lower chambers of the above transwells. After 1 d, BMDCs were collected and stained with anti-mouse CD11c-FITC, anti-mouse CD86-PE-Cy7, and anti-mouse MHC II-PE, and analyzed by flow cytometry. The immature BMDCs treated with LPS (25 ng/mL) and GM-CSF (20 ng/mL) for 1 d were set as positive control and negative control, respectively.

For macrophage polarization experiment, the procedures were similar to those in the above experiment, and M2-like RAW 264.7 cells (pretreated with IL-4) were used to replace the above BMDCs. The RAW 264.7 cells treated with different samples were stained with anti-mouse F4/80-FITC and anti-mouse CD11c-PE antibodies, and then analyzed by flow cytometry. The RAW 264.7 cells treated with culture medium, IL-4 (20 ng/mL), and IFN-γ (100 ng/mL) + LPS (25 ng/mL) for 1 d were set as the M0, M2, and M1 types of macrophages, respectively.

### Hemolysis assay
The blood sample (1 mL) was collected from healthy BALC/c mice. RBCs were obtained by the centrifugation (600 g, 5 min) of the above blood sample and resuspended by PBS (50 mL) to obtain an RBC suspension. Next, various concentrations of HDDA NB suspensions (DOX: 0, 1.8, 3.6, 7.2, 14.4, and 29.0 µg/mL; 0.5 mL; PBS as the solvent) were mixed with the RBC suspensions (0.5 mL), respectively. After treatment for 2 h, the supernatants were collected via centrifugation (600 g, 10 min). RBCs in water and PBS were used as positive control and negative control, respectively.

### In vivo antitumor experiments
The 4T1 tumor-bearing BALB/c mice were divided into five groups (n = 9/group) for different treatments as follows for three times (with an interval between each injection of 1 day): (1) i.v. injected with PBS (control), (2) i.v. injected with DOX solution, (3) i.v. injected with HDA suspension, (4) i.v. injected with HDD suspension, and (5) i.v. injected with HDDA suspension. Note that the groups 2–5 had the same DOX dose of 5 mg/kg and the same injection volume (100 µL). The tumor size and body weight were recorded for 38 and 30 d, respectively. Tumor volume (V) was calculated as width$^2$ × length/2.

The B16F10 tumor-bearing C57BL/6 mice were divided into 5 groups ($n = 10$/group) and i.v. injected with PBS (control), DOX, HDA, HDD, and HDDA for 3 times (DOX: 5 mg/kg), respectively. On the next day of each injection, half of the mice in each group were i.p. injected with the anti-PD-1 antibody (10 mg/kg). The tumor size and body weight were recorded for 30 and 22 d, respectively.

## Tumor immunofluorescence analysis

Briefly, the 4T1 tumor-bearing BALB/c mice or the B16F10 tumor-bearing C57BL/6 mice were sacrificed after various treatments, and their tumors were excised for the immunofluorescence analyses of CRT, Gr-1, CD206, FoxP3, and granzyme B following standard protocols.

For TUNEL assay, a TUNEL apoptosis in situ detection kit (KeyGEN BioTECH, China) was used following the instructions from the manufacturer.

## Biological TEM (Bio-TEM) measurements

Briefly, the 4T1 tumor-bearing BALB/c mice were sacrificed after HDDA or PBS (control) treatment, and their tumors were excised for Bio-TEM measurements. First, the tumor tissues were fixed with 4% glutaraldehyde at 4 °C overnight, followed by immobilization in $OsO_4$ solution (1%) at room temperature for 1 h. Next, the tumor tissues were dehydrated using ethanol with graded concentrations and then embedded in epoxy resin. An ultramicrotome (Leica Microanalysis) was used to obtain ultrathin tissue sections. Then, the obtained sections were stained with uranyl acetate/lead citrate and imaged under a transmission electron microscope (H-600(4), Hitachi, Japan).

## Transcriptomic analysis, proteomic analysis, and cytokine profile analysis

For total cellular transcriptomic analysis, 4T1 cells were incubated with culture medium (control) or HDDA NBs (DOX: 3.6 μg/mL) for 1 d, and the RNA samples were extracted following the TRIzol-based procedure. For tumor transcriptomic analysis, the 4T1 tumor-bearing BALB/c mice were sacrificed 7 d after PBS (control) or HDDA NB treatment (DOX: 5 mg/kg), and the RNA samples of the tumors were collected following the TRIzol-based procedure. Total cellular RNA and tumor RNA were analyzed by Novogene (Beijing, China).

Proteomic analysis was performed using mass spectrometry in data-dependent acquisition mode. The proteins of 4T1 cells, "Soni" sample, and HMVs were analyzed by Shanghai Applied Protein Technology Co., Ltd. (Shanghai, China).

The ELISA analyses of the extracellular levels of TNF-α, IL-1β, and TGF-β produced by RAW 264.7 cells that were treated with the culture media of differently treated 4T1 cells, and the intratumoral and serum levels of IL-6, IL-1β, IFN-γ, TNF-α, and IL-10 were performed following the instructions of the ELISA kits.

## Flow cytometry analyses of tumor tissues, spleens, and TDLNs

The tumor tissues, spleens, and TDLNs of 4T1 tumor-bearing BALB/c mice and B16F10 tumor-bearing C57BL/6 mice after different treatments were retrieved for flow cytometry analyses. The tumor tissues were cultured in the dissociation buffer (100 μg/mL deoxyribonuclease I and 1 mg/mL collagenase IV) at 37 °C for 30 min. The spleens and TDLNs were soaked in PBS and passed through a cell strainer (70 μm) to obtain the single-cell suspensions. The single-cell suspensions were washed with 1 wt% BSA buffer (solvent: PBS), and then stained with anti-CD11c-FITC, anti-CD80-PE (or anti-MHC II-PE), anti-CD86-PE-Cy7, anti-CD11c-PE, anti-F4/80-FITC, anti-CD3-PE, anti-CD4-FITC, and/or anti-CD8-PE-Cy7 following the instructions of the manufacturers to detect the maturation of DCs (CD11c$^+$CD86$^+$MHC II$^+$ or CD11c$^+$CD86$^+$CD80$^+$), the M1-like TAMs (F4/80$^+$CD11c$^+$), the helper T cells (CD3$^+$CD4$^+$), and the cytotoxic T cells (CD3$^+$CD8$^+$), respectively. All the antibodies were used at a dilution of 1:200.

The antibodies-stained single-cell suspensions were analyzed by flow cytometry.

## Histological staining and blood analysis

The BALB/c mice were i.v. injected with PBS (control), DOX, HDA, HDD, or HDDA. The C57BL/6 mice were i.v. injected with PBS (control), DOX, HDA, HDD, or HDDA and i.p. injected with the anti-PD-1 antibody (10 mg/kg). 7 d after administration, the major organs of the treated mice were collected for H&E assay following a standard protocol. Besides, the routine blood analyses were carried out by an automatic hematology analyzer (BC-2800Vet, Mindray, China), and the biochemical tests were performed on an automated biochemical analyzer (SMT-100V, Seamaty, China). For the PBS- and HDDA-treated BALB/c mice, their major organs were also collected for Masson, Van Gieson (VG), hexamine silver, and TUNEL staining assays following standard protocols, respectively.

## In vivo tumor prevention experiments

To study the tumor prevention effect of HMVs, BALB/c mice were divided into 3 groups ($n = 4$/group) for different treatments as follows for three times (with an interval between each injection of 3 days): (1) subcutaneously (s.c.) injected with PBS; (2) s.c. injected with the "Soni" sample (protein content: 1 mg per mouse); (3) s.c. injected with the HMV suspension (protein content: 1 mg per mouse). Next, the mice were s.c. injected with 4T1 cells ($8 \times 10^6$ cells per mouse) to build the subcutaneous 4T1 tumor models. The tumor size and body weight were recorded for 30 and 28 d, respectively. To study the immune memory effect of HDDA, the HDDA-healed BALB/c mice were s.c. or i.v. rechallenged by 4T1 tumor cells (s.c. injected with $8 \times 10^5$ cells per mouse or i.v. injected with $8 \times 10^4$ cells per mouse), and the tumor size or the number of tumor nodules were recorded. The untreated mice were set as the control group. To study the immune memory effect of "HDDA + anti-PD-1", the "HDDA + anti-PD-1"-healed C57BL6 mice were i.v. rechallenged by B16F10 tumor cells ($8 \times 10^4$ cells per mouse), and the number of tumor nodules was recorded. The untreated mice were set as the control group.

## Statistics and reproducibility

Most of the numeric data are expressed as mean ± SD unless otherwise indicated. The significance between two groups was analyzed by two-tailed Student's $t$-test. For multiple comparisons, one-way analysis of variance (ANOVA) with Tukey's post-hoc test was adopted. $P$ values of less than 0.05 were considered significant. $*P < 0.05$, $**P < 0.01$, $***P < 0.001$, $****P < 0.0001$. All statistical analyses were performed by GraphPad Prism 9, Excel 2016, or Excel 2019. The flow cytometry data were processed using NovoExpress (version 1.5.0., Agilent).

## Reporting summary

Further information on research design is available in the Nature Research Reporting Summary linked to this article.

# Data availability

The data supporting the results in this study are available within the Article, Supplementary Information or Source Data file. All the raw data used to make the figures in this study are available in the Source data file and also from figshare with the identifier: https://doi.org/10.6084/m9.figshare.20267598.v2. The sequencing data of the transcriptomic analyses in this study are available from the Sequence Read Archive (SRA) Run Selector of the National Center Biotechnology Information (NCBI) database with the NCBI BioProject accession number PRJNA798199 for the transcriptomic analysis of 4T1 tumors and with the NCBI BioProject accession number PRJNA798200 for the transcriptomic analysis of 4T1 cells. The mass spectrometry-based proteomics data have been deposited to the ProteomeXchange

Consortium via the iProX partner repository with the dataset identifier PXD035315. Source data are provided with this paper.

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

## Acknowledgements

This work was supported by the Natural Science Foundation of Jiangsu Province (BK20211510), National Natural Science Foundation of China (21673037), and the Fundamental Research Funds for the Central Universities.

## Author contributions

Y.G. and F.G.W. conceived the research and designed the experiments. Y.G., S.Z.W., Xinping Z., S.Y.W., and Y.L. performed the experiments. Y.G., H.R.J., Y.X.Z., Xiaodong Z., G.G., Y.W.J., C.L., X.C., and F.G.W. analyzed the data. Y.G. and F.G.W. wrote the manuscript. F.G.W. supervised the project. All authors read and approved the manuscript.

## Competing interests

The authors declare no competing interests.

## Ethics approval

All animal experiments were performed in accordance with the permission from the ethics committee of Southeast University, China (No. 20211224002). All experiments were performed in compliance with the Regulations for the Administration of Affairs Concerning Experimental Animals of China. All experiments followed institutional guidelines.
