## [Peer Review File · Nature Communications]

In situ generation of micrometer-sized tumor cell-derived vesicles as autologous cancer vaccines for boosting systemic immune responsesREVIEWER COMMENTS

Reviewer #1 (Remarks to the Author): with expertise in dendrimers, nanoparticles, drug delivery

In this manuscript, the author prepared doxorubicin (DOX)- and tyrosine kinase inhibitor (TKI)-loaded and hyaluronic acid (HA)-coated dendritic polymers as the nanobombs (NBs) to induce robust cancer cell death. The synthesized NBs could in situ produce massive tumor cell-derived vesicles (termed HMVs) as autologous cancer vaccines for boosting systemic immune responses. The size and morphology of the NBs were analyzed. The cellular internalization and subcellular distribution of the NBs were also analyzed. And the ability of the NBs to convert cancer cells into HMVs was confirmed in vitro. Lastly, the NBs have been demonstrated to restrain the tumor growth, induce immunogenic cell death and produce HMVs in vivo, translate the tumor tissues to the antigen depot, evoke immune response, and establish immunological memory effects. This paper could be accepted for publication in Nature Communications after the authors have addressed the following minor concerns:

1. First of all, the mechanism of Apa (TKI) to inhibit the growth of cancer cells should be supplemented. Secondly, why the free Apa could induce the generation of ROS (Figure 2c)?
2. The colloidal stability of the prepared NBs should be assessed.
3. Supplementary Fig. 12b, "cytoplasm" and "cell membrane" were inverted on the x-coordinate.
4. Page 19, Line 14, the author wrote as "the DOX-induced apoptosis could not change the immunosuppressive TME". However, according to the results of in vivo antitumor assays, the HDD group that only contained DOX could reverse the immunosuppressive TME, for example, reduce the number of MDCs and Tregs (Fig. 4j). Please modify this sentence according to the experimental results.
5. Page 26, Line 5 from the bottom, the author should explain why the HMVs could target the lymph nodes after subcutaneous injection.
6. The full name of the abbreviation should only be given when it was first appeared. For example, the full names of Apa and PD-1 appeared multiple times in the manuscript.

Reviewer #2 (Remarks to the Author): with expertise in cancer immunology

The manuscript entitled "In situ massive generation of micrometer-sized tumor cell-derived vesicles as autologous cancer vaccines for boosting systemic immune responses" by Guo et al. explores the role of tumor cell-derived vesicles as autologous cancer vaccines. The authors performed extensive experiments that in general support the anti-tumor effect of hyaluronic acid (HA)-coated dendritic polymer (HDDT) nanobombs, but fail to convincingly support the role (or even existence?) of tumor cell-derived vesicles in the process.

In more detail, the authors showed:

- (i) effect of HDDT nanobombs in target cell disintegration and cell death on cell culture models, that the authors identify as an apoptotic process based on PI/annexin detection and transcriptome analysis of target cells
- (ii) authors performed basic experiments to understand if the observed apoptosis is an immunogenic process (immunogenic cell death), e.g. they show that DAMPS (calreticulin, HMGB1, ATP) are exposed or secreted from cells and that something in the "secretome" of HDDT-treated cells induces maturation of dendritic cells and repolarization into M1-macrophages (transwell experiments), but failed to further explore the mechanism of this process although it is central to the role as cancer vaccines (e.g. knockdown, blockage experiments etc; Fucikova et al, Cell Death Dis, 2020).
- (iii) durable anti-tumor effect (apoptosis in tissue, decreased tumor volume and prolonged survival) of HDDT nanobombs in 4T1 tumor-bearing BALB/c mice
- (iv) HDDA-treated tumor tissue showed signs of immunostimulation, e.g. increased calreticulin-staining, release of pro-inflammatory cytokines, increased proportion of M1-macrophages, mature dendritic cells, helper T cells and cytotoxic T cells, increase in granzyme B; enrichment in transcripts associated to inflammatory signaling pathways

- (v) HDDA nanobomb treatment had effect on remote immune tissues like tumor-draining lymph nodes and spleen: e.g. enrichment of matured dendritic cells, increased proportion of the M1-macrophages, increased CD4+ and CD8+ T cells; and increased the level of proinflammatory cytokines in serum
- (vi) HDDA nanobomb showed certain level of biocompatibility, e.g. HDDA showed low level of hemolysis in vitro, no major changes were observed in routine blood or histological tissue analysis in vivo and no apparent weight loss, but HDDAs targeting to other tissues (Suppl. 19, 20) and off target effect was not examined in detail
- (vii) in vivo anti-tumor effects of HDDA nanobomb were additionally shown in the highly aggressive melanoma tumor mouse model (B16F10)

Importantly, the proposed role of HDDA-treated tumor cell-derived vesicles is not supported by the experiments. The authors did not characterize the vesicles released from HDDA-treated cultured cells, as proposed in Minimal information for studies of extracellular vesicles 2018 (published in the societies' journal JEV in 2018). Therefore, it is not really clear what these vesicles represent and how they separate in characteristics from cells/cell debris and importantly, if they contain HDDA nanobombs or not.

The only experiments performed on purified vesicles (the isolation protocol itself is not very clear) are determination of protein concentration and protein profile on SDS-PAGE gel, and the authors claim lower annexinV and caspase 3 presence in vesicles, while detecting CD44 in vesicles, but I could not access those figures. Although CD44 might be present on vesicles, this is not a specific marker and might not be the best way to detect vesicles in tumor tissues. Later, the authors isolated vesicles released from HDDA-treated cells and labelled them with DiD. Again, they provided no characterization of these labelled vesicles (do they contain HDDA nanobombs?), and directly used them in BALB/c and C57BL/6 mouse models to show immune-modulatory role, but failed to include any controls of DiD-vesicles.

The authors also failed to report on previous studies on apoptotic vesicles in general and specifically in cancer in the Introduction (reviewed in Gregory et al, Front Immunol. 2018; Muhsin-Sharafaldine et al, Immunol Cell Biol, 2018), so it is not clear how familiar are they with the subject.

Reviewer #3 (Remarks to the Author): with expertise in nanoparticles, drug delivery

In this manuscript, authors reported a strategy based on the HDDT NBs to induce cancer cell death and realize the in situ massive production of HMVs as cancer vaccines. The work was systematically investigated. The results obtained seem to be interesting. I think it can be accepted for publication after some revisions.

Comments:

- (1) Authors declared that "The encapsulation ratio (ER) of both Apa and DOX in HDDA NBs was almost 100%". If the Apa and DOX were co-encapsulated into Den simply via hydrophobic interaction, the drugs were not completely encapsulated. Please give the preparation process and the calculation method for ER.
- (2) How were the cancer cells changed to HMVs? It is not clearly described in text. In Fig.1f, there were almost no micrometer-sized vesicles in the confocal microscopic images of the HDDA-treated 4T1 cells at 1 h, however, they could be found in Fig.1g,k. The cell membrane, the nuclei and lysosomes can be labelled with fluorescent probe, and the HDDA-treated cancer cells can be observed by confocal images to visualize the cancer cells change.
- (3) Could the HDDA induce normal cell death? Please provide the data.

Reviewer #4 (Remarks to the Author): with expertise in cancer immunology

The authors provide here evidence for the use of naobombs to efficiently convert tumor cells into

micrometer sized vesicles, that act as vaccines to induce strong anti-tumor responses. The concept is very interesting based on its effectiveness validate in vitro and in vivo, and the simplicity of the preparation. This could be highly helpful in the context of improved vaccine based cancer immunotherapy.

Yet, I believe that a major revision of the current manuscript is needed to improve the work.

The current manuscript is very long, heavy to read, reporting on an enormous number of results shown in complex Figures. As a consequence, the readers is overwhelmed by many initial in vitro data, while losing the ultimate impact of the findings reported towards the end that attest on the in vivo potential of the NBs. Therefore, the authors should absolutely select the most important results to be displayed in the main figures, shorten and be more concise in the result and conclusion sections, by omitting methodological details and background information.

Also, the current displaying of the panels in the main figures is unusual and disordered. This should be rectified, to facilitate the understanding of the panels.

Main points:

- 1) Introduction: The authors refer to some old references on cancer vaccines (ex Ref 15). Please update with more relevant and recent ones. Also, the authors neglect the success of neoantigen long-synthetic peptide-based vaccinations and mRNA-based vaccines that have given very encouraging results in cancer patients. Please update.
- 2) CD44 molecule: the authors claim that the uptake of the NB occurs via CD44. This is an important claim, that however has not been tested experimentally. Is the presence of the molecule correlated with uptake? How is the uptake affected by knocking out CD44? CD44 being a widely expressed molecules, how do the authors explain the limited toxicity to non-tumor cell types? CD44 staining is only shown on one tumor cell line (Supp Fig 3). What about the other ones used, and additional controls.
- 3) P.9: It is not clear what was the rationale to pursue the studies with HDDA NB. This should be explained.
- 4) Fig 3: the modulation of M1/M2 should extend to other analyses in addition to flow cytometry based markers. Are these cells functionally converted to M1 macrophages, in terms of secreted cytokines and enzyme content?
- 5) Immune cell infiltration in treated tumors in vivo: the immune cell evaluation is insufficient to support the claims of the authors. The authors should quantify the different immune cell populations, instead of providing a single representative IF image of CD206, FOXP3,...staining. Similar flow cytometry based analyses as the ones presented in Supp Figure 25 should be provided for M2 macrophages, MDSCs, Tregs, and naïve/memory T cells in the different in vivo treatment settings. Otherwise, it is difficult to conclude and compare the different settings. For instance, from Fig 4j one would conclude that M2 macrophages are similarly reduced in all treatments. "Fewer" red signal is not a quantification term.
- 6) No indication about sample size is provided in several experiments. In particular, it is not clear on how many samples rely the transcriptomic analyses shown in different figures (ex. Fig 2g). This is highly relevant to define the statistical power of the observations. Also, in Figure 5, how comparable where the results of the 3 mice per group in terms of transcriptomics? How does a PCA comparing treated and not-treated groups show?
- 7) Immunological memory: no staining on T cells are shown attesting on their memory phenotype in the in vivo settings.

Minor:

- 1) Why the authors display flow cytometry data (ex Supp Fig 11, Supp Fig 16 and others) as a.u? Better would be to show either % of positive cells of gMFI.
- 2) No dose dependence studies are shown through the manuscript. For instance, is there in Fig 1h a dose-dependent effect? How was the final dose selected?

3) Fig 4k: when/how were the intratumoral cytokine content analyses performed?

4) Fig 6d: it is unclear how the CD4 and CD8 T cells are represented here. The y-axis labels indicate CD4+ or CD8+ cells in CD3+ cells (%). If this is the case the sum of CD4 and CD8 % should be close to 100 in each treatment setting, what is not the case. What is actually displayed in the graphs?

Responses to Reviewer 1:

General comment: In this manuscript, the author prepared doxorubicin (DOX)- and tyrosine kinase inhibitor (TKI)-loaded and hyaluronic acid (HA)-coated dendritic polymers as the nanobombs (NBs) to induce robust cancer cell death. The synthesized NBs could in situ produce massive tumor cell-derived vesicles (termed HMVs) as autologous cancer vaccines for boosting systemic immune responses. The size and morphology of the NBs were analyzed. The cellular internalization and subcellular distribution of the NBs were also analyzed. And the ability of the NBs to convert cancer cells into HMVs was confirmed in vitro. Lastly, the NBs have been demonstrated to restrain the tumor growth, induce immunogenic cell death and produce HMVs in vivo, translate the tumor tissues to the antigen depot, evoke immune response, and establish immunological memory effects. This paper could be accepted for publication in Nature Communications after the authors have addressed the following minor concerns:

Author reply: We deeply appreciate the respected reviewer for his/her recognition of our manuscript! The following very professional suggestions/comments significantly improve the quality of our work! Thank you so much!

Comment 1: First of all, the mechanism of Apa (TKI) to inhibit the growth of cancer cells should be supplemented. Secondly, why the free Apa could induce the generation of ROS (Figure 2c)?

Author reply: This is a very professional comment! Cancer cells express vascular endothelial growth factor (VEGF) and VEGF receptors (VEGFRs), which can interact with each other to prevent cell apoptosis and facilitate sustainable cell growth. (*Oncotarget*, 2016, 7, 17220–17229.) In the tumor tissues, inhibiting tumor cell apoptosis may promote tumor growth. Apa, a VEGFR2 inhibitor, can inhibit the anti-apoptosis function of the VEGF signaling, and thus induce cancer cell death.

On the other hand, cancer cells can produce reactive oxygen species (ROS) at various subcellular locations (e.g., mitochondrion, plasma membrane, and endoplasmic reticulum) and maintain the high intracellular ROS levels due to their high metabolic activity. (*Free Radical Research*, 2010, 44, 479–496.; *Chem. Rev.* 2019, 119, 4881–4985.) To maintain the

intracellular levels of ROS and prevent cells from being destroyed by ROS, various small molecules (e.g., glutathione (GSH), flavenoids, and vitamins A, C, and E) and antioxidant enzymes are expressed by cancer cells to consume intracellular ROS. (*Free Radical Research, 2010, 44, 479–496.*) As we mentioned above, Apa can inhibit the VEGF/VEGFR signaling in cancer cells and induce cell apoptosis. Such a process can disrupt cell metabolism in cancer cells, thus resulting in the accumulation of ROS.

Comment 2: The colloidal stability of the prepared NBs should be assessed.

Author reply: As suggested, we tested the colloidal stability of the HDDA NBs for 7 days by measuring their hydrodynamic diameters. As shown in Fig. R1 below (also in the revised supplementary material as Supplementary Fig. 2), the dynamic light scattering (DLS) results indicated that the hydrodynamic diameter of the HDDA NBs did not have a notable change in the first three days (which are sufficient for a therapeutic use) and became a bit larger (but still at around 120 nm) on day 5. In addition, the polydispersity index (PDI) value remained smaller than 0.3 during the 7 days of observation, indicating the good aqueous dispersity of the HDDA NBs. The result demonstrated that the HDDA NBs had relatively good colloidal stability. Thanks a lot for this very important suggestion!

Fig. R1. Hydrodynamic diameters of HDDA NBs measured by DLS at indicated time points.

Comment 3: Supplementary Fig. 12b, “cytoplasm” and “cell membrane” were inverted on the x-coordinate.

Author reply: We greatly appreciate this reviewer for pointing out this flaw of our manuscript,

and we have corrected it in the corresponding place in the revised manuscript.

Comment 4: Page 19, Line 14, the author wrote as “the DOX-induced apoptosis could not change the immunosuppressive TME”. However, according to the results of in vivo antitumor assays, the HDD group that only contained DOX could reverse the immunosuppressive TME, for example, reduce the number of MDCs and Tregs (Fig. 4j). Please modify this sentence according to the experimental results.

Author reply: This is a very important comment! We completely agree with the reviewer that the HDD could partly reverse the immunosuppressive TME, and the DOX-induced apoptosis plays a major role during this process. The reasons why HDD NPs could partly reverse immunosuppressive TME but free DOX could not are as follows: The HDD NPs can actively target CD44-overexpressed cancer cells and passively target tumor tissues via the enhanced permeability and retention (EPR) effect, while the free DOX molecules cannot. Thus, the unsatisfactory immunostimulation results of the DOX group may be attributed to the poor tumor-targeting and insufficient ICD-inducing ability of free DOX. As suggested, we have changed the sentence to “In contrast, the Gr-1 and Foxp3 signals were only slightly influenced in the DOX group, possibly due to the poor tumor-targeting and insufficient ICD-inducing ability of free DOX.” in the corresponding place in the revised manuscript.

Comment 5: Page 26, Line 5 from the bottom, the author should explain why the HMs could target the lymph nodes after subcutaneous injection.

Author reply: We sincerely appreciate the reviewer for raising this very constructive comment! The lymph node-targeting capacity of cancer vaccines is very important for them to stimulate the immune system in vivo. Many papers in the field of cancer vaccines have reported that different kinds of vaccines can target the lymph nodes for immunostimulation after subcutaneous or intramuscular injection. (*Nat. Biomed. Eng.* 2022, 6, 19–31.; *Nat. Nanotechnol.* 2020, 15, 1043–1052.; *J. Immunol.* 2021, 206, 264–272.) Fortunately, the lymph node-targeting ability of HMs after subcutaneous injection has been proved by the flow cytometric results (Supplementary Fig. 54b,f) in our manuscript. As for the reasons for lymph node targeting, we consulted a recent review (*Nat. Rev. Mater.* 2022, 7, 174–195.), which claimed that after

intramuscular or subcutaneous injection, the activated dendritic cells (DCs) and the vaccines can travel through afferent lymph vessels to the draining lymph nodes. We expect that the HMVs could also activate the DCs at the injection site and travel through afferent lymph vessels with the help of and together with the activated DCs to the draining lymph nodes. Besides, the appropriate particle size of HMVs may also facilitate the lymph node traveling process. The related discussions have been added in the revised manuscript (Page 29). Thanks a lot!

Comment 6: The full name of the abbreviation should only be given when it was first appeared. For example, the full names of Apa and PD-1 appeared multiple times in the manuscript.

Author reply: As suggested, we have carefully checked and deleted the repeated full names in the revised manuscript. Thank you so much for this very helpful comment!

Responses to Reviewer 2:

General comment: The manuscript entitled "In situ massive generation of micrometer-sized tumor cell-derived vesicles as autologous cancer vaccines for boosting systemic immune responses" by Guo et al. explores the role of tumor cell-derived vesicles as autologous cancer vaccines. The authors performed extensive experiments that in general support the anti-tumor effect of hyaluronic acid (HA)-coated dendritic polymer (HDDT) nanobombs, but fail to convincingly support the role (or even existence?) of tumor cell-derived vesicles in the process. In more detail, the authors showed:

Author reply: We sincerely appreciate this respected reviewer's very constructive, professional, and helpful comment! We are extremely grateful for the precious time, invaluable expertise, and superb professionalism the respected reviewer has put in improving the quality of our manuscript!

We also deeply appreciate this reviewer for pointing out some drawbacks of our manuscript, especially for the characterization and antitumor performance of the tumor cell-derived vesicles (we termed HMTVs), and we have tried our best to perform additional experiments to support our conclusions. We sincerely hope that the respected reviewer will be satisfied with our following replies. Thank you so much for your very professional comments!

Comment 1: (i) effect of HDDT nanobombs in target cell disintegration and cell death on cell culture models, that the authors identify as an apoptotic process based on PI/annexin detection and transcriptome analysis of target cells.

Comment 3: (iii) durable anti-tumor effect (apoptosis in tissue, decreased tumor volume and prolonged survival) of HDDT nanobombs in 4T1 tumor-bearing BALB/c mice.

Comment 4: (iv) HDDA-treated tumor tissue showed signs of immunostimulation, e.g. increased calreticulin-staining, release of pro-inflammatory cytokines, increased proportion of M1-macrophages, mature dendritic cells, helper T cells and cytotoxic T cells, increase in granzyme B; enrichment in transcripts associated to inflammatory signaling pathways.

Comment 5: (v) HDDA nanobomb treatment had effect on remote immune tissues like tumor-draining lymph nodes and spleen: e.g. enrichment of matured dendritic cells, increased

proportion of the M1-macrophages, increased CD4⁺ and CD8⁺ T cells; and increased the level of proinflammatory cytokines in serum.

Comment 7: (vii) In vivo anti-tumor effects of HDDA nanobomb were additionally shown in the highly aggressive melanoma tumor mouse model (B16F10).

Author reply: We would like to express our sincere appreciation for these professional comments! We also deeply appreciate the reviewer for his/her recognition of our manuscript!

Comment 2: (ii) authors performed basic experiments to understand if the observed apoptosis is an immunogenic process (immunogenic cell death), e.g. they show that DAMPS (calreticulin, HMGB1, ATP) are exposed or secreted from cells and that something in the “secretome” of HDDT-treated cells induces maturation of dendritic cells and repolarization into M1-macrophages (transwell experiments), but failed to further explore the mechanism of this process although it is central to the role as cancer vaccines (e.g. knockdown, blockage experiments etc; Fucikova et al, Cell Death Dis, 2020).

Author reply: We greatly appreciate this reviewer for raising this very professional comment, and the mentioned article was very helpful and has been cited in the revised manuscript. First, we have proved that the HDDA-induced cell death is an immunogenic process and DAMPs (e.g., calreticulin (CRT), HMGB1, and ATP) could be exposed or secreted from the HDDA-treated tumor cells, and we thank this reviewer for his/her recognition of this part of the experiments.

Next, we carried out transwell experiments to prove that the “secretome” of HDDT-treated cells could induce maturation of dendritic cells (DCs) and M1-like repolarization of macrophages. However, the “secretome” contains not only HMVs, but also many small/macro molecules (e.g., DAMPs and tumor-associated antigens). The purpose of the transwell experiments was to simulate the actual situation in the tumor microenvironment (TME), because the immature DCs and M2-like macrophages in the tumor tissues could interact with not only HMVs but all of these “secretome” after HDDA-induced tumor cell death. On the other hand, we deeply agree with this reviewer that the transwell experiments are not sufficient to prove the immunostimulatory effect of HMVs.

As suggested, to further prove the immunostimulation capacity of HMVs in vitro, we first

studied the CRT (an important DAMP) expression levels on HMVs. The immunofluorescence staining result indicated that the HMVs carried a certain amount of CRT, ensuring their potential immunostimulation effect (Fig. R1). Next, the proteomic analysis of HMVs was also carried out (the detailed results were shown in the response to *Comment 8* below). The results demonstrated that the contents of some immune system-related proteins on HMVs were higher than those of the control samples (4T1 cells) (Fig. R13). These proteins may be the reasons of the strong immunostimulation effect of HMVs.

Fig. R1. Representative flow cytometric plot (a) and corresponding quantification results (b) of CRT protein (stained by anti-CRT primary antibody and FITC-labeled secondary antibody) expression level on the HMVs. The HMVs without antibody staining were set as the blank group.

Further, the *in vitro* immunostimulation effect of HMVs was also studied. The sonication-induced cell debris (termed “Soni” sample) was used for comparison purpose. As shown in the flow cytometric results, the rates of matured DCs ($CD11c^+CD80^+CD86^+$) (64.78%) and M1-like macrophages ($CD11c^+F4/80^+$) (31.25%) were remarkably increased after HMV treatment, and the immunostimulation effect of HMVs was stronger than that of the “Soni” sample (Figs. R2 and R3). Moreover, we also analyzed various cytokines secreted by macrophages after HMV treatments via ELISA assay (Fig. R4). The results further proved the immunostimulation effect of HMVs. The related discussions and the figures have been added in the revised manuscript (Page 17). Thanks a lot!

Fig. R2. Representative flow cytometric plots of matured DCs (CD11c⁺CD80⁺CD86⁺) after incubation with culture medium (control), "Soni" sample, and HMVs, respectively.

Fig. R3. Representative flow cytometric plots of M1-like macrophages (CD11c⁺F4/80⁺) after incubation with culture medium (control), "Soni" sample, and HMVs, respectively.

Fig. R4. Expression levels of TNF- α , IL-1 β , and TGF- β produced by RAW264.7 cells after different treatments, as analyzed by ELISA. The IL-4-incubated RAW264.7 cells (M2-like macrophages) were set as the negative control. The culture media from differently treated 4T1 cells-incubated M2-like RAW264.7 cells were set as the control, DOX + Apa, HDA, HDD, and HDDA groups, respectively. $n = 4$ biologically independent samples per group. * $P < 0.05$, ** $P < 0.01$, *** $P < 0.001$, **** $P < 0.0001$.

On the other hand, we completely agree with this respected reviewer that the knockdown and blockage experiments can strongly assist in proving the immunostimulation effect of these

DAMPs. As suggested, we used the CRT short hairpin RNA (shRNA)-encapsulated lentivirus to transfect 4T1 cells, and we hoped that the stable transgenic cell lines with CRT knockdown can be screened. Unfortunately, we did not successfully screen out CRT knockdown cell lines, possibly owing to the insufficient virus titer. As for the blockage experiment, the interferon receptor antibody cannot be purchased due to the COVID-19 pandemic. In fact, the association and biological mechanism between the release of DAMPs and immunogenicity have been reported in many articles (*Nat. Rev. Cancer*, 12, 860–875 (2012).; *Nat. Rev. Immunol.*, 17, 97–111 (2017).). Moreover, many articles have pointed out that the production of DAMPs is directly related to immunogenicity (*Nat. Biomed. Eng.*, 4, 1102–1116 (2020).; *Sci. Adv.* 2020, 6, eabc4373.; *Nat. Commun.*, 11, 4951, (2020).). Thus, we think that the above data (Fig. 3 and Figs. R1–4) can adequately validate our conclusion (the immunostimulation effect of HMVs).

We greatly appreciate this respected reviewer for his/her very precious suggestions, and we will also try these experiments in subsequent studies when conditions permit.

Comment 6: (vi) HDDA nanobomb showed certain level of biocompatibility, e.g. HDDA showed low level of hemolysis in vitro, no major changes were observed in routine blood or histological tissue analysis in vivo and no apparent weight loss, but HDDAs targeting to other tissues (Suppl. 19, 20) and off target effect was not examined in detail.

Author reply: We deeply thank the reviewer for this very important comment! First, we have provided some experiments to prove the biocompatibility of HDDA NBs in the manuscript (e.g., the hemolysis analysis, the routine blood and biochemical indexes analyses, the images of H&E-stained tissue slices of major organs, and the body weight monitoring results), and we thank this reviewer for his/her recognition of this part of the experiments.

As for the HDDAs targeting to other tissues, we have provided the in vivo and ex vivo fluorescence images of the 4T1 tumor-bearing BALB/c mice at various time points after intravenous injection of the HDDA NBs (Supplementary Figs. 31 and 32). The results indicated that some fluorescence signals of the DiR@HDDA NBs were detected in the liver and kidneys within 24 h postinjection, and the fluorescence levels in the major organs and tumor sites became very low at 7 d postinjection, suggesting that the NBs could be excreted by the mice via liver and kidneys after a certain time period.

Next, we further studied the off-target effects of the HDDA NBs in detail as suggested. First, the Masson and Van Gieson (VG) staining of tissue slices of the major organs (hearts, livers, spleens, lungs, and kidneys) excised from the BALB/c mice 14 d after PBS or HDDA treatment were carried out to visualize the collagen fibers of the various tissues. The results illustrated that no obvious tissue damage of the major organs was observed after HDDA treatment (Figs. R5 and R6). Moreover, no tissue damage was observed in the pictures of the hexamine silver-stained tissue slices of the lungs and kidneys after HDDA treatment (Fig. R7). Finally, only sporadic TUNEL staining signals (green) were detected in the tissue slices of the major organs in the HDDA group, demonstrating that HDDA did not cause cell apoptosis in normal tissues (Fig. R8). Besides, the urine routine test was also carried out for the BALB/c mice 14 d after intravenous injection of PBS (control) or HDDA NBs. All the urine routine indexes in the HDDA group were similar to those in the control group and remained in the normal ranges, further ensuring the biosafety of the HDDA NBs (Fig. R9).

Since the HMVs were prepared from 4T1 tumor cells, we further studied their proliferative ability in vitro. The crystal violet staining and MTT assay results indicated that the HMVs could not proliferate like live 4T1 cells, ensuring their further applications in vivo (Fig. R10). The above discussions and the figures have been added in the revised manuscript (Page 27). Thanks a lot!

Fig. R5. Masson-stained tissue slices of the major organs (hearts, livers, spleens, lungs, and kidneys) excised from the BALB/c mice sacrificed on the 14th day after intravenous injection of PBS (control) or HDDA suspension (DOX dose: 5 mg/kg).

Fig. R6. VG-stained tissue slices of the major organs (hearts, livers, spleens, lungs, and kidneys) excised from the BALB/c mice sacrificed on the 14th day after intravenous injection of PBS (control) or HDDA suspension (DOX dose: 5 mg/kg).

Fig. R7. Hexamine silver-stained tissue slices of the lungs and kidneys excised from the BALB/c mice sacrificed on the 14th day after intravenous injection of PBS (control) or HDDA suspension (DOX dose: 5 mg/kg).

Fig. R8. Confocal microscopic images of representative TUNEL assay results of the major organs (hearts, livers, spleens, lungs, and kidneys) excised from the BALB/c mice sacrificed on the 14th day after intravenous injection of PBS (control) or HDDA suspension (DOX dose: 5 mg/kg).

Fig. R9. Semiquantitative urine routine test results of the BALB/c mice collected on the 14th day after intravenous injection of PBS (control) or HDDA suspension (DOX dose: 5 mg/kg). $n = 4/\text{group}$. The urine indexes including WBC, KET, NIT, URO, BIL, PRO, GLU, pH, and BLD indicate the number of white blood cells, concentration of ketone body, nitrite, urobilinogen, bilirubin, protein, and glucose, the pH value, and the occult blood, respectively.

Fig. R10. Crystal violet staining (a) and MTT assay (b) results showing the cell viabilities of HMVs and 4T1 cells. The RPMI 1640 culture medium (blank) was used as negative control.

Comment 8: Importantly, the proposed role of HDDA-treated tumor cell-derived vesicles is not supported by the experiments. The authors did not characterize the vesicles released from HDDA-treated cultured cells, as proposed in Minimal information for studies of extracellular vesicles 2018 (published in the societies' journal JEV in 2018). Therefore, it is not really clear what these vesicles represent and how they separate in characteristics from cells/cell debris and importantly, if they contain HDDA nanobombs or not.

Author reply: This is a very professional comment! As suggested, we carefully read the mentioned paper and further supplemented the characterization of HMVs. As one of the most widely used cell debris, the sonication-induced cell debris ("Soni" sample) was also prepared for comparison. The preparation and isolation procedures of HMVs and "Soni" sample were provided in the response to **Comment 9** below. Since basic characterizations of HMVs (e.g., vesicle size, vesicle generation process, and expression levels of some representative proteins) have already been provided in the manuscript (Fig. 1f-i and Supplementary Figs. 9, 10, 12, 13, 16, 17, 18, 25, and 33), we mainly focused on the protein profile of HMVs here. First, the sodium dodecyl sulfate-polyacrylamide gel electrophoresis (SDS-PAGE) assay was carried out to study the cytoproteins (control), "Soni" sample proteins, and HMV proteins. The results indicated that the protein profile of HMVs was very different from the other two samples,

demonstrating that HMVs were different from cells and cell debris (Fig. R11).

Fig. R11. SDS-PAGE results of the cytoproteins (control), “Soni” sample proteins, and HMV proteins in (or from) 4T1 cells. M: protein marker.

Next, to further study the protein profile of HMVs, the proteomic analyses of 4T1 cells (control), 4T1 cell-derived “Soni” sample, and HMVs were carried out. The statistic results of total detected proteins and the Venn diagram revealed that the 4T1 cells (control) and the “Soni” sample contained more types of proteins than the HMVs (Control: 3603, “Soni” sample: 3412, HMVs: 1496) (Fig. R12a,b), indicating that most of the proteins were lost during the preparation and isolation process of HMVs. Compared with 4T1 cells, 1970 kinds of proteins were absent in HMVs and 44 kinds of proteins were only present in HMVs (Fig. R12c). These 44 types of proteins may play an important role in the generation of HMVs, which would be analyzed later. Besides, compared with 4T1 cells, 288 kinds of proteins were downregulated and 255 kinds of proteins were upregulated in HMVs (Fig. R12c). On the contrary, the 4T1 cells and “Soni” sample shared similar protein profiles (Fig. R12a–c), indicating that most of the proteins were retained when preparing the “Soni” sample. Then, the profiles of the 1426 kinds of proteins shared by the three samples were shown via the heat map (Fig. R12d). The results indicated the control and “Soni” sample had similar protein profiles with regard to these shared proteins, and the protein profile of HMVs was largely different from those of the other two samples. Furthermore, the cytolocalization of the detected proteins of these three samples was analyzed. The results indicated that HMVs contained the proteins from all the subcellular

structures of the cells, the cytolocalizations of the proteins were similar in these three samples, and the proportions of the proteins from lysosome, Golgi apparatus, and endoplasmic reticulum (ER) in the HMVs were slightly higher than those of the other two samples (Fig. R12e–g).

Fig. R12. Proteomic analyses of the cytoproteins (control), "Soni" sample proteins, and HMV proteins in (or from) 4T1 cells. **a**, Statistics showing the detected number of protein types in each sample. $*P < 0.05$, $****P < 0.0001$. **b**, Venn diagram of the detected proteins in the control, "Soni" sample, and HMVs groups. The numbers in the figure indicate the number of the detected protein types. **c**, Statistics showing the protein numbers of "presence", "absence", "upregulation", and "downregulation" for HMVs vs "Soni" sample, "Soni" sample vs Control, and HMVs vs Control, respectively. The numbers in the figure indicate the number of the protein types. **d**, Heat map showing the levels of shared proteins in the control, "Soni" sample, and HMVs groups. Red: upregulation; blue: downregulation; P value < 0.05 and fold change $>$

2.0. **e–g**, Cytolocalization of the detected proteins in the control, “Soni” sample, and HMVs groups, respectively. The numbers in the middle of the cycles indicate the total number of the protein types.

Furthermore, we compared the protein profiles of the control and HMVs samples. First, the representative proteins that could only be detected in the HMVs sample were analyzed (Fig. R13a and Table R1). Notably, some extracellular matrix (ECM)- and cell adhesion-related proteins were detected in HMVs (e.g., secreted protein, acidic and rich in cysteine (SPARC), transforming growth factor-beta-induced protein ig-h3, prolyl endopeptidase fibroblast activation protein (FAP), neogenin, ECM protein 1, and thrombospondin-4), indicating that the ECM and cell adhesion ability of cells were influenced during the HMVs-producing process. Then, several protein kinase-related proteins (e.g., epidermal growth factor (EGF)-containing fibulin-like ECM protein 1, protein kinase C-binding protein NELL2, and neurobeachin) were also detected in HMVs, indicating that the TKI Apa (an important component of HDDA NBs) played an important role in HMV production. Moreover, some immune system-related proteins (e.g., prolyl endopeptidase FAP, cell adhesion molecule 1, and complement C1q tumor necrosis factor (TNF)-related protein 3) in HMVs may promote the immunostimulation effect of HMVs. Next, the proteins that were overexpressed in HMVs were also analyzed (Fig. R13b and Table R2, we only choose the fold change > 10.0 proteins). The results further verified the important role of protein kinases in HMV production, and importantly, some immune system-related proteins (e.g., protein canopy homolog 4, mitogen-activated protein kinase 4, cation-independent mannose-6-phosphate receptor, junctional adhesion molecule A, coxsackievirus and adenovirus receptor homolog, and X-linked inhibitor of apoptosis protein (XIAP)-associated factor 1) were also overexpressed in HMVs, which might be the reason of the strong immunostimulation effect of HMVs. Besides, some lysosome-, Golgi-, and ER-related proteins were detected in HMVs (Fig. R13b and Table R2) and the protein content in HMVs from these three organelles was also higher than that in control sample (Fig. R13e–f), indicating that lysosome, Golgi apparatus, and ER participated in the production of HMVs.

Moreover, the Gene Ontology (GO) and Kyoto Encyclopedia of Genes and Genomes (KEGG) enrichment analyses of the control and HMVs samples were conducted. The GO term

analysis revealed that the proteins related with different kinds of membranes and membrane-containing organelles were overexpressed in HMVs, indicating that the HMVs were derived from different membranes (Fig. R13c). Some immune-related proteins were also detected by the GO and KEGG analyses (Fig. R13c,d), ensuring the immunostimulation effect of HMVs. The above discussions, figures, and tables have been added in the revised manuscript (Page 13). Thanks a lot!

Fig. R13. Comparison of the proteomic results of the control and HMVs groups. a, Representative proteins that are only detected in the HMVs group. SPARC: secreted protein, acidic and rich in cysteine, FAP: fibroblast activation protein, AMP: adenosine monophosphate, NELL2: neural epidermal growth factor-like 2, EGF: epidermal growth factor, TNF: tumor necrosis factor. **b,** Representative proteins that are overexpressed in the HMVs group. ER: endoplasmic reticulum, ATPase: adenosine triphosphatase, MORC2A: microorchidia 2A, TIAR:

TIA-1 (T cell cytoplasmic antigen)-related nucleolysin, XIAP: X-linked inhibitor of apoptosis protein. P value < 0.05 and fold change > 10.0 . $*P < 0.05$, $**P < 0.01$, $***P < 0.001$, $****P < 0.0001$. **c**, GO term enrichment analysis of the representative differentially expressed proteins (DEPs) that are overexpressed in the HMVs group. BP: biological process, MF: molecular function, CC: cellular component, ER: endoplasmic reticulum, MAPK: mitogen-activated protein kinase, MHC: major histocompatibility complex. P value < 0.05 . The numbers in the figure indicate the rich factors. **d**, Dot plot showing the KEGG enrichment analysis results of representative DEPs between the HMVs group and the control group. P value < 0.05 . PPAR: peroxisome proliferators-activated receptor, mTOR: mammalian target of rapamycin, ROS: reactive oxygen species, ECM: extracellular matrix.

As for if HMVs contain HDDA NBs, we first used confocal microscopy to visualize the fluorescence of DOX in the HDDA NBs, the results indicated that the fluorescence of DOX were not detectable in HMVs (Figs. R14 and R15, Fig. 1f, and Supplementary Figs. 12, 13, and 16). Furthermore, we also quantified the DOX and Apa molecules (the toxic components in the HDDA NBs) in the HMVs via high-performance liquid chromatography (HPLC). The protein content of HMVs was determined by a bicinchoninic acid (BCA) protein assay kit (Beyotime Institute Biotechnology, Shanghai, China). The HPLC analysis results indicated that the DOX and Apa contents versus the protein content in the HMVs were 0.081% and 0.084%, respectively (Fig. R16). Thus, we could confirm that the contents of DOX and Apa in the HMVs were at extremely low levels and could be reasonably neglected. The above discussions and the figure have been added in the revised manuscript (Page 9). Thanks a lot!

In conclusion, the HMVs have been extensively characterized and they are different from cells and cell debris. Thank you so much for raising such an excellent comment!

Fig. R14. Confocal microscopic images of the 4T1 cells after incubation with different concentrations of HDDA NBs (DOX concentrations in the HDDA NBs were 1.8, 3.6, 5.0, 7.2, 10.0, or 14.5 $\mu\text{g/mL}$, respectively.) for 1 or 4 h. Scale bar: 50 μm .

Fig. R15. Confocal microscopic images of 4T1 cells after incubation with HDDA NBs for 1 h. Before incubation with HDDA NBs, the cells were prestained with Hoechst 33342, LysoTracker green, MitoTracker deep red, or DiO to visualize the cell nucleus, lysosome, mitochondrion, or cell membrane, respectively. The DOX concentration in HDDA was 7.2 $\mu\text{g}/\text{mL}$. The yellow pseudo color was used to represent the fluorescence of MitoTracker deep red.

Fig. R16. a, HPLC traces of HMVs and various concentrations of DOX. Wavelength for detection: 476 nm. **b**, Linear fitting result of the area of the characteristic peak of DOX (between 17 min and 19 min) versus the DOX concentration. **c**, HPLC traces of HMVs and various concentrations of Apa. Wavelength for detection: 341 nm. **d**, Linear fitting result of the area of the characteristic peak of Apa (between 9 min and 12 min) versus the Apa concentration. Protein concentration of the used HMVs: 5.8 mg/mL (determined by the BCA protein assay kit). All the samples were dispersed in 90% methanol/10% H₂O mixed solutions.

Comment 9: The only experiments performed on purified vesicles (the isolation protocol itself is not very clear) are determination of protein concentration and protein profile on SDS-PAGE gel, and the authors claim lower annexinV and caspase 3 presence in vesicles, while detecting CD44 in vesicles, but I could not access those figures. Although CD44 might be present on vesicles, this is not a specific marker and might not be the best way to detect vesicles in tumor tissues. Later, the authors isolated vesicles released from HDDA-treated cells and labelled them with DiD. Again, they provided no characterization of these labelled vesicles (do they contain HDDA nanobombs?), and directly used them in BALB/c and C57BL/6 mouse models to show

immune-modulatory role, but failed to include any controls of DiD-vesicles.

Author reply: This is a very constructive and professional comment! First, the preparation methods of HMVs and sonication-induced cell debris (“Soni” sample) were provided below:

To prepare and isolate HMVs, 4T1 cells were seeded into 6-well plates and incubated for 24 h. Next, the cells were treated with HDDA NBs (DOX: 14.5 µg/mL) for 24 h. The suspensions were collected, washed with phosphate-buffered saline (PBS), and purified via ultrafiltration (MWCO = 100 kDa) for at least three times to obtain the PBS suspension of HMVs. To prepare and isolate “Soni” sample, 1×10^7 4T1 cells were collected via centrifugation (1000 rpm, 5 min), and were resuspended in PBS to obtain the 4T1 cell suspension. Then, the cell suspension was treated with bath sonication for 10 min (200 W) and washed with PBS via ultrafiltration (MW = 100 kDa) for at least three times to obtain the PBS suspension of “Soni” sample. The protein contents of HMVs and “Soni” sample were determined by a bicinchoninic acid (BCA) protein assay kit (Beyotime, China). The detailed experimental procedures have been added in the revised manuscript (Page 45).

On the other hand, the expression levels of CD44, caspase-3, and cleaved caspase-3 were studied via Western blot, and the expression level of annexin V was studied via flow cytometry. The results showed that the caspase-3 and cleaved caspase-3 contents on HMVs were very low, and the annexin V expression level on HMVs was lower than that on the 4T1 cells without apoptosis, which had low annexin V expression level (Figs. R17 and R18). The above results indicated that the HMVs were not apoptotic bodies (ApoBDs), because ApoBDs had high expression levels of annexin V and cleaved caspase-3 (*Front. Immunol.* 2018, 9, 1486 2018.; *Sci. Adv.* 2020, 6, eaba2987.).

As for CD44, the Western blot result indicated that the content of CD44 on HMVs was much higher (more than 3 times) than that on 4T1 cells (Fig. R17), and the immunofluorescence staining image of HMVs further confirmed the high expression level of CD44 on HMVs (Fig. R19). Thus, we detected HMVs in tumor tissues by visualizing CD44 molecules of the tumor tissues. The result in Fig. 4h (revised manuscript) indicated that although tumor cells also expressed CD44 to a certain extent, the expression of CD44 on HMVs was much higher than that on tumor cells. The in vivo result was consistent with the in vitro one. Anyway, we still agree with this respected reviewer that CD44 is possibly not a specific marker of HMVs, and

CD44 staining may be not the best way to detect HMVs *in vivo*. For this reason, CD44 was only used for supplementary verification, and Bio-TEM was also used to observe the *in situ* HMV production in the HDDA-treated tumors in a more direct way (Fig. 4i and supplementary Figs. 34 and 35). Thus, we confirmed that HMVs were produced in the tumor tissues after HDDA treatment. As for finding the specific marker for HMVs, further in-depth mechanism researches are needed, which may not be covered by this work, and we will continue our exploration in future researches.

Fig. R17. (a) Western blot results and (b) corresponding statistical data of the CD44, caspase-3, and cleaved caspase-3 expression levels in the 4T1 cells, cell membrane (4T1), cytoplasm (4T1), and HMVs that were collected from 4T1 cells. GAPDH was used as the loading control.

Fig. R18. Flow cytometric results showing the relative annexin V expression levels in 4T1 cells and HMVs.

Fig. R19. Confocal microscopic images of the representative immunofluorescence staining results of the CD44 (green) protein in HMVs collected from the HDDA-treated 4T1 cells.

As for the characterization of DiD@HMV, we collected confocal microscopic images of the DiD@HMV as suggested. The results confirmed the successful loading of DiD molecules on HMVs, and there was almost no fluorescence of DOX molecule in the DiD@HMV (Fig. R20).

Fig. R20. Confocal microscopic images of the DiD@HMV. The green pseudo color was used to represent the fluorescence of DOX, and the red color represents the fluorescence of DiD.

Next, we further performed the *in vivo* immunostimulation effect and the tumor prevention ability of HMVs and used the “Soni” sample (one of the most widely used cancer cell debris) as a new control as suggested. The BALB/c mice were subcutaneously (s.c.) injected with PBS (control), “Soni” sample (100 μ L, protein concentration: 10 mg/mL), or HMVs (100 μ L, protein concentration: 10 mg/mL), and the lymph nodes (LNs) were collected 2 d after injection. The flow cytometry results indicated that compared with the other two groups, the mice after HMV treatment showed significant enrichment of matured DCs (CD11c⁺CD86⁺CD80⁺), M1-like macrophages (F4/80⁺CD11c⁺), cytotoxic T cells (CD3⁺CD8⁺), and helper T cells (CD3⁺CD4⁺) in LNs (Fig. R21). The enhanced immunostimulation effect of HMVs may be attributed to the

higher proportion of immune-related proteins in HMVs.

Fig. R21. Representative flow cytometric plots of matured DCs (CD11c⁺CD80⁺CD86⁺) (a), M1-like macrophages (CD11c⁺F4/80⁺) (b), helper T cells (CD3⁺CD4⁺), and cytotoxic T cells (CD3⁺CD8⁺) (c) in the LNs retrieved from BALB/c mice 2 d after PBS (control), "Soni" sample, or HMV treatment.

Moreover, the prevention of tumor growth in the BALB/c mice inoculated with PBS (control), HMVs, or "Soni" sample was evaluated. Compared with the other two groups, the HMVs showed the most significant tumor suppression during the 30 d observation period with negligible body weight loss (Fig. R22a–d). In addition, a large proportion of the tumor tissues was destroyed in the HMVs group as revealed by the representative H&E- and TUNEL assay kit-stained tumor tissue slices (Fig. R22e,f). The above discussions and the figures have been added in the revised manuscript (Page 30). Thanks a lot!

Fig. R22. HMVs as cancer vaccines for tumor prevention. (a) Experimental procedure for assessing the 4T1 tumor prevention capacity of the “Soni” sample or HMVs in BALB/c mice. (b,c) Individual 4T1 tumor growth curves with the complete tumor regression (CR) fractions (b) and average 4T1 tumor growth curves (c) of the BALB/c mice pretreated with PBS (control), “Soni” sample, or HMVs. (d) Body weight changes of the 4T1-bearing BALB/c mice after different treatments ($n = 4/\text{group}$). (e,f) Confocal microscopic images of representative H&E staining results (scale bar: $300\ \mu\text{m}$) (e) and TUNEL assay results (scale bar: $100\ \mu\text{m}$) (f) of the tumor tissue slices from the BALB/c mice pretreated with PBS (control), “Soni” sample, or HMVs before 4T1 tumor implantation.

Comment 10: The authors also failed to report on previous studies on apoptotic vesicles in general and specifically in cancer in the Introduction (reviewed in Gregory et al, Front Immunol. 2018; Muhsin-Sharafaldine et al, Immunol Cell Biol, 2018), so it is not clear how familiar are

they with the subject.

Author reply: This is a very professional and constructive comment! As suggested, we carefully read the mentioned references and introduced some previous studies on apoptotic vesicles. Besides, we also proved that the HMTVs belong to other types of extracellular vesicles (EVs) but not ApoBDs or ApoMVs in the revised manuscript (Page 11) and in the response to *Comment 9*.

Table R1. Names and functions of the proteins shown in Fig. R13a. Protein functions were provided by Applied Protein Technology (Shanghai, China).

Protein name	Function
SPARC	Appears to regulate cell growth through interactions with the extracellular matrix and cytokines.
Albumin	Binds water, Ca ²⁺ , Na ⁺ , K ⁺ , fatty acids, hormones, bilirubin, and drugs. Its main function is the regulation of the colloidal osmotic pressure of blood.
4-Hydroxyphenylpyruvate dioxygenase	Key enzyme in the degradation of tyrosine.
Transforming growth factor-beta-induced protein ig-h3	Plays a role in cell adhesion. May play a role in cell-collagen interactions.
Prolyl endopeptidase FAP	Cell surface glycoprotein serine protease that participates in ECM degradation and involved in many cellular processes including tissue remodeling, fibrosis, wound healing, inflammation, and tumor growth. Participates in the cell invasiveness towards the ECM in malignant melanoma cancers. Enhances tumor growth progression by increasing angiogenesis, collagen fiber degradation, and apoptosis and by reducing antitumor response of the immune system. Acts as a tumor suppressor in melanocytic cells through regulation of cell proliferation and survival in a serine protease activity-independent manner.
Neogenin	Multi-functional cell surface receptor regulating cell adhesion in many diverse developmental processes, including neural tube and mammary gland formation, myogenesis and angiogenesis.
Receptor-type tyrosine-protein phosphatase gamma	Possesses tyrosine phosphatase activity.

Cyclic AMP-dependent transcription factor-4	AMP-Transcription factor that binds the cAMP response element (CRE) and acts both as a regulator of normal metabolic and redox processes, and as a master transcription factor during the integrated stress response (ISR). Core effector of the ISR, which is required for adaptation to various stress, such as ER stress, amino acid starvation, mitochondrial stress or oxidative stress. Protects cells against metabolic consequences of ER oxidation by promoting expression of genes linked to amino acid sufficiency and resistance to oxidative stress. Together with DDIT3/CHOP, mediates ER-mediated cell death by promoting expression of genes involved in cellular amino acid metabolic processes, mRNA translation and the unfolded protein response (UPR) in response to ER stress. Activates expression of genes required to promote cell recovery in response to mitochondrial stress.
Protein kinase C-binding protein NELL2	C- Required for neuron survival through the modulation of MAPK pathways.
ECM protein 1	Stimulates the proliferation of endothelial cells and promotes angiogenesis. Inhibits MMP9 proteolytic activity.
Regucalcin	Gluconolactonase with low activity towards other sugar lactones, including gulonolactone and galactonolactone. Catalyzes a key step in ascorbic acid (vitamin C) biosynthesis. Calcium-binding protein. Modulates Ca ²⁺ signaling, and Ca ²⁺ -dependent cellular processes and enzyme activities.
Complement component C8 beta chain	Constituent of the membrane attack complex (MAC) that plays a key role in the innate and adaptive immune response by forming pores in the plasma membrane of target cells.
EGF-containing fibulin-like protein 1	Binds EGFR, the EGF receptor, inducing EGFR autophosphorylation and the activation of downstream signaling pathways. May play a role in cell adhesion and migration.
Cell adhesion molecule 1	Mediates homophilic cell-cell adhesion in a Ca ²⁺ -independent manner. Interaction with CRTAM promotes natural killer (NK) cell cytotoxicity and interferon- γ (IFN- γ) secretion by CD8 ⁺ T-cells in vitro as well as NK cell-mediated rejection of tumors expressing CADM1 in vivo. By interacting with CRTAM and thus promoting the adhesion between CD8 ⁺ T-cells and CD8 ⁺ dendritic cells, regulates the retention of activated CD8 ⁺ T-cell within the draining lymph node. Required for the intestinal retention of intraepithelial CD4 ⁺ and CD8 ⁺ T-cells and, to a lesser extent, intraepithelial and lamina propria CD8 ⁺ T-cells and CD4 ⁺ T-cells. Interaction with CRTAM promotes the adhesion to gut-associated CD103 ⁺ dendritic cells, which may facilitate the expression of gut-homing and adhesion molecules on T-cells and the conversion of CD4 ⁺ T-cells into CD4 ⁺ and CD8 ⁺ T-cells.
Plexin domain-containing protein 2	May play a role in tumor angiogenesis.

Neurobeachin	Binds to type II regulatory subunits of protein kinase A and anchors/targets them to the membrane. May anchor the kinase to cytoskeletal and/or organelle-associated proteins. May have a role in membrane trafficking.
Complement C1q	TNF-related protein 3
Sacsin	Co-chaperone which acts as a regulator of the Hsp70 chaperone machinery and may be involved in the processing of other ataxia-linked proteins.
Cadherin-13	Cadherins are calcium-dependent cell adhesion proteins. They preferentially interact with themselves in a homophilic manner in connecting cells; cadherins may thus contribute to the sorting of heterogeneous cell types.
Thrombospondin-4	Adhesive glycoprotein that mediates cell-to-cell and cell-to-matrix interactions and is involved in various processes including cellular proliferation, migration, adhesion, and attachment. Binds to structural ECM proteins and modulates the ECM in response to tissue damage, contributing to cardioprotective and adaptive ECM remodeling. Plays a role in ER stress response, via its interaction with the activating transcription factor 6 alpha (ATF6) which produces adaptive ER stress response factors and protects myocardium from pressure overload.

Table R2. Names and functions of the proteins shown in Fig. R13b. Protein functions were provided by Applied Protein Technology (Shanghai, China).

Protein name	Function
ER membrane protein complex subunit 4	May mediate anti-apoptotic activity.
Signal transducing molecule 1	Involves in intracellular signal transduction mediated by cytokines and growth factors. Upon IL-2 and GM-CSF stimulation, it plays a role in signaling leading to DNA synthesis and MYC induction. May also play a role in T-cell development. Involved in down-regulation of receptor tyrosine kinase via multivesicular body (MVBs) when complexed with HGS (ESCRT-0 complex). The ESCRT-0 complex binds ubiquitin and acts as sorting machinery that recognizes ubiquitinated receptors and transfers them to further sequential lysosomal sorting/trafficking processes.
Fibronectin	Fibronectins bind cell surfaces and various compounds including collagen, fibrin, heparin, DNA, and actin. Fibronectins are involved in cell adhesion, cell motility, opsonization, wound healing, and maintenance of cell shape. Fibronectin polymer, named superfibronectin, exhibits enhanced adhesive properties. Both anastellin and superfibronectin inhibit tumor growth, angiogenesis and metastasis.

ATPase MORC2A	During DNA damage response, regulates chromatin remodeling through ATP hydrolysis. During DNA damage response, may regulate chromatin remodeling through ATP hydrolysis.
Cyclin-dependent kinase 2	Serine/threonine-protein kinase involved in the control of the cell cycle; essential for meiosis, but dispensable for mitosis. Phosphorylates CTNNB1, USP37, p53/TP53, NPM1, CDK7, RB1, BRCA2, MYC, NPAT, EZH2.
Protein canopy homolog 4	Plays a role in the regulation of the cell surface expression of TLR4.
Endoglin	Vascular endothelium glycoprotein that plays an important role in the regulation of angiogenesis.
Golgin subfamily A member 4	Involved in vesicular trafficking at the Golgi apparatus level. May play a role in delivery of transport vesicles containing GPI-linked proteins from the trans-Golgi network through its interaction with MACF1. Involved in endosome-to-Golgi trafficking.
Mitogen-activated protein kinase 4	Serine/threonine kinase that may play a role in the response to environmental stress and cytokines such as TNF- α .
Guanine nucleotide exchange factor DBS	Guanine nucleotide exchange factor that catalyzes guanine nucleotide exchange on RHOA and CDC42, and thereby contributes to the regulation of RHOA and CDC42 signaling pathways. Seems to lack activity with RAC1. Becomes activated and highly tumorigenic by truncation of the N-terminus.
Cation-independent mannose-6-phosphate receptor	Acts as a positive regulator of T-cell coactivation, by binding DPP4. Transport of phosphorylated lysosomal enzymes from the Golgi complex and the cell surface to lysosomes. Lysosomal enzymes bearing phosphomannosyl residues bind specifically to mannose-6-phosphate receptors in the Golgi apparatus and the resulting receptor-ligand complex is transported to an acidic prelysosomal compartment where the low pH mediates the dissociation of the complex. This receptor also binds IGF2.
Aspartyl/asparaginyl hydroxylase	beta-Specifically hydroxylates an Asp or Asn residue in certain epidermal growth factor-like (EGF) domains of a number of proteins.
Ras-related protein Rab-32	Acts as an A-kinase anchoring protein by binding to the type II regulatory subunit of protein kinase A and anchoring it to the mitochondrion. Also involved in synchronization of mitochondrial fission.
Adapter molecule crk	Involved in cell branching and adhesion mediated by BCAR1-CRK-RAPGEF1 signaling and activation of RAPI. Isoform CRK-II: Regulates cell adhesion, spreading and migration. Mediates attachment-induced MAPK8 activation, membrane ruffling and cell motility in a Rac-dependent manner. Involved in phagocytosis of apoptotic cells and cell motility via its interaction with DOCK1 and DOCK4.
Junctional adhesion molecule A	Plays a role in regulating monocyte transmigration involved in integrity of epithelial barrier. Ligand for integrin alpha-L/beta-2 involved in

	memory T-cell and neutrophil transmigration. Involved in platelet activation.
Nucleolysin TIAR	RNA-binding protein. Possesses nucleolytic activity against cytotoxic lymphocyte target cells. May be involved in apoptosis.
Vacuolar protein sorting-associated protein 4A	Involved in late steps of the endosomal multivesicular bodies (MVB) pathway. Recognizes membrane-associated ESCRT-III assemblies and catalyzes their disassembly, possibly in combination with membrane fission. Redistributes the ESCRT-III components to the cytoplasm for further rounds of MVB sorting. MVBs contain intraluminal vesicles (ILVs) that are generated by invagination and scission from the limiting membrane of the endosome and mostly are delivered to lysosomes enabling degradation of membrane proteins, such as stimulated growth factor receptors, lysosomal enzymes and lipids.
Coxsackievirus and adenovirus receptor homolog	Component of the epithelial apical junction complex that may function as a homophilic cell adhesion molecule and is essential for tight junction integrity. It results in proliferation and production of cytokines and growth factors by T-cells that in turn stimulate epithelial tissues repair.
Uridine-cytidine kinase 2	Phosphorylates uridine and cytidine to uridine monophosphate and cytidine monophosphate. Does not phosphorylate deoxyribonucleosides or purine ribonucleosides. Can use ATP or GTP as a phosphate donor. Can also phosphorylate cytidine and uridine nucleoside analogs.
XIAP-associated factor 1	Seems to function as a negative regulator of members of the IAP (inhibitor of apoptosis protein) family. Inhibits anti-caspase activity of BIRC4. Induces cleavage and inactivation of BIRC4 independent of caspase activation. Mediates TNF- α -induced apoptosis and is involved in apoptosis in trophoblast cells. May be a tumor suppressor by mediating apoptosis resistance of cancer cells.

Responses to Reviewer 3:

General comment: In this manuscript, authors reported a strategy based on the HDDT NBs to induce cancer cell death and realize the in situ massive production of HMs as cancer vaccines. The work was systematically investigated. The results obtained seem to be interesting. I think it can be accepted for publication after some revisions.

Author reply: We deeply appreciate this respected reviewer for raising these very helpful suggestions! We are very grateful for the precious time, invaluable expertise, and superb professionalism the reviewer has put in improving the quality of our work!

Comment 1: Authors declared that "The encapsulation ratio (ER) of both Apa and DOX in HDDA NBs was almost 100% ". If the Apa and DOX were co-encapsulated into Den simply via hydrophobic interaction, the drugs were not completely encapsulated. Please give the preparation process and the calculation method for ER.

Author reply: This is a very important and professional comment! As suggested, the preparation processes of the HDDA NBs as well as other HDDT NBs were provided in the revised manuscript. Briefly, Den was dissolved in methanol (1 mM). DOX•HCl was also dissolved in methanol and was neutralized with triethylamine to obtain the HCl-free DOX solution (10 mM). Apa was dissolved in methanol with a concentration of 5.8 mg/mL. Then 50 μ L of DOX solution, 25 μ L of Apa solution, and 150 μ L of Den solution were mixed and the suspension was vortexed for 20 s, followed by evaporation under nitrogen gas. The obtained powder was resuspended with 1 mL of deionized water under bath sonication for 2 min, and the resultant suspension was centrifuged (10000 rpm for 10 min) to remove the precipitates. The DDA NPs were in the supernatant and the concentrations of DOX and Apa in the NPs were quantified by UV-vis spectroscopy.

To calculate the ER of Apa and DOX in DDA NBs, the UV-vis spectra of the methanol solutions of different concentrations of Apa and DOX were measured first (Fig. R1a,c). The absorbance values at 341 nm and 476 nm of Apa and DOX were used for their quantification, respectively. As shown in Fig. R1b,d, the standard curves of Apa and DOX were also obtained by the linear fitting of the absorbance values at 341 and 476 nm in the UV-vis spectra versus

the concentration of Apa or DOX, respectively. Then, the UV–vis spectrum of the methanol solution of the purified DDA solution was measured and the absorbance values at 341 nm and 476 nm were compared with the standard curves of Apa and DOX, respectively. The ER values of Apa and DOX were calculated via the equation (1) shown below:

$$\text{ER (\%)} = \text{Weight of Apa (or DOX) loaded in DDA} / \text{Weight of Apa (or DOX) fed initially} \times 100\% \quad (1)$$

After calculation, the ER values of Apa and DOX were ~85% and ~100%, respectively. The ER values were slightly different for each preparation, possibly due to the temperature, manipulation, and the different batches of drugs. We have accordingly modified the related data in the revised manuscript.

For HA coating, 1 mL of the above-obtained DDA suspension (with the DOX concentration of 0.5 mM) was added to 1 mL of HA solution (1 mg/mL) dropwise under agitation and the mixture was further vortexed for 20 s to form the HDDA NBs. The resultant HDDA suspension was dialyzed (MWCO = 250 kDa) to remove the free HA molecules. We have confirmed via the UV–vis spectra that the above process did not influence the ER values of Apa and DOX, possibly because they are hydrophobic molecules.

Fig. R1. a–d, UV–vis absorption spectra of the solutions of different concentrations of Apa and DOX (solvent: methanol) (**a**, **b**) and the corresponding linear fitting results of the absorbance of Apa and DOX (at 341 nm and 476 nm, respectively) versus the Apa or DOX concentration (**c**, **d**). (**e**) UV–vis absorption spectrum of the methanol solution of DDA NPs.

Comment 2: How were the cancer cells changed to HMVs? It is not clearly described in text. In Fig.1f, there were almost no micrometer-sized vesicles in the confocal microscopic images of the HDDA-treated 4T1 cells at 1 h, however, they could be found in Fig.1g,k. The cell membrane, the nuclei and lysosomes can be labelled with fluorescent probe, and the HDDA-treated cancer cells can be observed by confocal images to visualize the cancer cells change.

Author reply: This is a very professional and constructive comment! First, the concentration of the used HDDA in Fig. 1f (DOX concentration: 3.6 $\mu\text{g}/\text{mL}$) was lower than that used in Fig. 1g,k (DOX concentration: 7.2 $\mu\text{g}/\text{mL}$). The purpose of the experiment of Fig. 1f was to study the subcellular localization of the HDDA NBs, and thus the low content of HDDA was used to ensure the survival of cells. The low content of HDDA was not sufficient for the production of HMVs. Besides, we incubated the 4T1 cells with various concentrations of HDDA NBs (DOX concentration: 1.8, 3.6, 5.0, 7.2, 10, and 14.5 $\mu\text{g}/\text{mL}$) and observed the generation of HMVs. As shown in Fig. R2, massive HMVs were observed in the 7.2 $\mu\text{g}/\text{mL}$ group after 4 h incubation. Thus, we chose this concentration for the HMV production. The above discussions and the figures have been added in the revised manuscript (Page 8). Thanks a lot!

Fig. R2. Confocal microscopic images of the 4T1 cells after incubation with different concentrations of HDDA NBs (DOX concentrations in the HDDA NBs were 1.8, 3.6, 5.0, 7.2, 10.0, or 14.5 $\mu\text{g}/\text{mL}$, respectively.) for 1 or 4 h. Scale bar: 50 μm .

Furthermore, as suggested, we used various fluorescent probes to study the subcellular origination of HMVs. The Hoechst 33342 (abbreviated as Hoechst, blue), LysoTracker green (abbreviated as LysoTracker, green), MitoTracker deep red (abbreviated as MitoTracker, yellow (pseudo color)), and DiO (green) were used to visualize cell nucleus, lysosome, mitochondrion, and cell membrane, respectively. The 4T1 cells were pretreated with these dyes and incubated with HDDA NBs (DOX concentration: 7.2 $\mu\text{g}/\text{mL}$) for 1 h. Unfortunately, none of these dyes as well as HDDA (which could be tracked via the red fluorescence of DOX) were observed in the produced HMVs (see the enlarged images), possibly because the dyes dropped during the efflux (formation) process of HMVs (Fig. R3).

To figure out the subcellular origination of HMVs, we performed the proteomic analysis of HMVs and summarized the subcellular localization of the proteins in HMVs. The results indicated that HMVs contained the proteins from all the subcellular structures of the cells, and the cyto localizations of the proteins in HMVs were similar to those of the 4T1 cells and the sonication-induced cell debris (“Soni” sample; which was obtained via the bath sonication of 4T1 cells and purified by ultrafiltration. The detailed preparation process was provided in the revised manuscript (Page 45)), and the proportions of the proteins from lysosome, Golgi apparatus, and ER in the HMVs were slightly higher than those of the other two samples (Fig. R4). The above discussions and the figures have been added in the revised manuscript (Page 13). Thanks a lot!

Fig. R3. Confocal microscopic images of 4T1 cells after incubation with HDDA NBs for 1 h. Before incubation with HDDA NBs, the cells were prestained with Hoechst 33342, LysoTracker green, MitoTracker deep red, or DiO to visualize the cell nucleus, lysosome, mitochondrion, or cell membrane, respectively. The DOX concentration in HDDA was 7.2 $\mu\text{g}/\text{mL}$. The yellow pseudo color was used to represent the fluorescence of MitoTracker deep red.

Fig. R4. Cytolocalization of the detected proteins in the control (4T1 cells) (a), “Soni” sample (b), and HMVs groups (c), respectively.

Comment 3: Could the HDDA induce normal cell death? Please provide the data.

Author reply: This is a very constructive suggestion! As suggested, we incubated NIH 3T3 (a mouse embryo fibroblast line) and HPAEpiC (a human pulmonary alveolar epithelial cell line) cells with HDDA NBs (DOX concentration: 7.2 $\mu\text{g}/\text{mL}$). After 24 h incubation, the cells were also changed to HMVs, indicating that the HDDA NBs could also induce normal cell death and change them into HMVs (Fig. R5). Thus, we could also produce normal cell line-derived HMVs through the reported method for further applications. The corresponding results and discussions have been added in Page 9 and Supplementary Fig. 12.

Fig. R5. Confocal microscopic images of HPAEpiC and NIH 3T3 cells after HDDA NB treatment. Scale bars: 30 μm . The DOX concentration in the HDDA NBs was 7.2 $\mu\text{g}/\text{mL}$.

Responses to Reviewer 4:

General comment: The authors provide here evidence for the use of naobombs to efficiently convert tumor cells into micrometer sized vesicles, that act as vaccines to induce strong anti-tumor responses. The concept is very interesting based on its effectiveness validate in vitro and in vivo, and the simplicity of the preparation. This could be highly helpful in the context of improved vaccine based cancer immunotherapy.

Yet, I believe that a major revision of the current manuscript is needed to improve the work.

The current manuscript is very long, heavy to read, reporting on an enormous number of results shown in complex Figures. As a consequence, the readers is overwhelmed by many initial in vitro data, while losing the ultimate impact of the findings reported towards the end that attest on the in vivo potential of the NBs. Therefore, the authors should absolutely select the most important results to be displayed in the main figures, shorten and be more concise in the result and conclusion sections, by omitting methodological details and background information.

Also, the current displaying of the panels in the main figures is unusual and disordered. This should be rectified, to facilitate the understanding of the panels.

Author reply: We sincerely appreciate this respected reviewer for his/her very professional comments and suggestions, which help us to improve the quality of the manuscript significantly! As suggested, we have carefully modified the organization and panels of the revised manuscript: For example, Fig. 1f,g has been moved to the supplementary information in the revised manuscript.), and some discussions have also been moved to the supplementary information (Pages 53 and 54). After modification, each figure in the revised manuscript was arranged to demonstrate an integrated problem. Figure 1 shows the preparation, characterization, and HMV production abilities of HDDA NBs; Figure 2 demonstrates the mechanism of the HDDA-induced cell death; Figure 3 investigates the ICD-inducing effect of HDDA NBs *in vitro*; Figure 4 aims to show the antitumor effect of HDDA NBs in the 4T1 tumor model; Figure 5 is used to illustrate the transcriptomic analysis results of the 4T1 tumor tissues after HDDA treatment; Figure 6 shows the immunostimulation and immunological memory effects of HDDA NBs; Figure 7 aims to investigate the antitumor and immunostimulation effects of HDDA NBs in the

B16F10 tumor model. Besides, we used subtitles to classify the contents of these figures in the revised manuscript to facilitate the understanding of the readers.

Nevertheless, if the respected reviewer still feels that some issues were not appropriately addressed, please feel free to tell us, and we will try our best to further improve the mentioned issues. Thank you so much!

Main points:

Comment 1: Introduction: The authors refer to some old references on cancer vaccines (ex Ref 15). Please update with more relevant and recent ones. Also, the authors neglect the success of neoantigen long-synthetic peptide-based vaccinations and mRNA-based vaccines that have given very encouraging results in cancer patients. Please update.

Author reply: We deeply appreciate the respected reviewer for raising this very valuable suggestion! As suggested, we have updated the references in the revised manuscript, and some statements and references about the neoantigen- and mRNA-based vaccines have also been added.

Comment 2: CD44 molecule: the authors claim that the uptake of the NB occurs via CD44. This is an important claim, that however has not been tested experimentally. Is the presence of the molecule correlated with uptake? How is the uptake affected by knocking out CD44? CD44 being a widely expressed molecules, how do the authors explain the limited toxicity to non-tumor cell types? CD44 staining is only shown on one tumor cell line (Supp Fig 3). What about the other ones used, and additional controls.

Author reply: This is a very professional and constructive comment! A large number of reports have shown that HA can recognize HA receptors (such as cluster determinant 44, CD44) overexpressed in various tumor types and thus is widely used for tumor targeting. (*Chem. Rev.* 1998, 98, 2663.; *ACS Appl. Mater. Interfaces* 2017, 9, 32509.) Besides, we pretreated the CD44-overexpressed 4T1 cells with HA before the addition of the HDDA NBs, for blocking the CD44 molecules on the cell surface. The results indicated that the cell internalization of HDDA NBs (which could be tracked via the fluorescence of DOX) was partially influenced by the pretreated HA molecules (Fig. R1), demonstrating the cell internalization of HDDA NBs was related to

the CD44 molecules on the cell surface.

As for the limited toxicity to non-tumor cell types, we have shown that the HDDA NBs were also toxic to non-tumor cell types in vitro and could change these cells into HMVs (Fig. R2). On the other hand, the limited off-target toxicity of the HDDA NBs in vivo was mainly attributed to the enhanced permeability and retention (EPR) effect and the CD44-targeting effect of the HDDA NBs (note that in cellular experiments, the cells might have to “eat” the HDDA NBs even they do not overexpress CD44 molecules because they were exposed to a culture medium full of these nanodrugs; the situation was quite different in the in vivo experiments since the tumor tissue was not passively exposed to a medium full of the nanodrugs), which thus reduced the toxicity of HDDA on normal tissues.

Fig. R1. Flow cytometric results showing the DOX fluorescence intensities in the 4T1 cells after incubation with HDDA NBs (DOX concentration: 3.6 $\mu\text{g}/\text{mL}$) for different time periods as indicated. For the HA + HDDA group, the 4T1 cells were pretreated with HA (1 mg/mL) before incubating with the HDDA NBs. gMFI: geometric mean fluorescence intensity. * $P < 0.05$.

Fig. R2. Confocal microscopic images of HPAEpiC and 3T3 cells after HDDA NB treatment. Scale bars: 30 μ m. The DOX concentration in the HDDA NBs was 7.2 μ g/mL.

We have demonstrated that the CD44-overexpressed 4T1 cells have higher internalization level of HDDA than HMEC-1 and NIH 3T3 cells (Supplementary Fig. 4). As suggested, we also studied the endocytosis of HDDA NBs using two other CD44-overexpressed tumor cell lines (e.g., A549 (a human lung cancer cell line) and MDA-MB-231(a triple-negative human breast cancer cell line)). As shown in Fig. R3, the HDDA endocytosis level of the MDA-MB-231 cells was similar to that of the 4T1 cells and was higher than that of the A549 cells. The different internalization endocytosis levels may be attributed to the various CD44 expression levels of these cell lines. The above discussions and the figures have been added in the revised manuscript (Page 7). Thanks a lot!

Fig. R3. Flow cytometric results showing the DOX fluorescence intensities in the 4T1, A549, and MDA-MB-231 cells after incubation with HDDA NBs (DOX concentration: 3.6 μ g/mL) for 3 h. gMFI: geometric mean fluorescence intensity.

Comment 3: P.9: It is not clear what was the rationale to pursue the studies with HDDA NB. This should be explained.

Author reply: The HDDA NBs were the first synthesized HDDT NBs, and the proportion of various components of the HDDA NBs has been systematically studied and the best drug proportion has been determined. Then, the HMVs-inducing ability of the HDDT NBs were first discovered by using the HDDA NBs, and the HMVs-producing property of the HDDA NBs was carefully investigated by a live cell imaging system. On the other hand, Apatinib (Apa) is one of the most widely used tyrosine kinase inhibitors (TKIs) and is undergoing the phase III clinical trial. Numerous studies have shown the good biosafety and the superb anticancer activity of Apa. Thus, we chose Apa as the representative TKI for our study. Thank you so much for this very professional and constructive comment! The above discussions have been added in the revised manuscript (Page 5). Thanks a lot!

Comment 4: Fig 3: the modulation of M1/M2 should extend to other analyses in addition to flow cytometry based markers. Are these cells functionally converted to M1 macrophages, in terms of secreted cytokines and enzyme content?

Author reply: As suggested, we studied the M2-like macrophage (the IL-4-treated RAW264.7 cells were used as M2-like macrophages)-secreted cytokines (e.g., TNF- α , IL-1 β , and TGF- β) by ELISA assay. Briefly, the 4T1 cells were treated with PBS (control), DOX + Apa, HDA, HDD, or HDDA (DOX concentration: 7.2 $\mu\text{g}/\text{mL}$) for 4 h first, and the culture medium was replaced with fresh culture medium and incubated overnight. Next, the M2-like macrophages were incubated with the above culture medium for 24 h. The macrophage-secreted cytokines were analyzed by ELISA assay.

Compared with the negative control (M2-like macrophages), the levels of proinflammatory cytokines (e.g., TNF- α and IL-1 β) were significantly elevated and the antiinflammatory TGF- β was decreased in the HDDA group (Fig. R4), indicating that the secretions of the HDDA-treated 4T1 cells could repolarize the M2-like macrophages to the M1-like phenotype, which could secrete antitumor proinflammatory cytokines. The above discussions have been added in the revised manuscript (Page 17). Thanks a lot!

Fig. R4. Expression levels of TNF- α , IL-1 β , and TGF- β produced by RAW264.7 cells after different treatments, as analyzed by ELISA. The IL-4-incubated RAW264.7 cells (M2-like macrophages) were set as the negative control. The culture media from differently treated 4T1 cells-incubated M2-like RAW264.7 cells were set as the control, DOX + Apa, HDA, HDD, and HDDA groups, respectively. $n = 4$ biologically independent samples per group. $*P < 0.05$, $**P < 0.01$, $***P < 0.001$, $****P < 0.0001$.

Comment 5: Immune cell infiltration in treated tumors in vivo: the immune cell evaluation is insufficient to support the claims of the authors. The authors should quantify the different immune cell populations, instead of providing a single representative IF image of CD206, FOXP3, ...staining. Similar flow cytometry based analyses as the ones presented in Supp Figure 25 should be provided for M2 macrophages, MDSCs, Tregs, and naïve/memory T cells in the different in vivo treatment settings. Otherwise, it is difficult to conclude and compare the different settings. For instance, from Fig 4j one would conclude that M2 macrophages are similarly reduced in all treatments. “Fewer” red signal is not a quantification term.

Author reply: We sincerely appreciate the reviewer for raising this very important comment! As suggested, we repeated the animal experiments and carefully analyzed the M2-like macrophages (F4/80⁺CD163⁺), MDSCs (CD11b⁺Gr-1⁺), and regulatory T cells (Tregs) (CD3⁺CD4⁺Foxp3⁺) in the tumor regions by flow cytometry. The results indicated the proportions of M2-like macrophages, MDSCs, and Tregs were all decreased in the HDDA group (Fig. R5), which was similar to the results shown in Fig. 4j of the manuscript. The above discussions and figures have been added in the revised manuscript (Page 21). Thanks a lot!

Besides, the memory T cells were also studied and the results were shown in Fig. R7 (response to **Comment 7**) below.

Fig. R5. Representative flow cytometric plots of M2-like macrophages (F4/80⁺CD163⁺), MDSCs (CD11b⁺Gr-1⁺), and regulatory T cells (Tregs) (CD3⁺CD4⁺Foxp3⁺) in the tumor regions retrieved from the 4T1-bearing BALB/c mice 7 d after different treatments.

Comment 6: No indication about sample size is provided in several experiments. In particular, it is not clear on how many samples rely the transcriptomic analyses shown in different figures (ex. Fig 2g). This is highly relevant to define the statistical power of the observations. Also, in Figure 5, how comparable where the results of the 3 mice per group in terms of transcriptomics? How does a PCA comparing treated and not-treated groups show?

Author reply: We greatly thank this reviewer for raising this very professional comment! We agree with this reviewer that the sample size is very important for the credibility of the results, and we have provided the sample sizes for most of the experiments (at least 3 biologically independent samples per group) and also repeated at least twice for some experiments. However, we ignored to provide the sample sizes of some experiments (e.g., Fig. 2g), and as suggested, we have added the ignored sample sizes in the revised manuscript. As for the transcriptomic

analyses, there were two samples per group for the transcriptomic analysis of 4T1 cells and three samples per group for the transcriptomic analysis of 4T1 tumors.

On the other hand, the principal component analysis (PCA) result of the transcriptomic analysis of the 4T1 tumor tissues was provided. As shown in Fig. R6, there are great differences between the treated and nontreated samples.

Fig. R6. PCA result of the transcriptomic analysis samples of the PBS (control)- and HDDA-treated 4T1 tumors.

Comment 7: Immunological memory: no staining on T cells are shown attesting on their memory phenotype in the in vivo settings.

Author reply: As suggested, we studied the central memory T cells (T_{CM} , $CD3^+CD8^+CD44^+CD62L^+$) and effector memory T cells (T_{EM} , $CD3^+CD8^+CD44^+CD62L^-$) in the spleens of the 4T1-bearing BALB/c mice 30 d after different treatments. The flow cytometric results indicated that the percentages of T_{EM} and T_{CM} in the $CD3^+CD8^+$ T cells in

the HDDA group showed a significant increase compared with those in the control mice (Fig. R7). Besides, the HDDA-healed mice could prevent the subcutaneous or intravenous 4T1 tumor rechallenge (Fig. 6h–k, in the revised manuscript). Together, we could confirm the strong immune memory effect of the HDDA-treated mice. Thank you very much for this very valuable suggestion! The above discussions and figure have been added in the revised manuscript (Page 27). Thanks a lot!

Fig. R7. Representative flow cytometric plots of T_{CM} (CD3⁺CD8⁺CD44⁺CD62L⁺) and T_{EM} (CD3⁺CD8⁺CD44⁺CD62L⁻) in the spleens retrieved from the 4T1-bearing BALB/c mice 30 d after different treatments.

Minor:

Comment 8: Why the authors display flow cytometry data (ex Supp Fig 11, Supp Fig 16 and others) as a.u? Better would be to show either % of positive cells of gMFI.

Author reply: We greatly thank this reviewer for raising this very professional comment! As suggested, we have displayed the flow cytometry data as geometric mean fluorescence intensity (gMFI) (e.g., Supplementary Fig. 25, Fig. R1, and Fig. R3). As for Supplementary Fig. 11 (changed to Supplementary Fig. 17 in the revised manuscript), the data were displayed as relative annexin V expression levels, which were calculated based on the gMFI.

Comment 9: No dose dependence studies are shown through the manuscript. For instance, is there in Fig 1h a dose-dependent effect? How was the final dose selected?

Author reply: As suggested, we incubated 4T1 cells with various concentrations of HDDA NBs (DOX concentration: 1.8, 3.6, 5.0, 7.2, 10, and 14.5 µg/mL) and observed the generation of HMVs. As shown in Fig. R8, massive HMVs were observed in the 7.2 µg/mL group after 4 h incubation. Thus, we chose this concentration for the HMVs production. The above

discussions and the figure have been added in the revised manuscript (Page 8). Thanks a lot!

Fig. R8. Confocal microscopic images of the 4T1 cells after incubation with different concentrations of HDDA NBs (DOX concentrations in the HDDA NBs were 1.8, 3.6, 5.0, 7.2, 10.0, or 14.5 $\mu\text{g}/\text{mL}$, respectively.) for 1 h or 4 h. Scale bar: 50 μm .

Comment 10: Fig 4k: when/how were the intratumoral cytokine content analyses performed?

Author reply: This is a very professional comment! First, the tumor tissues of the mice were collected 3 days after various treatments. Then, the weights of the tumor tissues were weighed accurately, and the tumor tissues were mixed with normal saline (the proportion of tissue weight (mg) : normal saline (μL) = 1 : 9). The mixtures were homogenized under ice bath for 10 min to obtain the 10% homogenization suspensions. Next, the suspensions were centrifuged for 10 min at 3000 rpm, and the supernatants were collected for ELISA assay to determine the intratumoral cytokine contents.

Comment II: Fig 6d: it is unclear how the CD4 and CD8 T cells are represented here. The y-axis labels indicate CD4⁺ or CD8⁺ cells in CD3⁺ cells (%). If this is the case the sum of CD4 and CD8 % should be close to 100 in each treatment setting, what is not the case. What is actually displayed in the graphs?

Author reply: We thank the respected reviewer for this very important comment! The proportions of CD4⁺ or CD8⁺ cells shown in Fig. 6d were the quantification results of the corresponding flow cytometric plots in Fig. 6c. The representative flow cytometry gating strategy for CD3⁺CD4⁺ or CD3⁺CD8⁺ T cells was also shown in Supplementary Fig. 43. The the sum of CD4⁺ and CD8⁺ % was less than 100% in Fig. 6d (the highest number was ~59%).

REVIEWERS' COMMENTS

Reviewer #1 (Remarks to the Author):

The revision has addressed most of the reviewers' comments, and the current version is acceptable for publication in NC.

Reviewer #2 (Remarks to the Author):

In the new version of the manuscript "In situ massive generation of micrometer-sized tumor cell-derived vesicles as autologous cancer vaccines for boosting systemic immune responses", the authors have adequately responded to most of my concerns about the nature and the role of tumor cell-derived vesicles (HMMVs) by performing several additional experiments.

One thing that for me still doesn't make sense is that HDDA nanobombs induce apoptosis of target cells, but that according to the authors, HMMVs that are the result of this process are not apoptotic vesicles. How is this logical and what strong data do the authors have to support this? In the EV field, the exact definition of apoptotic EVs is still missing. With new knowledge there is an acceptance that apoptotic EVs can be very heterogeneous in size, and there is also no consensus about their composition.

On another note, I would like to comment on the structure of the manuscript. The manuscript is very long and main sections, like Results or Discussion are not clearly indicated. The authors also included the discussion of their data into the Introduction (in paragraph 3 and 4), instead of reporting on what is already known/published on the topic (including on conundrum about apoptotic EVs). The new results on the proteome of HMMVs are described in the Methods section and not in the Results. The authors sometimes make statements/conclusions in the main text, that are not supported by results or by reference to published data (e.g. p.7: "The different internalization endocytosis levels may be attributed to the various CD44 expression levels of these cell lines."; p. 9: "The differences in the sizes of the produced

HMMVs might be attributed to the different tolerance to HDDA of these cell lines. In particular, the results also indicated that the HDDA NBs could reverse the multidrug resistance of cancer." etc...).

The authors should improve on this and at the same time try to improve the readability of the manuscript, as at the moment it is very hard to follow.

Reviewer #3 (Remarks to the Author):

This manuscript has been revised according to reviewers comments, so I think it can be accepted for publication without change.

Reviewer #4 (Remarks to the Author):

The authors performed an impressive amount of work to address the reviewer's comments.

I have few remaining minor comments:

Introduction: reduce the summary of the results to 2-3 sentences. The current section is redundant with the initial part of the conclusions.

Page 9: remove the sentence "In particular, the results also indicated that the HDDA NBs could reverse the multidrug resistance of cancer"

Supp Fig 14: cell viability over 100% does not make sense. Change the axis of the graph.

Page 13: change verb to present form "The details were shown..."

Page 15: the authors refer to Fig 1j, but there is none.

Page 19: change verb to present from "tissue slices from different groups were illustrated..."

Page 21 and Supp Fig 39: the authors state "The confocal fluorescence images illustrated that the HDDA treatment could significantly increase the number of the intratumoral helper T cells (CD3+CD4+) and cytotoxic T cells (CD3+CD8+) (Supplementary Fig. 39)". However there is no quantification and the CD3+CD8+ is not very convincing.

Page 21: Refer to Fig 4j for Granzyme B

Responses to Reviewer 1:

General comment: The revision has addressed most of the reviewers' comments, and the current version is acceptable for publication in NC.

Author reply: We are very happy to get the approval of this respective reviewer! Thank you so much for your recognition and your very professional comments!

Responses to Reviewer 2:

General comment: In the new version of the manuscript "In situ massive generation of micrometer-sized tumor cell-derived vesicles as autologous cancer vaccines for boosting systemic immune responses", the authors have adequately responded to most of my concerns about the nature and the role of tumor cell-derived vesicles (HMTVs) by performing several additional experiments.

Author reply: We deeply thank this reviewer for his/her recognition of this version of the manuscript. We also want to express our great appreciation to this reviewer for the precious time, invaluable expertise, and superb professionalism the respected reviewer has put in improving the quality of our manuscript! Besides, we have tried our best to modify our manuscript following the comments blow. Thank you so much for your very professional comments!

Comment 1: One thing that for me still doesn't make sense is that HMTVs induce apoptosis of target cells, but that according to the authors, HMTVs that are the result of this process are not apoptotic vesicles. How is this logical and what strong data do the authors have to support this? In the EV field, the exact definition of apoptotic EVs is still missing. With new knowledge there is an acceptance that apoptotic EVs can be very heterogeneous in size, and there is also no consensus about their composition.

Author reply: We sincerely appreciate the reviewer for raising this very important comment! After reading some references about apoptotic EVs, we deeply agree with this reviewer that we do not have strong data to confirm that the HMTVs are not apoptotic EVs. As suggested, we have changed our claim in the revised manuscript as suggested (Page 11). Thanks a lot for pointing out this flaw of our manuscript.

Comment 2: On another note, I would like to comment on the structure of the manuscript. The manuscript is very long and main sections, like Results or Discussion are not clearly indicated. The authors also included the discussion of their data into the Introduction (in paragraph 3 and 4), instead of reporting on what is already known/published on the topic (including on

conundrum about apoptotic EVs). The new results on the proteome of HMVs are described in the Methods section and not in the Results. The authors sometimes make statements/conclusions in the main text, that are not supported by results or by reference to published data (e.g. p.7: "The different internalization endocytosis levels may be attributed to the various CD44 expression levels of these cell lines."; p. 9: "The differences in the sizes of the produced HMVs might be attributed to the different tolerance to HDDA of these cell lines. In particular, the results also indicated that the HDDA NBs could reverse the multidrug resistance of cancer." etc...).

The authors should improve on this and at the same time try to improve the readability of the manuscript, as at the moment it is very hard to follow.

Author reply: We deeply appreciate this respected reviewer for raising these very helpful suggestions! We have carefully modified the manuscript following these suggestions and improve the readability of the manuscript by compressed some statements in the Introduction and the Results sections. Thanks a lot!

Responses to Reviewer 3:

General comment: This manuscript has been revised according to reviewers comments, so I think it can be accepted for publication without change.

Author reply: We deeply appreciate this respected reviewer for his/her recognition of our manuscript. Thanks a lot!

Responses to Reviewer 4:

General comment: The authors performed an impressive amount of work to address the reviewer's comments.

Author reply: We sincerely appreciate this respected reviewer for his/her very professional comments and suggestions, which help us to improve the quality of the manuscript significantly! We are also very happy for your recognition. As suggested, we have carefully modified the manuscript following the comments below. Thank you so much!

Comment 1: I have few remaining minor comments:

Introduction: reduce the summary of the results to 2-3 sentences. The current section is redundant with the initial part of the conclusions.

Page 9: remove the sentence "In particular, the results also indicated that the HDDA NBs could reverse the multidrug resistance of cancer"

Supp Fig 14: cell viability over 100% does not make sense. Change the axis of the graph.

Page 13: change verb to present form "The details were shown..."

Page 15: the authors refer to Fig 1j, but there is none.

Page 19: change verb to present from "tissue slices from different groups were illustrated..."

Page 21 and Supp Fig 39: the authors state "The confocal fluorescence images illustrated that the HDDA treatment could significantly increase the number of the intratumoral helper T cells (CD3+CD4+) and cytotoxic T cells (CD3+CD8+) (Supplementary Fig. 39)". However there is no quantification and the CD3+CD8+ is not very convincing.

Page 21: Refer to Fig 4j for Granzyme B

Author reply: We sincerely thank this respected reviewer for raising these very helpful suggestions and for pointing out these flaws in the manuscript! We have carefully modified the manuscript following these suggestions, and all of the mentioned flaws have been corrected at the corresponding places in the revised manuscript. Thanks a lot!